# Cytotoxic T cells swarm by homotypic chemokine signalling

Jorge Luis Galeano Niño[1,2†], Sophie V Pageon[1,2†], Szun S Tay[1,2†], Feyza Colakoglu[1,2], Daryan Kempe[1,2], Jack Hywood[3], Jessica K Mazalo[1,2], James Cremasco[1,2], Matt A Govendir[1,2], Laura F Dagley[4,5], Kenneth Hsu[6], Simone Rizzetto[2,7], Jerzy Zieba[2,8], Gregory Rice[9], Victoria Prior[6,10], Geraldine M O'Neill[6,10], Richard J Williams[11,12], David R Nisbet[11,13], Belinda Kramer[6], Andrew I Webb[4,5], Fabio Luciani[2,7], Mark N Read[14], Maté Biro[1,2]*

[1]EMBL Australia, Single Molecule Science node, University of New South Wales, Sydney, Australia; [2]School of Medical Sciences, Faculty of Medicine, University of New South Wales, Sydney, Australia; [3]Sydney Medical School, The University of Sydney, Sydney, Australia; [4]The Walter and Eliza Hall Institute of Medical Research, Melbourne, Australia; [5]Department of Medical Biology, University of Melbourne, Melbourne, Australia; [6]Children's Cancer Research Unit, The Children's Hospital at Westmead, Sydney, Australia; [7]The Kirby Institute for Infection and Immunity in Society, UNSW, Sydney, Australia; [8]Neuroscience Research Australia (NeuRA), Randwick, Australia; [9]Department of Statistics and Actuarial Science, University of Waterloo, Waterloo, Canada; [10]Discipline of Child and Adolescent Health, University of Sydney, Sydney, Australia; [11] Biofab3D, St. Vincent's Hospital, Melbourne, Australia; [12]Institute for Innovation in Mental and Physical Health and Clinical Translation (iMPACT), School of Medicine, Deakin University, Victoria, Australia; [13]Advanced Biomaterials Lab, Research School of Engineering, ANU, Canberra, Australia; [14]School of Computer Science, Westmead Initiative, and Charles Perkins Centre, University of Sydney, Sydney, Australia

*For correspondence:
m.biro@unsw.edu.au

[†]These authors contributed equally to this work

Competing interests: The authors declare that no competing interests exist.

**Abstract** Cytotoxic T lymphocytes (CTLs) are thought to arrive at target sites either via random search or following signals by other leukocytes. Here, we reveal independent emergent behaviour in CTL populations attacking tumour masses. Primary murine CTLs coordinate their migration in a process reminiscent of the swarming observed in neutrophils. CTLs engaging cognate targets accelerate the recruitment of distant T cells through long-range homotypic signalling, in part mediated via the diffusion of chemokines CCL3 and CCL4. Newly arriving CTLs augment the chemotactic signal, further accelerating mass recruitment in a positive feedback loop. Activated effector human T cells and chimeric antigen receptor (CAR) T cells similarly employ intra-population signalling to drive rapid convergence. Thus, CTLs recognising a cognate target can induce a localised mass response by amplifying the direct recruitment of additional T cells independently of other leukocytes.

## Introduction

Cytotoxic T lymphocytes (CTLs) constitutively migrate as single cells in search of infected or malignant cells (*Weninger et al., 2014*). CTLs are key effectors of adoptive cell transfer immunotherapies (*Guedan et al., 2019*), but their efficacy remains limited with solid tumours (*van der Woude et al., 2017*), which they infiltrate in insufficient numbers (*Galon et al., 2006*). Thus far, CTLs have been

**eLife digest** Immune cells known as cytotoxic T lymphocytes, or CTLs for short, move around the body searching for infected or damaged cells that may cause harm. Once these specialised killer cells identify a target, they launch an attack, removing the harmful cell from the body. CTLs can also recognise and eliminate cancer cells, and can be infused into cancer patients as a form of treatment called adoptive cell transfer immunotherapy. Unfortunately, this kind of treatment does not yet work well on solid tumours because the immune cells often do not infiltrate them sufficiently.

It is thought that CTLs arrive at their targets either by randomly searching or by following chemicals secreted by other immune cells. However, the methods used to map the movement of these killer cells have made it difficult to determine how populations of CTLs coordinate their behaviour independently of other cells in the immune system. To overcome this barrier, Galeano Niño, Pageon, Tay et al. employed a three-dimensional model known as a tumouroid embedded in a matrix of proteins, which mimics the tissue environment of a real tumour in the laboratory. These models were used to track the movement of CTLs extracted from mice and humans, as well as human T cells engineered to recognise cancer cells.

The experiments showed that when a CTL identifies a tumour cell, it releases chemical signals known as chemokines, which attract other CTLs and recruit them to the target site. Further experiments and computer simulations revealed that as the number of CTLs arriving at the target site increases, this amplifies the chemokine signal being secreted, resulting in more and more CTLs being attracted to the tumour. Other human T cells that had been engineered to recognize cancer cells were also found to employ this method of mass recruitment, and collectively 'swarm' towards targeted tumours.

These findings shed new light on how CTLs work together to attack a target. It is possible that exploiting the mechanism used by CTLs could help improve the efficiency of tumour-targeting immunotherapies. However, further studies are needed to determine whether these findings can be applied to solid tumours in cancer patients.

thought to arrive at target sites either via random search (*Krummel et al., 2016*) or following signals by other cell types (*Feig et al., 2013*; *Harlin et al., 2009*).

Intravital imaging studies have been instrumental in uncovering some of the complex migration patterns of CTLs at various stages of an immune response. They have revealed, for instance, that CTLs employ highly evolved cell-intrinsic search strategies that are more efficient than Brownian motion (*Harris et al., 2012*). During priming in lymph nodes, CD8[+] T cells can follow local chemoattractant gradients to migrate directionally towards sites of antigen presentation (*Castellino et al., 2006*; *Hickman et al., 2011*; *Hugues et al., 2007*). In the tumour microenvironment (TME), the movements of CTLs and their interplay with various cells have been unveiled (*Boissonnas et al., 2007*; *Deguine et al., 2010*). Recent studies indicate that lymphocytes can recruit each other indirectly into tumours; natural killer (NK) lymphocytes produce chemokines that attract dendritic cells (*Böttcher et al., 2018*), which in turn can recruit CTLs (*Spranger et al., 2017*). Although intravital microscopy enables imaging of cellular interactions in the TME in situ, it is typically restricted to relatively short imaging periods and sub-millimetre fields of view (*Gabriel et al., 2018*). Furthermore, the inherent complexities of the TME, its constituent cell populations and its biochemical landscape have limited our ability to uncover the contribution of an individual immune subset or signalling mechanism to progressive, large-scale phenomena.

Here, we developed a 3D tumouroid model and in silico simulations to reveal independent collective behaviour in CTL populations attacking tumour masses. We show that CTLs coordinate their migration in a process reminiscent of the swarming observed in insects (*Avitabile et al., 1975*) and neutrophils (*Lämmermann et al., 2013*). CTLs engaging tumour targets induce rapid chemotaxis in distant T cells through homotypic chemokine signalling. Newly arriving CTLs augment the chemotactic signal, further accelerating mass recruitment in a positive feedback loop. Furthermore, we show that local chemokine delivery triggers directed CTL movement through dense tumour tissue in vivo, and sustained secretion of CCL3 and CCL4 from tumours promotes CTL recruitment. Human effector and chimeric antigen receptor (CAR) T cells similarly employ intra-population signalling to drive

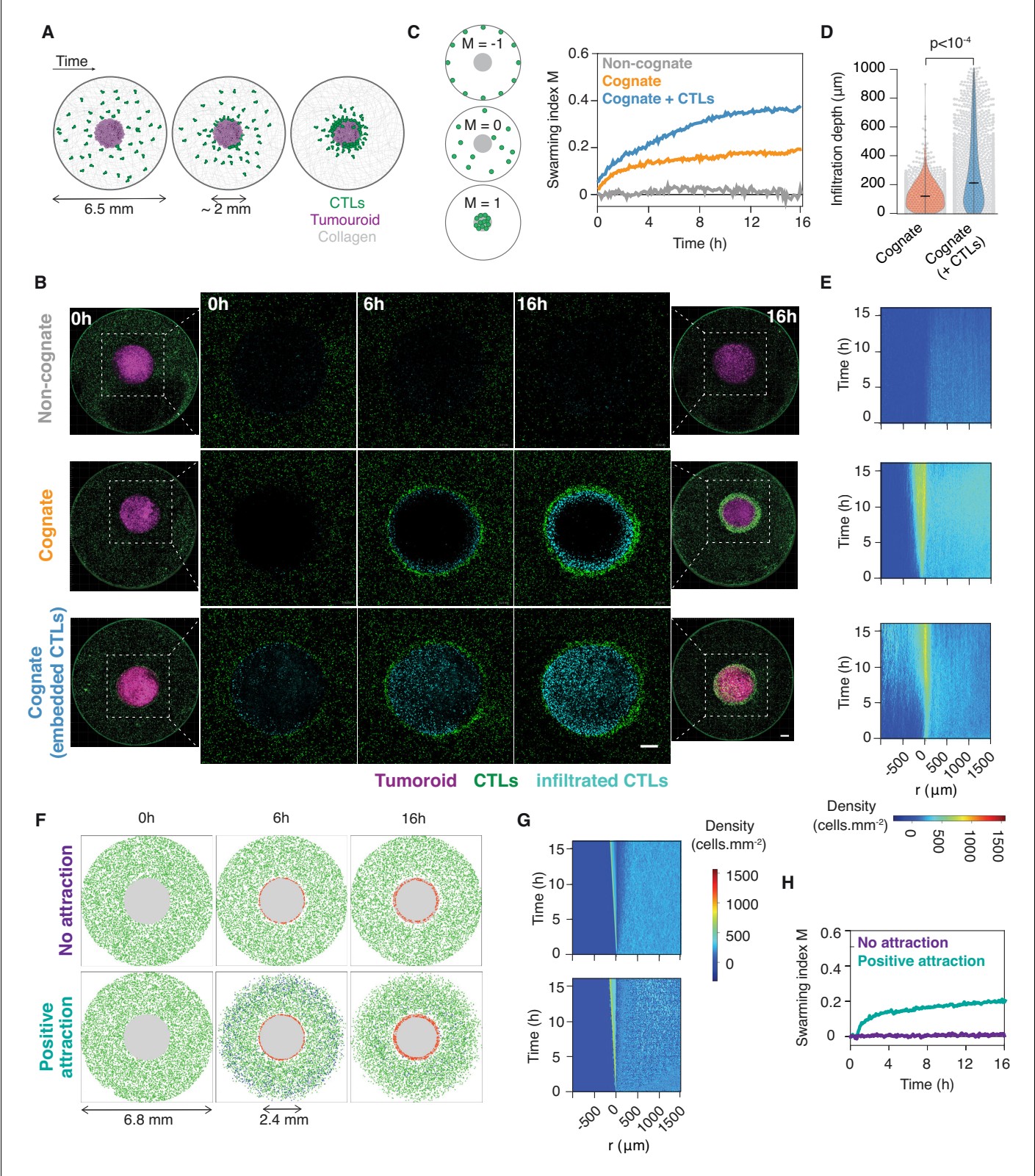

**Figure 1.** CTLs swarm as they attack tumour masses. (**A**) Schematic of ex vivo tumouroid model: a high-density mass of tumour cells and collagen (tumouroid, magenta) is surrounded by a 3D collagen matrix containing dispersed CTLs (green) in a closed well with a 6.5 mm diameter and is imaged live over time. (**B**) Representative confocal images from three independent experiments showing CTLs responding to a non-cognate EL4 tumouroid (top), a cognate antigen-presenting tumouroid comprising EL4 pulsed with SIINFEKL (middle) or a cognate tumouroid pre-embedded with tumour-

*Figure 1 continued on next page*

*Figure 1 continued*

reactive CTLs (bottom). Insets show the proportion of CTL infiltrates for each condition as indicated. Scale bars: 500 μm. (**C**) Left: Schematic showing the swarming index 'M' that quantifies the evolution of CTL spatial distribution around the tumouroid over time. A value of −1 denotes perfect anti-swarming, that is all cells evenly distributed on the well perimeter; a value of 0 corresponds to a uniform distribution of CTLs outside the tumouroid; and all CTLs having infiltrated the tumouroid yields a value of 1. Right: Quantification of CTL convergence at the tumour masses shown in (**B**) via the swarming index. (**D**) Infiltration depth of CTLs into tumouroid (measured as shortest distance from tumouroid edge) for cognate tumouroids with and without pre-embedded tumour-reactive CTLs. Violin plots show the distribution of pooled data from three independent experiments, black bars represent medians, grey dots are individual data points. p-values from Mann Whitney U test. (**E**) Density kymographs depicting cell density over space and time for data shown in (**B**). r: distance from tumouroid edge. (**F**) In silico simulations of T cells surrounding a tumour mass (grey) as agent-based processes. No attraction: agents (green) reach the tumouroid by random non-directed movement only and remain within following contact (red). 'Positive attraction': agents arriving at the tumouroid (red) secrete chemoattractant that induces directional motility towards the tumouroid in surrounding agents within range (blue), which can revert to non-directed motility in excess chemoattractant concentration. (**G, H**) Swarming index and density kymographs for in silico simulations.

The online version of this article includes the following source data and figure supplement(s) for figure 1:

**Source data 1.** Source data file for *Figure 1*.
**Figure supplement 1.** In silico model of CTL recruitment towards sites of tumour engagement.
**Figure supplement 1—source data 1.** Source data file for *Figure 1—figure supplement 1*.

rapid convergence. Our findings provide insights into how CTL populations amplify directed recruitment to an effector site independently of other leukocytes.

## Results

### CTLs swarm towards cognate tumouroids

We sought to investigate the population-wide movements and signals mediating interactions between CTLs during tumour clearance. To this end, we developed an ex vivo model enabling us to study the large-scale movements of primary CTLs around solid tumouroids embedded in three-dimensional (3D) collagen matrices (*Figure 1A*). We used primary murine CTLs isolated from OT1 (*Hogquist et al., 1994*) and gBT1 (*Coles et al., 2003*) T cell receptor transgenic mice that recognise ovalbumin (SIINFEKL) or herpes simplex virus glycoprotein B (SSIEFARL) residues, respectively, both in the context of the H-2K$^b$ class I major histocompatibility complex. CTLs engaging a cognate tumouroid were rapidly recruited to its edge, where they accumulated over time (*Figure 1B* and *Video 1*). The marked accumulation of CTLs at the edge of the tumouroid is reminiscent of

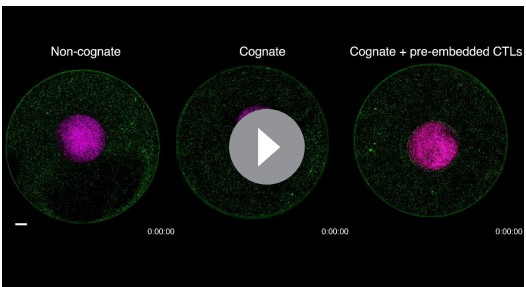

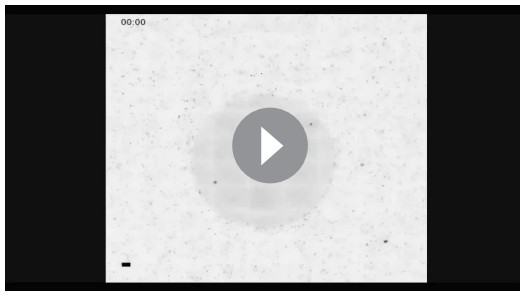

**Video 1.** CTLs swarm around a cognate tumouroid. (Total view) Long-term time-lapse imaging (40 hr) of OT1$^{GFP}$ CTLs (dark) embedded in 3D collagen matrix around cognate EL4 tumouroid (light grey disc). (Inset view) Brightfield (BF) and fluorescence (FL, OT1$^{GFP}$) imaging of OT1 CTLs amassing at tumouroid edge. Scale bars, 100 μm. Time in h:min.
https://elifesciences.org/articles/56554#video1

**Video 2.** CTLs infiltrate deep into cognate tumour masses embedded with tumour-reactive CTLs. Long-term (16 hr) time-lapse imaging of OT1 CTLs (green) embedded in collagen matrix around EL4 tumouroid (magenta). (Left) Non-cognate tumouroid; no significant accumulation of CTLs at tumouroid. (Centre) Cognate tumouroid; rapid recruitment and swarming of CTLs around tumouroid edge. (Right) Cognate tumouroid embedded with tumour-reactive CTLs (red); swarming and deep infiltration of tumouroid by CTLs. Scale bar, 500 μm. Time in h:min:s.
https://elifesciences.org/articles/56554#video2

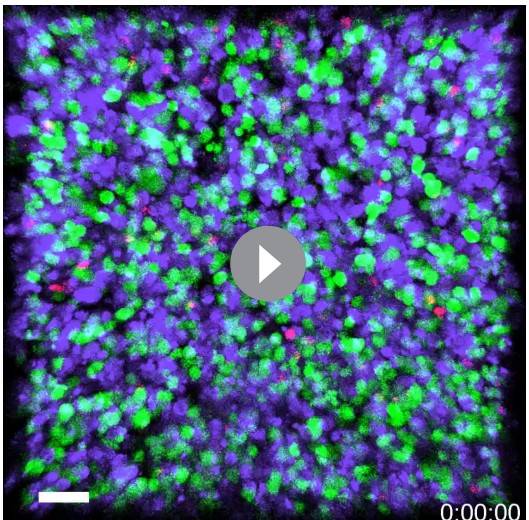

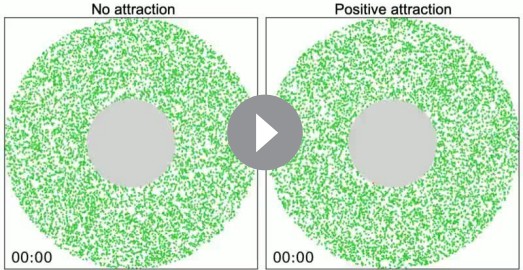

**Video 4.** In silico simulations of CTL movements around cognate tumouroid as agent-based processes. Motility models bootstrapped from experimental data. Gray: tumouroid; green: agents migrating without influence of chemoattractive signal; blue: agents encountering chemoattractant concentrations above threshold migrate directionally; red: agents infiltrating tumouroid. (Left) Agents arrive at tumouroid by random (non-directed) movement only. (Right) Agents arriving at tumouroid secrete diffusing chemoattractant that induces spatial bias in the movement of other agents towards tumouroid. Time in h:min.
https://elifesciences.org/articles/56554#video4

**Video 3.** Tumour-reactive CTLs actively eliminate cognate tumour cells within the tumouroid. 3D time-lapse (3 hr) imaging of OT1 CTLs (green) embedded in a tumouroid composed of cognate tumour cells (purple) and dense collagen matrix. Propidium iodide was added to the medium to detect lysed cells in real time (red). Scale bar, 50 µm. Time in h:min:s.
https://elifesciences.org/articles/56554#video3

swarming, the coordinated collective convergence of large numbers of self-propelled individuals, which has been observed in as varied biological systems (*Okubo, 1986*) as mammals, birds, fish, insects, and, at the cellular level, neutrophils (*Lämmermann et al., 2013*). This behaviour was not observed with control tumouroids lacking cognate antigen (*Figure 1B* and *Video 2*).

Cell movements were quantified using a novel swarming index 'M' (see Materials and methods; *Figure 1C*). When an additional CTL population was embedded in the cognate tumouroid where it actively eliminated tumour target cells (*Video 3*), the surrounding CTLs amassed at the tumouroid (*Figure 1B*) and infiltrated it to a greater extent (*Figure 1D,E* and *Video 2*). This enhanced infiltration was also observed into masses constituted of cognate antigen-coated beads, which, in contrast to tumour cells, do not progressively give way to additional physical space due to lysis by CTLs (*Figure 1—figure supplement 1A–E*).

In order to assess whether the observed accumulation of CTLs around tumour masses involves emergent behaviour or is merely a cumulative result of randomly scanning CTLs arresting upon cognate antigen recognition, we modelled and simulated our experiments as agent-based processes in silico. Simulated cell motility characteristics were sampled from experimental data by bootstrapping (Materials and methods; *Figure 1—figure supplement 1F,G*). We simulated the spatiotemporal scales of our experiments, with a diffusive chemoattractive signal progressively amplified by agents arriving at the tumouroid and inducing a directional bias in distant agent movements. The introduction of a desensitisation threshold in chemokine concentration, above which agents revert to unbiased motion, was necessary to recapitulate our experimental observations (*Figure 1F–G*, *Figure 1—figure supplement 1H* and *Video 4*).

## CTLs migrate rapidly and highly directionally towards sites of cognate tumour cell engagement

To further characterise how CTLs are recruited to the tumour mass, we next used an assay enabling tracking of individual CTL movements at higher spatiotemporal resolution (*Video 5*) relative to a tumouroid exposing a straight interface in a constant direction from adjacent CTLs (*Figure 2A*). Using this approach, we found that surrounding CTLs migrate rapidly and highly directionally towards the edge of a cognate tumour mass (*Figure 2B*), containing pre-embedded CTLs at 1:1 or 1:5 ratios with tumour cells, or in the complete absence of pre-embedded CTLs (*Figure 2B*). By

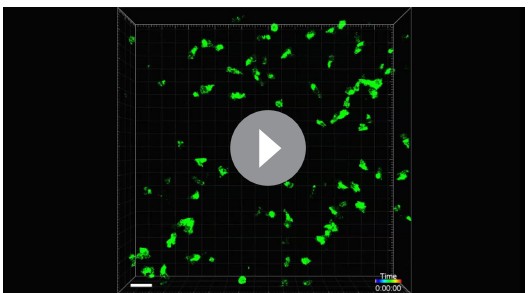

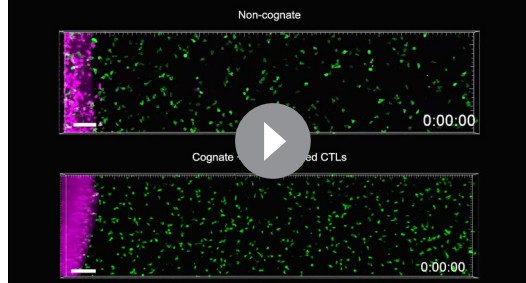

**Video 5.** Tracking individual CTL movements in 3D at high temporal resolution. 3D timelapse (1 hr) imaging of OT1 CTLs (green) embedded in collagen matrix near cognate tumouroid exposing straight interface (out of view to the left) with embedded tumour-reactive CTLs. Track colour, time; scale bar, 50 µm.
https://elifesciences.org/articles/56554#video5

**Video 6.** Tracking individual CTL movements up to 2 mm from tumouroid. 3D time-lapse (2 hr) imaging of OT1 CTLs (green) embedded in collagen matrix adjacent to a tumouroid (magenta). (Top) Non-cognate tumouroid. (Bottom) Cognate tumouroid with embedded tumour-reactive CTLs. Scale bars, 200 µm. Time in h:min:s.
https://elifesciences.org/articles/56554#video6

contrast, CTLs adjacent to control tumouroids exhibited no chemotaxis, albeit, interestingly, some chemokinesis (enhanced speed) (*Figure 2B*).

## CTLs engage in homotypic recruitment via a diffusive signal

The above results indicate that a factor diffusing from the site of engagement leads to distal recruitment. Indeed, cell-free supernatant from the co-incubation of CTLs with cognate tumour cells (cognate supernatant) attracts other CTLs in a transmigration assay (*Figure 2—figure supplement 1A*) and thus contains the soluble chemoattractants mediating the recruitment (*Figure 2—figure supplement 1B,C*). Interestingly, 10-fold dilutions of cognate supernatants also induced CTL transmigration (*Figure 2—figure supplement 1B*), indicating that the CTL response to the soluble chemotactic factors is highly sensitive.

To distinguish whether the factors originate from CTLs or lysed tumour cells, we replaced tumour cells with polystyrene beads coated with cognate antigen (*Figure 2C*) and found that recruitment of CTLs was preserved (*Figure 2C,D*). CTLs engaging cognate targets therefore produce factors that induce distal recruitment via homotypic signalling. Combined, our experimental and simulation data reveal emergence in CTL mass recruitment, which is progressively amplified by the arrival of further CTLs that contribute to a diffusive homotypic signal.

We next tested whether the homotypic signal extends to T cells of different antigen specificities. Tumouroids containing monospecific CTLs and their cognate tumour targets (*Figure 2—figure supplement 1D*) are equally proficient at recruiting T cells of the same (*Figure 2E*), different (*Figure 2F*) and polyclonal (*Figure 2G*) specificity. No recruitment is observed towards tumouroids containing CTLs engaging non-cognate target cells. Therefore, whilst production of the homotypic signal requires recognition of specific cognate antigen, the signal recruits CTLs irrespective of the T cell receptor (TCR) they express. CTLs of unrelated specificity would however not be able to contribute to the amplification of the signal upon arrival at the tumour mass.

## Long-range diffusing gradients emanate from CTLs engaging cognate tumouroids

Next, we studied the movements of CTLs up to 1.6 mm from tumouroids (*Figure 3—figure supplement 1A*). At the onset of the experiment (0 hr), CTLs were distributed uniformly adjacent to both cognate and non-cognate tumouroids (*Figure 3A,B* and *Figure 3—figure supplement 1B,C*). After 2 hr, the distribution of CTLs was biased towards the cognate tumouroid (*Figure 3A,B*), whereas the CTL distribution adjacent to a control tumouroid remained unchanged (*Figure 3—figure supplement 1B,C*). Cell tracking analysis revealed that even the most distant CTLs show directional movement towards the cognate tumouroid (*Figure 3C,D* and *Video 6*) and reveals a moving wave of chemoattraction over time (*Figure 3E*). Cells close to the tumouroid start moving towards it at early

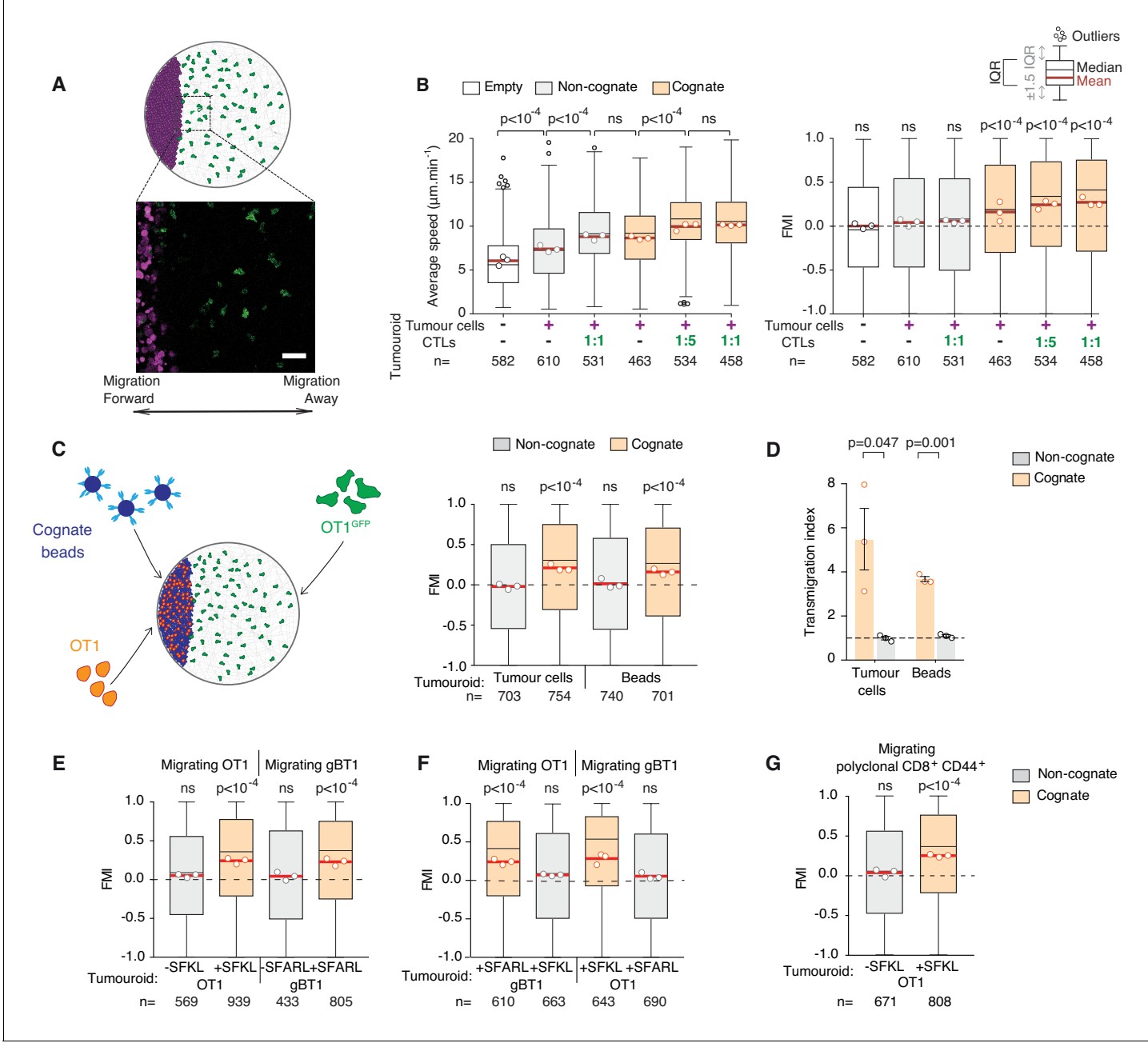

**Figure 2.** CTLs migrate rapidly and directionally towards sites of cognate target engagement due to a diffusing homotypic signal. (A) Schematic of assay enabling 3D tracking of CTL movements relative to a tumouroid exposing a straight interface (scale bar: 50 μm). (B) Average speed and forward migration index (FMI) of CTLs adjacent to tumouroids containing cognate (SIINFEKL-pulsed) EL4 tumour cells or non-cognate (unpulsed) EL4, and without (-) or with pre-embedded tumour-reactive CTLs at indicated CTL:tumour cell ratios. (C) Left: Schematic showing H-2K$^b$/SIINFEKL cognate antigen-coated beads co-embedded with CTLs (OT1) in a dense mass, with the migration of adjacent fluorescent CTLs (OT1$^{GFP}$) tracked in 3D over time. Right: FMI of CTLs migrating towards tumouroids containing OT1 CTLs and non-cognate or cognate tumour cells or beads. (D) Transmigration of CTLs towards supernatant obtained from CTLs conjugated with non-cognate or cognate cells and beads. Bars: mean from three independent experiments (data points). Error bars: SEM. p-values for comparisons of cognate and non-cognate conditions by t tests. (E) FMI of OT1 or gBT1 CTLs migrating towards tumouroids pre-embedded at a 1:1 ratio with OT1 or gBT1 CTLs respectively, showing 'self-recruitment' of CTLs. (F) FMI of OT1 or gBT1 CTLs migrating towards cognate tumoroids (EL4 pulsed with SIINFEKL or SSIEFARL as indicated) pre-embedded at a 1:1 ratio with gBT1 or OT1 CTLs, showing 'cross-recruitment' of CTLs of different antigen specificity towards tumour-reactive CTLs. Tumouroid densities are kept constant, and when pre-embedded with CTLs contain only half the number of tumour cells compared to masses in 2B constituted exclusively of tumour cells. (G) FMI of polyclonal CD8$^+$CD44$^+$ T cells migrating towards tumouroids containing OT1 CTLs with cognate or non-cognate tumour cells. Box-whiskers indicate medians and the interquartile range (IQR) with outliers outside whiskers. Red bars: mean of pooled data from three independent experiments. Data

*Figure 2 continued on next page*

Figure 2 continued

points: mean of each individual experiment. n: number of tracks. ns: p>0.05, where FMI in (B) to (G) were compared to a theoretical median of 0 using the two-tailed Wilcoxon signed rank test and average speeds in (B) were compared using the Kruskal-Wallis test followed by Dunn's multiple comparisons tests.

The online version of this article includes the following source data and figure supplement(s) for figure 2:

**Source data 1.** Source data file for *Figure 2*.
**Figure supplement 1.** CTLs transmigrate towards cognate supernatants.
**Figure supplement 1—source data 1.** Source data file for *Figure 2—figure supplement 1*.

timepoints but lose some of their directionality over time (*Figure 3E*). It has previously been shown that chemotactic responses can be lost under excess local concentrations of ligand (*Lim et al., 2018*), which could account for the behaviour observed here. For cells furthest from the tumouroid, directional migration commences at later timepoints (*Figure 3E*). No such pattern is observed with a control tumouroid, where cells at all distances exhibit undirected migration (*Figure 3C–E* and *Video 6*). These results indicate that a gradient of diffusing factors is established following cognate interactions between CTLs and tumour cells to induce the long-range chemoattraction of other T cells.

## CTL recruitment is driven by secretion of CCL3 and CCL4

To identify the molecular mediators of the chemoattraction, we first tested CTL migration in the presence of pertussis toxin (PTX), an inhibitor of $G\alpha_i$-protein-coupled receptors (GPCRs). PTX inhibits the enhanced average speed and directional migration of CTLs towards cognate tumouroids (*Figure 4A*), indicating that GPCR ligands mediate the chemoattractive signal. This result is consistent with previous work where the formation of small CTL clusters around malaria-infected hepatocytes required GPCR signalling (*Cockburn et al., 2013*). Unlike neutrophils that employ lipid signalling via leukotriene B4 to swarm (*Lämmermann et al., 2013*), this result suggests that chemokines, which are sensed by GPCRs, underpin homotypic signalling in CTLs. We then compared the transcriptomes of CTLs engaging cognate and non-cognate targets (dataset available in *Supplementary file 1*). Differentially expressed genes were filtered for secreted factors, among which the following GPCR ligands were identified: Chemokine (C-C motif) ligand 1 (CCL1), CCL3, CCL4, CCL9, and X-C motif chemokine ligand 1 (XCL1) (*Figure 4B*); their upregulation was further validated by quantitative reverse transcription PCR (*Figure 4—figure supplement 1A,B*). Mass-spectrometry-based proteomics analysis of cognate versus non-cognate secretomes reveals that CCL1, CCL3, CCL4, and XCL1 are more abundant in cognate supernatant (*Figure 4C* and *Supplementary file 2*), confirmed by quantitative detection assays (*Figure 4—figure supplement 1C*). The recombinant chemokines CCL3, CCL4, CCL5, and CXCL12 could attract CTLs in transwell migration assays (*Figure 4—figure supplement 1D*), in each case effectively inhibited by the corresponding neutralising antibody (*Figure 4—figure supplement 1E*). However, only CCL3 inhibition consistently disrupted CTL transmigration in transwell chambers towards cognate supernatants (*Figure 4—figure supplement 1F*). Directional migration towards a cognate tumouroid in more physiological 3D matrices was abolished only when both CCL3 and CCL4 were blocked using neutralising antibodies (*Figure 4—figure supplement 1G*), indicating that these two chemokines act redundantly in the homotypic attraction. Therefore, CCL3 and CCL4 constitute the diffusive homotypic signal that attracts distant CTLs during target engagement. Secreted CCL3 forms a gradient from the site of antigen-specific target engagement, visualised by in situ capture and staining (*Figure 4—figure supplement 2A–C*). Strikingly, CTLs sustain their ability to secrete CCL3 and CCL4 (*Figure 4—figure supplement 1H*), and to recruit distant T cells (*Figure 4—figure supplement 1I*), even after disengagement from cognate targets.

To confirm that CCL3 and CCL4 secretion are sufficient to induce chemoattraction in distant CTLs, we engineered tumour cells that constitutively secrete both chemokines (*Figure 4—figure supplement 3A,B*), or CCL3 or CCL4 alone. Secreting tumouroids induced enhanced rapid directional motility in CTLs (*Figure 4D*), swarming and infiltration (*Figure 4—figure supplement 3C–H*). CTLs infiltrate CCL3/CCL4-secreting cognate tumouroids as efficiently as tumouroids within which CTLs are actively engaging cognate targets (*Figure 1D*). In the absence of cognate antigen, CTLs do

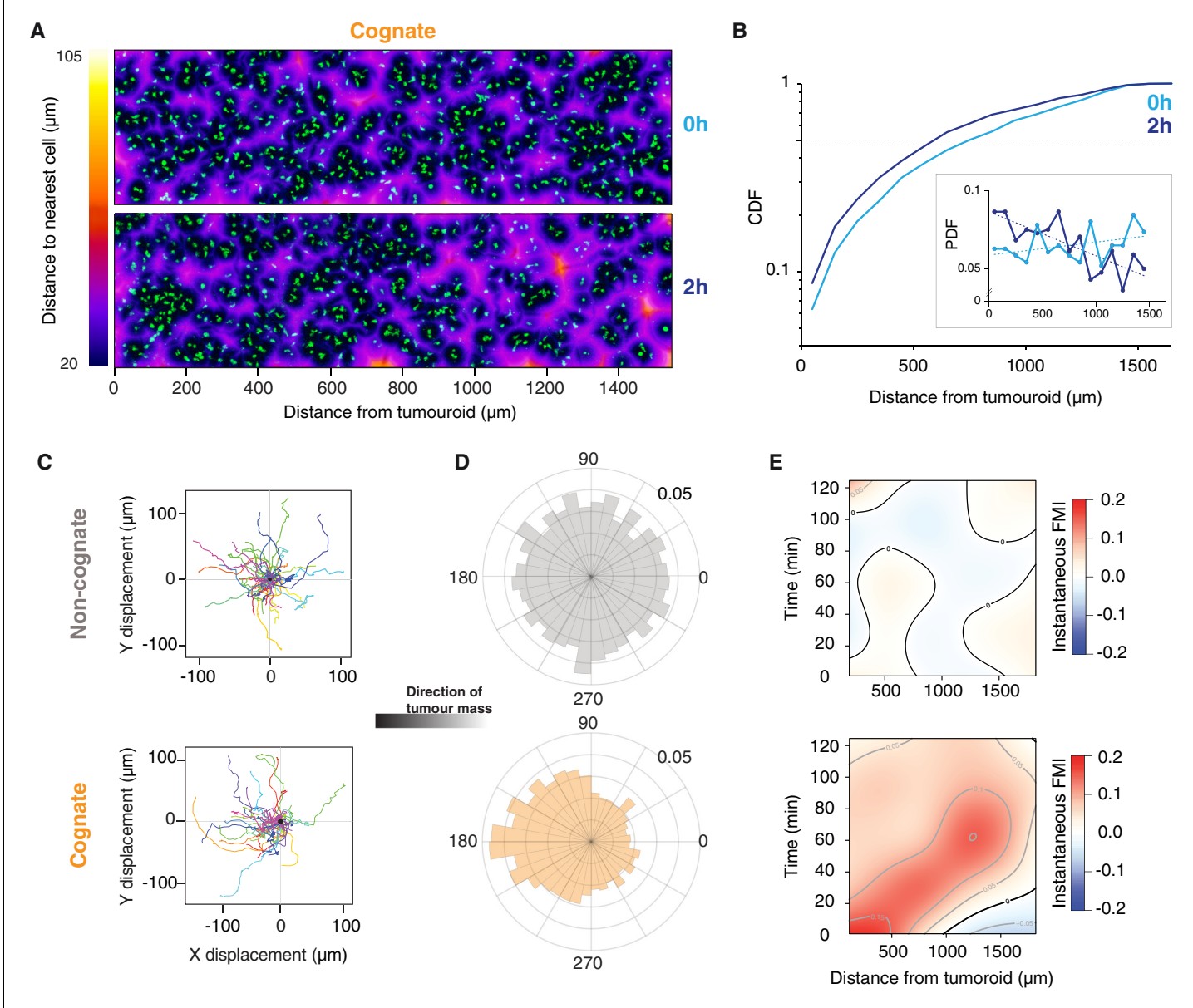

**Figure 3.** Engagement of cognate tumouroid gives rise to a diffusing wave of chemoattraction. (**A**) Confocal images showing the distribution of CTLs (green) in relation to a SIINFEKL-pulsed EL4 cognate tumouroid at 0 and 2 hr. Pseudocolour: distance to nearest cell in μm. (**B**) Cumulative distribution function (CDF) of nearest distances of CTLs to a cognate tumouroid at 0 hr and 2 hr. Insert shows the corresponding probability density function (PDF), where dashed lines are linear fits. (**C**) Randomly selected representative tracks of CTLs migrating towards a non-cognate (top) or cognate (bottom) tumouroid with pre-embedded tumour-reactive CTLs (located towards -X); outset of all tracks centred on origin. (**D**) Angular histograms of CTL track displacements. 0°: away from tumour, 180°: towards tumour. Data pooled from three independent experiments (n = 4947 for non-cognate; top, n = 3926 tracks for cognate; bottom). (**E**) Kymographs of instantaneous FMI (based on displacement vector at each timeframe) for data shown in (**D**). The online version of this article includes the following figure supplement(s) for figure 3:

**Figure supplement 1.** Long-range attraction assay in a non-cognate setting.

not stop at the edge of secreting tumouroids and thus infiltrate them deeper (*Figure 4—figure supplement 3E and H*). We next established an in vivo model to investigate if CCL3/CCL4-secretion influences endogenous leukocyte recruitment to tumours engrafted in mice (*Figure 4—figure supplement 4*), and showed that CCL3/CCL4-secreting tumours consistently recruit more endogenous NK cells than contralateral control tumours (*Figure 4—figure supplement 4B*).

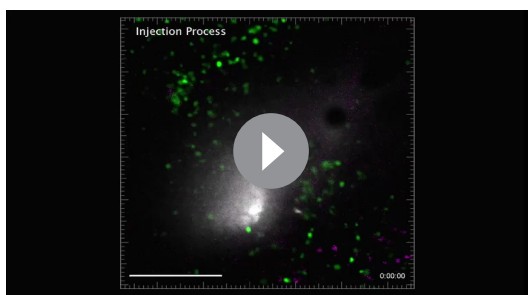

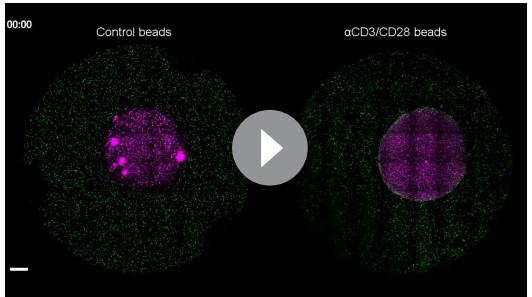

**Video 7.** Intravital imaging reveals CTL recruitment to site of intratumoural CCL3 injection. (Injection Process) 2D time-lapse imaging of intratumoural injection of shear-thinning hydrogel containing recombinant CCL3 and Dextran-AF647 (white). (Injection Site) 2D fluorescence confocal image showing tumour tissue with adoptively transferred WT CTLs (green) and CCR5$^{-/-}$ CTLs (magenta) immediately following injection. (Post Injection) 3D long-term (5.6 hr) intravital imaging around intratumoural injection site. Scale bars, 200 μm.
https://elifesciences.org/articles/56554#video7

**Video 8.** Activated primary human T cells engage in homotypic recruitment. Long-term (14 hr) time-lapse imaging of primary human T cells (green) embedded in collagen around bead-stimulated human T cells (magenta). (Left) Control (IgG-coated) beads. (Right) Activating (anti-CD3/CD28-coated) beads. Scale bar, 500 μm. Time in h:min:s.
https://elifesciences.org/articles/56554#video8

## CTL swarming is orchestrated via chemokine receptor CCR5

We next sought to identify the receptor mediating the chemoattractive signal. To this end, we tested the chemokine receptors CCR1, CCR2, CCR3, and CCR5 that have been associated with CCL3 or CCL4 sensing, and which are all expressed by CTLs (*Figure 5—figure supplement 1A*). Pharmacological inhibition of CCR1, CCR2, or CCR3 (*Figure 5—figure supplement 1B,C*) did not have any effect, whereas targeting CCR5 with Maraviroc or the CCR2/CCR5 dual antagonist Cenicriviroc abrogated the homotypic recruitment of CTLs (*Figure 5—figure supplement 1B,C*). Furthermore, polyclonal T cells isolated from Ccr5 knockout mice (Ccr5$^{-/-}$; *Figure 5—figure supplement 1D*) were not efficiently recruited towards cognate tumouroids (*Figure 5A*) or cognate supernatant (*Figure 5—figure supplement 1E*). When wild type (WT) and Ccr5$^{-/-}$ CTLs were co-embedded around a cognate tumouroid, the Ccr5$^{-/-}$ population exhibited severely impaired swarming (*Figure 5B,C*), and infiltration depth (*Figure 5D*). Moreover, Ccr5$^{-/-}$ CTLs exhibited a reduced capacity to eliminate cognate tumour masses (*Figure 5—figure supplement 1F,G*), despite fully functional antigen-specific cytotoxicity (*Figure 5—figure supplement 1H*) and chemokine secretion (*Figure 5—figure supplement 1I*). These findings demonstrate that CCR5 mediates the chemokine signal underpinning the homotypic recruitment of T cells.

In vivo, Ccr5$^{-/-}$ CTLs that were co-transferred with WT CTLs showed impaired homing into CCL3/CCL4-secreting tumours (*Figure 5—figure supplement 2A,B*) and, in contrast to WT CTLs, did not bind fluorescent CCL3 (*Figure 5—figure supplement 2C,D*). We also observed impaired CCR5-dependent recruitment through dense tumour tissue in response to the acute injection of a molecular hydrogel enabling shear-reversible containment of recombinant CCL3 (*Nisbet et al., 2018*; *Figure 5—figure supplement 3*, *Video 7*). To determine if tumour-reactive CTLs can enhance the recruitment of additional CTLs into tumours, we engrafted Rag$^{-/-}$ mice with SSIE-FARL-expressing tumours and delivered a

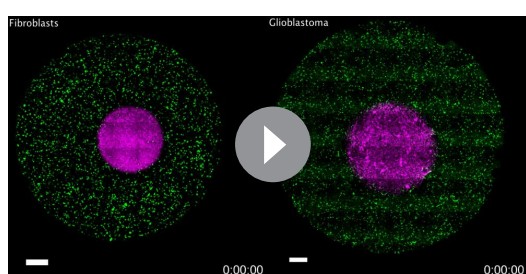

**Video 9.** CAR T cells engaging antigen-expressing glioblastoma tumouroids induce swarming. Long-term (14 hr) time-lapse imaging of human CAR T cells (green) embedded in collagen matrix around tumouroids (magenta). (Left) Tumouroid contains EphA2-specific CAR T cells and EPhA2-negative cells (NIH/3T3). (Right) Tumouroid contains EphA2-specific CAR T cells and glioblastoma cells (WK1). Scale bar, 500 μm. Time in h:min:s.
https://elifesciences.org/articles/56554#video9

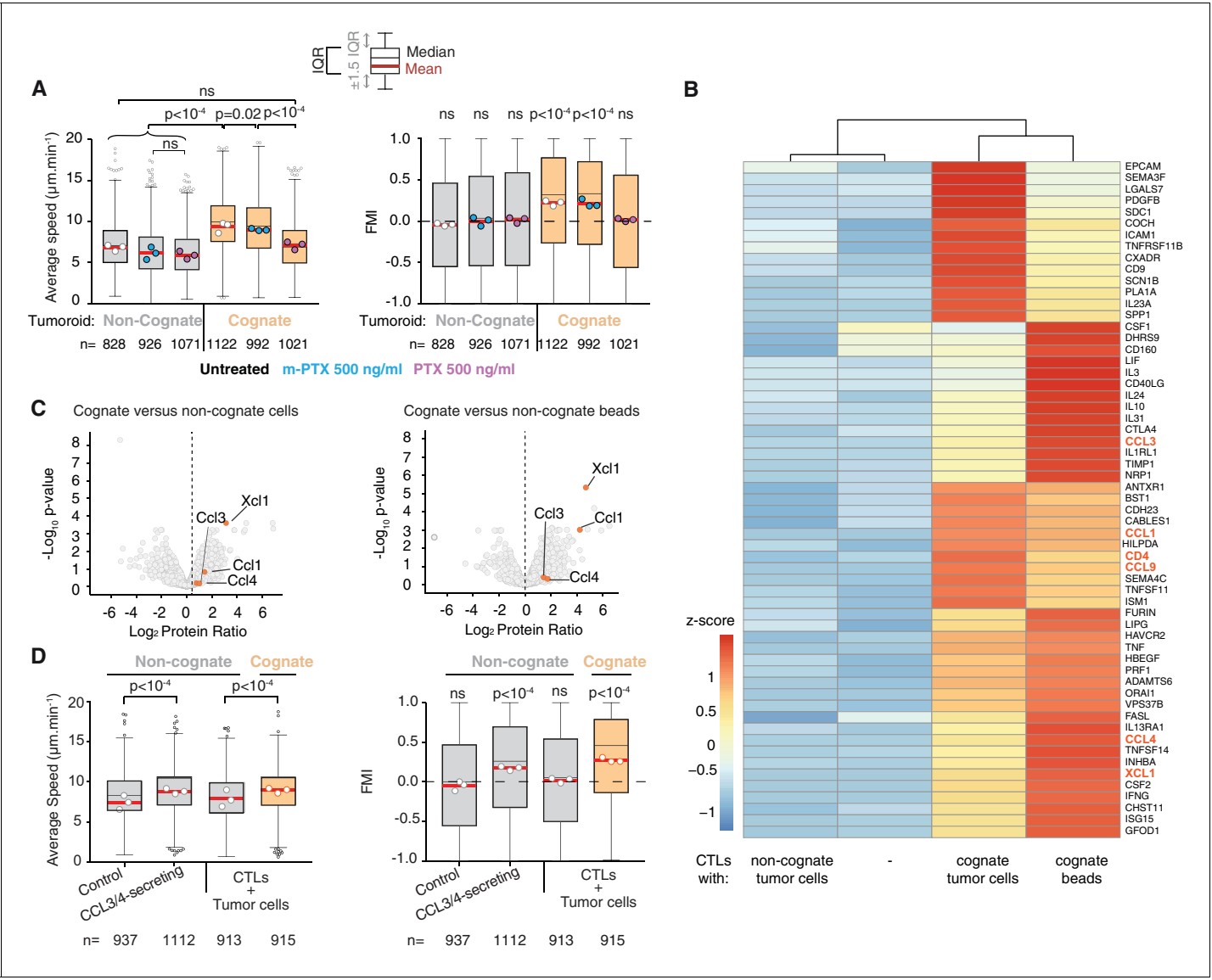

**Figure 4.** CCL3 and CCL4 secretion drives T cell swarming. (**A**) Average speed (Left) and FMI (Right) of OT1 CTLs adjacent to non-cognate EL4 or cognate SIINFEKL-pulsed EL4 tumouroids with pre-embedded tumour-reactive CTLs not treated with pertussis toxin (PTX) or its inactive variant m-PTX. Adjacent CTLs were either untreated, or treated with 500 ng/ml of PTX or m-PTX for 1 hr prior to the migration experiment. (**B**) Hierarchically clustered heat map of differentially expressed genes encoding secreted proteins in resting CTLs (-) or CTLs exposed to non-cognate, cognate tumour cells or cognate beads. GPCR ligands are highlighted. (**C**) Volcano plots of proteins secreted during CTL interactions with tumour cells (left) or beads (right) identified by quantitative mass spectrometry analysis. GPCR ligands identified in (**B**) are highlighted. (**D**) Average speed (Left) and FMI (Right) of CTLs adjacent to tumouroids containing control or CCL3/CCL4-secreting EL4 tumour cells compared with non-cognate and SIINFEKL-pulsed cognate tumouroids pre-embedded with tumour-reactive CTLs. For A and D, box-whiskers indicate medians and the interquartile range (IQR) with outliers outside whiskers. Red bars: mean of pooled data from three independent experiments. Data points: mean of each individual experiment. n: number of tracks. ns: p>0.05, p-values from two-tailed Wilcoxon signed rank test compared to a theoretical median of 0 for FMI. Average speeds were compared using the Kruskal-Wallis test followed by Dunn's multiple comparisons test.

The online version of this article includes the following source data and figure supplement(s) for figure 4:

Source data 1. Source data file for *Figure 4*.

Figure supplement 1. CCL3 and CCL4 are upregulated and secreted by activated CTLs and mediate homotypic recruitment.

Figure supplement 1—source data 1. Source data file for *Figure 4—figure supplement 1*.

Figure supplement 2. CTLs engaging cognate targets generate long-range chemokine gradients.

Figure supplement 3. CCL3/CCL4-secreting tumour cells induce swarming and tumour infiltration.

Figure supplement 3—source data 1. Source data file for *Figure 4—figure supplement 3*.

Figure supplement 4. Leukocyte tumour infiltration in vivo.

*Figure 4 continued on next page*

*Figure 4 continued*

**Figure supplement 4—source data 1.** Source data file for *Figure 4—figure supplement 4*.

primary transfer of tumour-reactive gBT1 CTLs (cognate). 48 hr later, a secondary cohort of WT and Ccr5$^{-/-}$ OT1 CTLs were co-transferred (both non-cognate for the tumours) (*Figure 5E*). The presence of tumour-reactive CTLs (*Figure 5—figure supplement 2E,F*) markedly increased the overall recruitment of all non-cognate CTLs into the tumours (*Figure 5F*). Furthermore, recruitment of Ccr5$^{-/-}$ OT1 CTLs into tumours containing activated gBT1 CTLs was compromised compared to WT CTLs. However, Ccr5$^{-/-}$ CTLs cleared tumours with comparable efficacy to WT CTLs (*Figure 5—figure supplement 2G*). Together, these results indicate that sustained release of CCL3 and CCL4 is sufficient to promote CCR5-dependent homing into tumours in vivo, and that the presence of tumour-reactive CTLs in a tumour promotes the recruitment of distant CCR5$^{+}$ CTLs. Homotypic CTL recruitment via CCR5 is, however, likely not the only or primary signalling circuit that results in CTL tumour infiltration or clearance. It remains to be explored whether adoptive transfers using lower CTL numbers or earlier following tumour engraftment (*Sharma et al., 2013*; *Chheda et al., 2016*) would result in a more dominant role for CCL3 and CCL4-mediated homotypic signalling in recruitment into solid tumours.

## Primary human T cells converge via homotypic recruitment

We next investigated whether swarming also occurs in primary human T cell populations. Human effector T cells activated by anti-CD3/CD28 antibody-coated beads were found to upregulate CCL3 and CCL4 expression (*Figure 6—figure supplement 1A*) as previously reported (*Cristillo et al., 2003*) and produced supernatant that attracts additional effector T cells (*Figure 6A*). When co-embedded with activating beads in place of a tumouroid, human T cells recruit distant T cells, which exhibit increased directionality (*Figure 6B,C*) and swarming (*Figure 6D–F* and *Video 8*). This directional recruitment is abolished by Cenicriviroc (*Figure 6C*).

Finally, we evaluated whether the homotypic recruitment mechanism is inducible via chimeric antigen receptor (CAR) engagement of T cells in an immunotherapeutic context. To this end, we engineered CAR T cells targeting Ephrin type-A receptor 2 (EphA2) that is abundantly overexpressed in glioblastoma (*Liu et al., 2006*). A truncated CD19 domain co-transferred with the CAR construct allows enrichment of genetically-modified CAR-expressing human primary T cells by magnetic sorting (*Figure 6—figure supplement 1B*). Human CD19$^{+}$ EphA2-specific CAR T cells efficiently killed glioblastoma cells (*Figure 6—figure supplement 1C*). When EphA2-specific CAR T cells were embedded in a tumouroid with glioblastoma cells, surrounding CAR T cells swarmed towards the tumouroid, whereas no recruitment was observed when CAR T cells were co-embedded with EphA2-negative fibroblasts in the tumouroid (*Figure 6G–I* and *Video 9*). Collectively, these results demonstrate that upon activation either via the TCR or an engineered CAR, human CTLs engage in homotypic chemokine signalling, thereby amplifying the recruitment of additional T cells to an effector site.

## Discussion

Swarming is a common collective behaviour in the natural world (*Okubo, 1986*). At the immune cell level, swarming has been described for neutrophils, during parasite infection in the lymph node (*Chtanova et al., 2008*) and in response to tissue injury (*Lämmermann et al., 2013*). Here, our data reveal that CTLs engaging targets employ homotypic signalling via secretion of the chemokines CCL3 and CCL4 to accelerate the direct recruitment of distant T cells. As additional antigen-specific CTLs are recruited and engage cognate targets, they contribute to the chemoattractive signal, thus further amplifying CTL recruitment in a positive feedback loop, which leads to swarming behaviour. Interestingly, it appears as though CTLs closest to the source of the chemotactic signal become desensitised over time and lose directional bias, which could be due to exposure to excess ligand concentrations (*Lim et al., 2018*) or rapid internalisation of the CCR5 receptor (*Escola et al., 2010*).

Pioneering work had uncovered that in lymph nodes, naïve CD8$^{+}$ T cells follow CCR5 ligands, including CCL3 and CCL4, to migrate towards dendritic cells interacting with CD4$^{+}$ and CD8$^{+}$ T cells

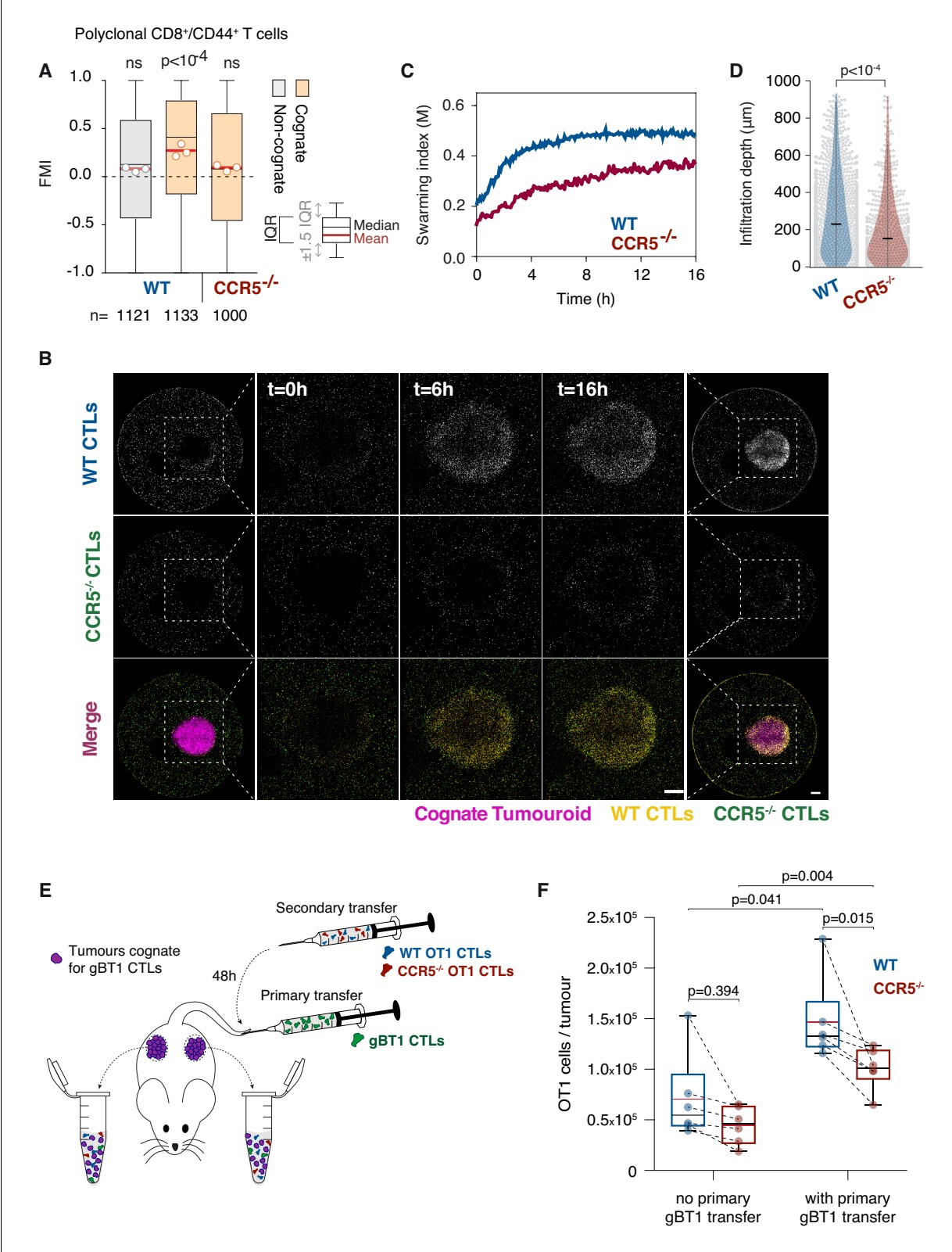

**Figure 5.** T cell swarming is CCR5-dependent. (**A**) FMI of polyclonal WT or Ccr5[-/-] T cells adjacent to non-cognate EL4 or cognate SIINFEKL-pulsed EL4 tumouroids pre-embedded with tumour-reactive CTLs. Box-whiskers indicate medians and the interquartile range (IQR) with outliers outside whiskers. Red bars: mean of pooled data from three independent experiments. Data points: mean of each individual experiment. n: number of tracks for each experimental condition. ns: p>0.05, p-values from Kruskal-Wallis test followed by Dunn's multiple comparison test. (**B**) WT and Ccr5[-/-] OT1 CTLs

*Figure 5 continued on next page*

*Figure 5 continued*

responding to a cognate tumouroid containing tumour-reactive CTLs. Insets highlight WT or Ccr5$^{-/-}$ CTLs infiltrating the cognate tumouroid. Scale bars: 500 μm. (C) Swarming index over time for WT and Ccr5$^{-/-}$ OT1 CTLs from data shown in (B). (D) Infiltration depth of WT and Ccr5$^{-/-}$ OT1 cells into tumouroid from data shown in (B). Violin plots show the distribution of probability densities, black bars represent medians, grey dots are individual data points. (E) *Rag*$^{-/-}$ mice were engrafted subcutaneously with SSIEFARL-expressing EL4 tumours. On day nine post-engraftment, 5 × 10$^6$ gBT1 CTLs were transferred intravenously where indicated. 48 hr after gBT1 transfer, WT and Ccr5$^{-/-}$ OT1 (5 × 10$^6$ each) were co-transferred intravenously. Tumour infiltrates were enumerated by flow cytometry a further 48 hr later. (F) Number of tumour-infiltrating WT and Ccr5$^{-/-}$ OT1 CTLs, with and without prior gBT1 CTL transfer. Box: medians and quartiles; whiskers: range; red bar: mean. Data points from one tumour each, n = 3 mice per group. p-values from Mann-Whitney U tests.

The online version of this article includes the following source data and figure supplement(s) for figure 5:

**Source data 1.** Source data file for *Figure 5*.
**Figure supplement 1.** Chemokine receptor CCR5 mediates homotypic recruitment of CTLs.
**Figure supplement 1—source data 1.** Source data file for *Figure 5—figure supplement 1*.
**Figure supplement 2.** In vivo tumour homing and rejection.
**Figure supplement 2—source data 1.** Source data file for *Figure 5—figure supplement 2*.
**Figure supplement 3.** Intravital imaging following local intratumoural injection of CCL3.

(*Castellino et al., 2006*; *Hickman et al., 2011*; *Hugues et al., 2007*), to optimise the efficiency of T cell priming. Maximal priming and generation of CTLs in infected or draining lymph nodes during antiviral and antitumour responses also depend on CCR5 expression on T cells (*González-Martín et al., 2011*; *Hickman et al., 2011*).

Rather than the behaviour of naïve T cells within secondary lymphoid organs, our study focused on interactions between effector CTLs, T cells that are active in peripheral tissues following priming and emigration from lymph nodes. Our findings reveal that CTLs can independently induce mass recruitment around an antigen-specific target. Our data also reveal that lymphocytes can promote the recruitment of distant lymphocytes into peripheral tissues, thus far thought to only occur indirectly through the action of professional antigen-presenting cell (APC) intermediaries (*Spranger et al., 2017*; *Böttcher et al., 2018*). Although the concept of bacterial quorum sensing has been applied to coordinated population responses by T cells, the amplification of local T cell densities was shown to occur via differentiation or proliferation (*Antonioli et al., 2019*), relatively long-term effects compared to the rapid directional recruitment of distal CTLs discussed here.

CTLs engaging tumour targets that do not present cognate antigen induce chemokinesis in distant CTLs, which could be mediated via the induced secretion of netrins, autotaxin, or semaphorins (*Boneschansker et al., 2016*; *Katakai et al., 2014*; *Takegahara et al., 2005*). Importantly, only CTLs that have directly engaged a cognate antigen-presenting target secrete the homotypic recruitment signal that induces chemotaxis in distant T cells. Unconjugated CTLs exposed to cognate supernatant (containing abundant CCL3, CCL4, and pro-inflammatory cytokines such as interferon gamma and tumour necrosis factor) did not themselves upregulate CCL3 or CCL4 expression (*Supplementary file 1* and *2*). This therefore excludes the possibility of a signal relay, whereby CTLs that sense the recruitment signal away from the target start secreting the homotypic signal themselves, which could presumably lead to aberrant local CTL clustering.

We thus propose that the individuals of a CTL population exist in three distinct states: a 'searcher' state whilst scanning for cognate antigen unaffected by the homotypic signal; a 'responder' state when they sense the homotypic signal and move rapidly towards the effector locus; and an 'engager/recruiter' state upon target cell recognition when they concurrently deliver their effector function and secrete CCL3 and CCL4 (*Figure 7*). The first CTLs to recognise a cognate target directly transition from search into engagement and recruitment, whereas subsequent cells arrive in a responder state. It is interesting to note that CTLs can also remain in an exclusive recruiter state for nearly 24 hr following target elimination in the absence of active engagement (*Figure 4—figure supplement 1H,I*).

This multistate recruitment model explains prior observations of non-specific T cells only deeply infiltrating tumour tissues in the presence of tumour-reactive CTLs (*Boissonnas et al., 2007*). Although antigen recognition is specific, the chemoattractive signal that is raised to amplify recruitment is generic and thus able to recruit a repertoire of CTLs regardless of their TCR specificity. Indeed, there have been prior observations of bystander recruitment of effector T cells to sites of

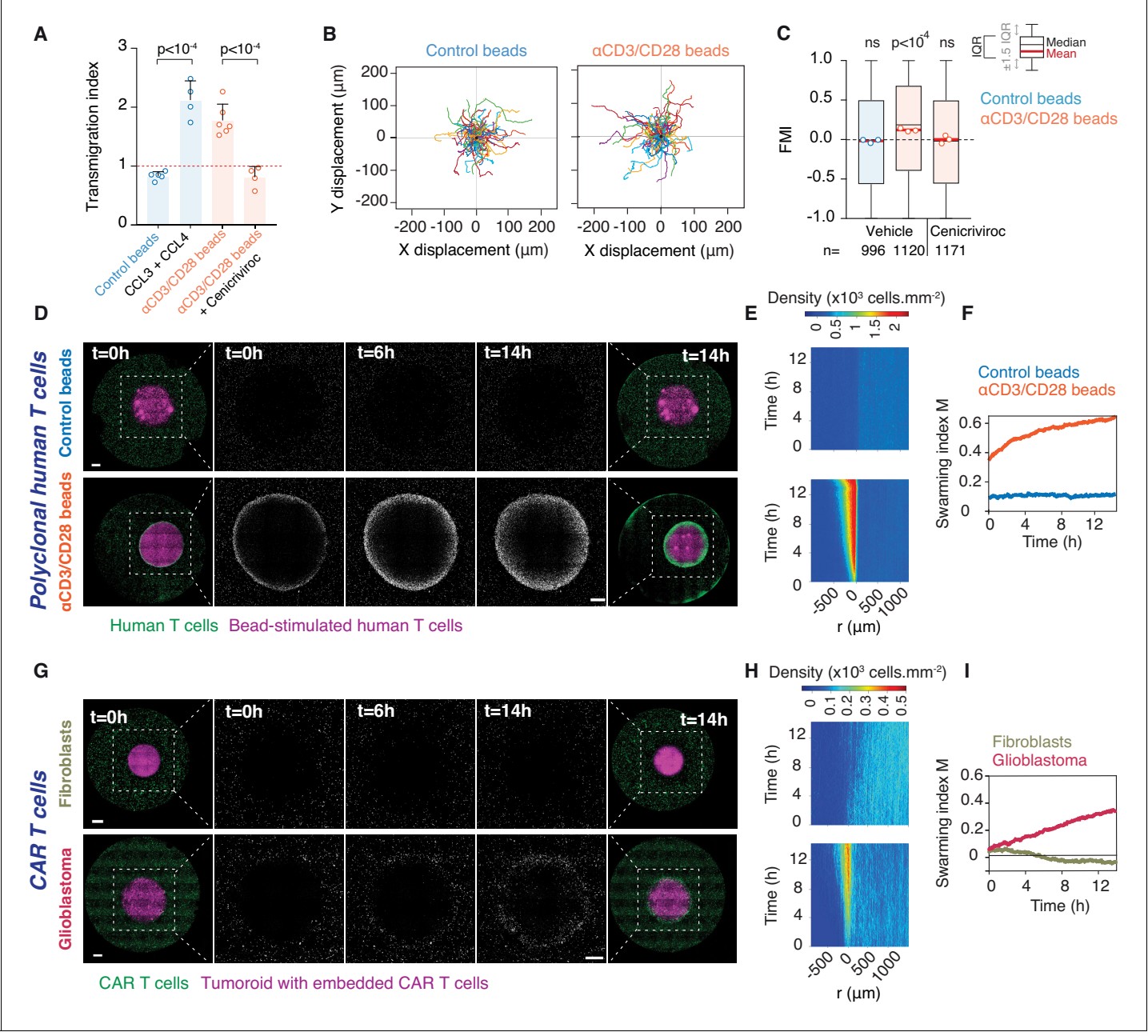

**Figure 6.** Homotypic recruitment of human polyclonal and CAR T cells. (**A**) Transmigration of human T cells towards supernatant from T cells conjugated with IgG-coated beads (control beads), recombinant human CCL3 and CCL4, or supernatant from T cells conjugated with αCD3/CD28-coated beads in the absence or presence of CCR2/CCR5 dual antagonist (Cenicriviroc), relative to transmigration towards basal medium. Bars: mean from four independent experiments (data points). Error bars: SEM. p-values from ANOVA and Tukey's multiple comparison test. (**B**) Cell trajectories of primary polyclonal human T cells adjacent to a tumouroid containing human T cells stimulated with control or αCD3/CD28 beads. Random representative tracks from dataset of three independent experiments, quantified in (**C**). (**C**) FMI of polyclonal human T cells as per (**B**), also assessed in the presence of Cenicriviroc. Box-whiskers indicate medians and the interquartile range (IQR). Red bars: mean of pooled data from three independent experiments. Data points: mean of each individual experiment. n: number of tracks for each experimental condition. ns: p>0.05, p-value from two-tailed Wilcoxon signed rank test compared to hypothetical median of 0. (**D**) Human polyclonal T cells (green) responding to a tumouroid containing T cells (magenta) stimulated with control (top) or αCD3/CD28 beads (bottom). Insets highlight CTLs infiltrating the tumouroid. Scale bars: 500 μm. (**E, F**) Swarming index and density kymographs quantify human T cell movements with respect to tumouroids in (**D**). r: distance from tumouroid edge. (**G**) Human CAR T cells (green) responding to tumouroids containing additional CAR T cells (magenta) and control fibroblast cells (top) or target glioblastoma cells (bottom). Insets highlight CTLs infiltrating the tumouroid. Scale bars: 500 μm. (**H, I**) Swarming index and density kymographs quantify CAR T cell movements with respect to tumouroids in (**G**). r: distance from tumouroid edge.

*Figure 6 continued on next page*

*Figure 6 continued*

The online version of this article includes the following source data and figure supplement(s) for figure 6:

**Source data 1.** Source data file for *Figure 6*.
**Figure supplement 1.** Chemokine expression in activated human polyclonal T cells, and CAR T cell sorting and cytotoxicity.
**Figure supplement 1—source data 1.** Source data file for *Figure 6—figure supplement 1*.

---

inflammation or tumours initiated by unrelated antigen (*Boissonnas et al., 2004*; *Topham et al., 2001*; *Ariotti et al., 2015*; *Hickman et al., 2015*). During an immune response resulting in the clonal expansion of antigen-specific CTLs, which increases their frequency up to $10^4$-fold (*Murali-Krishna et al., 1998*), such a generic recruitment mechanism is likely efficient in their local amplification in sufficient numbers for an effective response.

Interestingly, studies that compared tumours infiltrated by leukocytes ('hot') to those that excluded immune cell infiltration ('cold') identified more abundant CCL3, CCL4, and CCL5 in hot tumours (*Chakravarthy et al., 2018*; *Spranger et al., 2015*), including via single-cell transcriptomics that ascribed their expression to CD8[+] T cells (*Jerby-Arnon et al., 2018*; *Roider et al., 2020*). Whilst we have identified CCL3 and CCL4 as the key mediators of homotypic signalling in CTLs isolated from mice, CCL5 may also have a central role in homotypic recruitment of human CTLs.

We demonstrated that CTLs exhibit chemotaxis towards CCR5 ligands within dense tumour tissue by intravital imaging, and revealed that the presence of antigen-specific CTLs within a tumour promotes the recruitment of distant CTLs into the tumour in a manner partially dependent on CCR5. The activation of intratumoural CTLs were found by others to promote the recruitment of not only CTLs, but also natural killer cells and myeloid cells (*Rosato et al., 2019*). Alternate chemokine signalling circuits centred on CXCR3 and BLT1 have been implicated in CTL tumour trafficking and infiltration (*Chheda et al., 2016*; *Chow et al., 2019*; *Mikucki et al., 2015*; *Sharma et al., 2013*). Furthermore, IFNγ, produced by activated CTLs and abundant in inflamed tumours, can induce CXCL9 and CXCL10 secretion in various intermediary cells, including tumour-infiltrating myeloid cells (*Dobrzanski et al., 2001*; *Gordon-Alonso et al., 2017*; *Hickman et al., 2015*), which may enhance the role of CXCR3 in homing.

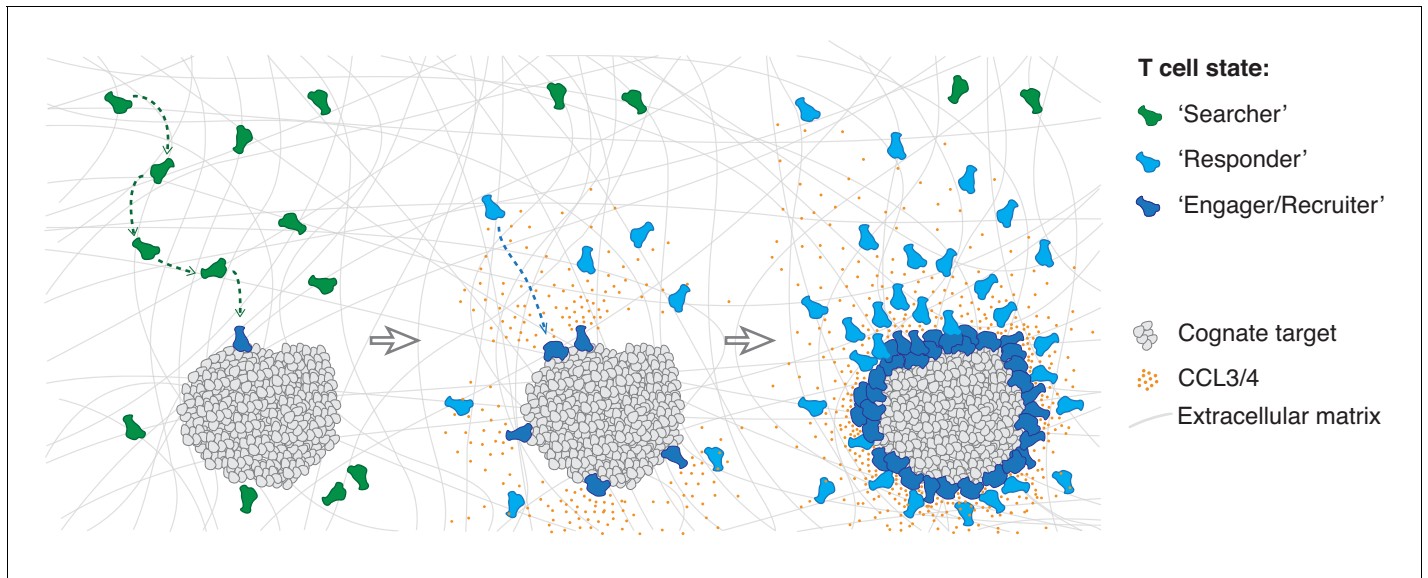

**Figure 7.** Model of multistate mass recruitment of CTLs to a target site. 'Searcher' CTLs migrate randomly in search of target cells. When they recognise targets, they become 'engager/recruiter' CTLs that secrete CCL3 and CCL4. CCR5-expressing CTLs within range migrate rapidly and directionally towards the effector site in a 'responder' state. Newly arriving CTLs engage and convert to recruiters: they contribute to the chemoattractive signal, thus further amplifying recruitment of other 'responder' CTLs from distal regions, resulting in sustained accumulation of large numbers of CTLs at the target site (swarming).

CCR5-expressing CTLs have been associated with severe autoimmune diseases (*Mackay, 2014*), and CCR5 ligands are abundant in the upper respiratory tract during acute hyperinflammation associated with viral infection (*Chua et al., 2020*); silencing homotypic recruitment may well prove beneficial in such contexts. Ultimately, the homotypic recruitment mechanism could be exploited to enhance control over positioning and tissue infiltration of T cells in adoptive cell transfer immunotherapies, not least given our findings that engineered CAR T cells, following antigen recognition via the CAR, are able to induce homotypic recruitment of distal T cell populations.

# Materials and methods

**Key resources table**

| Reagent type (species) or resource | Designation | Source or reference | Identifiers | Additional information |
|---|---|---|---|---|
| Strain, strain background (*M. musculus*) | Rag-deficient B6.129S7-Rag < tm1mom > JAusb (RAG1) | Australian BioResources | | Used for in vivo experiments |
| Strain, strain background (*M. musculus*) | B6.SJL-PtprcaPepcb/BoyJAusb (PTPRCA) | Australian BioResources | | Used for in vivo experiments |
| Strain, strain background (*M. musculus*) | B6.129p2-ccr5 tm1Kuz/J (CCR5$^{-/-}$) | Australian BioResources | | 7–12 week mice used to isolate CCR5KO CTLs |
| Strain, strain background (*M. musculus*) | B6-Tg-OTI (OT1) | Australian BioResources | | 7–12 week mice used to isolate OT1 CTLs |
| Strain, strain background (*M. musculus*) | B6-Tg(CAG-Lifeact/EGFP)-OTI (OT1$^{GFP}$) | *Galeano Niño et al., 2020* | | 7–12 week mice used to isolate LifeactGFP OT1 CTLs |
| Strain, strain background (*M. musculus*) | B6.gBT-I/uGFP (gBT1-GFP) | Gift from W. R. Heath | | 7–12 week mice used to isolate GFP gBT1 CTLs |
| Strain, strain background (*M. musculus*) | B6.gBT-I/dsRED (gBT1-dsRED) | Gift from W. R. Heath | | 7–12 week mice used to isolate dsRED gBT1 CTLs |
| Cell line (*M. musculus*) | EL-4 lymphoma | ATCC | Cat. #: TIB-39 | |
| Cell line (*M. musculus*) | NIH/3T3 fibroblast | ATCC | Cat. #: CRL-1658 | |
| Cell line (*Homo-sapiens*) | WK1 glioblastoma cells | *Day et al., 2013* | | Cultured as described in *Stringer et al., 2019* |
| Cell line (*Homo-sapiens*) | GP2-293 embryonic kidney cells | Clonetech | Cat. #: 631458 | |
| Cell line (*M. musculus*) | mCCL3-mScarletI secreting EL4 cells | This paper | | EL4 cells transduced with CCL3-mScarletI construct. Sequence available in Materials and methods. |
| Cell line (*M. musculus*) | mCCL4-miRFP670 secreting EL4 cells | This paper | | EL4 cells transduced with CCL4-miRFP670 construct. Sequence available in Materials and methods. |
| Cell line (*M. musculus*) | 'CCL3/4-secreting EL4' i.e. mCCL3-mScarletI and mCCL4-miRFP670 secreting EL4 cells | This paper | | EL4 cells transduced with CCL3-mScarletI and CCL4-miRFP670 construct. Sequences available in Materials and methods. |
| Cell line (*M. musculus*) | EL4.HSV | This paper | | EL4 cells transduced with bicistronic SSIEFARL epitope and mTagBFP2. Sequence available in Materials and methods. |
| Cell line (*M. musculus*) | EL4.OVA | This paper | | EL4 cells transduced with bicistronic SIINFEKL epitope and mTagBFP2. Sequence available in Materials and methods. |

*Continued on next page*

Continued

| Reagent type (species) or resource | Designation | Source or reference | Identifiers | Additional information |
|---|---|---|---|---|
| Cell line (*Homo-sapiens*) | CAR T cells | This paper | | Primary human T cells transduced with EphA2-specific CAR construct. Produced by Kramer lab at Children's Cancer Research Unit, The Children's Hospital at Westmead. Information on CAR constructs available in Materials and methods. |
| Transfected construct (*M. musculus*) | MSCV-based retroviral expression vector (LENC) | *Fellmann et al., 2013* Gift from J. Zuber | http://doi.org/10.1016/j.celrep.2013.11.020 | Retrovirus construct to express cDNA |
| Transfected construct (*M. musculus*) | pMD2.G | Gift from Didier Trono | Addgene plasmid # 12259; RRID:Addgene_12259 | Construct for VSVG pseudotyping retrovirus |
| Transfected construct (*M. musculus*) | mCCL3-mScarletI gBlocks | Integrated DNA Technologies | | gBlocks cloned into LENC |
| Transfected construct (*M. musculus*) | mCCL4-miRFP670 gBlocks | Integrated DNA Technologies | | gBlocks cloned into LENC |
| Transfected construct (*M. musculus*) | SSIEFARL-mTagBFP2 gBlocks | Integrated DNA Technologies | | gBlocks cloned into LENC |
| Transfected construct (*M. musculus*) | SIINFEKL-mTagBFP2 gBlocks | Integrated DNA Technologies | | gBlocks cloned into LENC |
| Transfected construct (*Homo- sapiens*) | EphA2-specific CAR construct | *Chow et al., 2013* and *Yi et al., 2018* | | Construct for lentiviral transduction of human T cells |
| Biological sample (*Homo-sapiens*) | Peripheral blood mononuclear cells (PBMCs) | Sydney Children's Hospitals Network | | Used to isolate human T cells |
| Antibody | Anti-XCL1 (polyclonal goat IgG) | R & D Systems | Cat. #: AF486 RRID:AB_2216915 | (5 µg/ml) |
| Antibody | Anti-CCL1 (monoclonal rat IgG2a) | R & D Systems | Cat. #: MAB845 RRID:AB_2070618 | (5 µg/ml) |
| Antibody | Anti-CXCR2 (monoclonal rat IgG2a) | R & D Systems | Cat. #: MAB2164-100 RRID:AB_358062 | (10 µg/ml) |
| Antibody | Anti-CCL3 (polyclonal goat IgG) | R & D Systems | Cat. #: AF-450-NA RRID:AB_354492 | (5 µg/ml) |
| Antibody | Anti-CCL4 (monoclonal rat IgG2a) | R & D Systems | Cat. #: MAB451-100 RRID:AB_2259676 | (5 µg/ml) |
| Antibody | Anti-CXCR4 ( monoclonal rat IgG2b) | R & D Systems | Cat. #: MAB21651-100 RRID:AB_2801441 | (10 µg/ml) |
| Antibody | Anti-CCL5 (polyclonal goat IgG) | R & D Systems | Cat. #: AF478 RRID:AB_10080077 | (1 µg/ml) |
| Antibody | Anti-CCL9/10 (monoclonal rat IgG1) | R & D Systems | Cat. #: MAB463 RRID:AB_2259783 | (5 µg/ml) |
| Antibody | Anti-CXCL10 (monoclonal rat IgG2a) | R & D Systems | Cat. #: MAB466 RRID:AB_2292486 | (1 µg/ml) |
| Antibody | Anti-CXC12 (monoclonal mouse IgG1) | R & D Systems | Cat. #: MAB310 RRID:AB_2276927 | (1 µg/ml) |
| Antibody | Anti-CD3ε (monoclonal Armenian Hamster) | BioLegend | Cat. #: 100302 RRID:AB_312667 | (1 µg/ml) |
| Antibody | Anti-CD28 (monoclonal Syrian Hamster) | BioLegend | Cat. #: 102101 RRID:AB_312866 | (1 µg/ml) |
| Antibody | Rat IgG1, κ Isotype control (Rat monoclonal) | BioLegend | Cat. #: 400402 RRID:AB_326508 | (1 µg/ml) |

*Continued on next page*

Continued

| Reagent type (species) or resource | Designation | Source or reference | Identifiers | Additional information |
|---|---|---|---|---|
| Antibody | Mouse IgG1, κ Isotype control (Mouse monoclonal) | BioLegend | Cat. #: 401401 RRID:AB_2801452 | (1 µg/ml) |
| Antibody | APC anti-CD44 (monoclonal rat IgG2b) | BD Biosciences | Cat. #: 553991 RRID:AB_10050405 | (1 µg/ml) |
| Antibody | Pacific Blue anti-CD8a (monoclonal rat IgG2a) | BD Biosciences | Cat. #: 558106 RRID:AB_397029 | (1 µg/ml) |
| Antibody | Biotin anti-CD3ε (monoclonal Armenian Hamster) | Thermo Fisher Scientific | Cat. #: 13-0031-82 RRID:AB_466319 | (1 µg/ml) |
| Antibody | Biotin anti-CD28 (monoclonal Syrian Hamster) | Thermo Fisher Scientific | Cat. #: 13-0281-82 RRID:AB_466411 | (1 µg/ml) |
| Antibody | Biotin anti-CD3 (monoclonal mouse IgG2a) | Thermo Fisher Scientific | Cat. #: 13-0037-82 RRID:AB_1234955 | (1 µg/ml) |
| Antibody | Biotin anti-CD28 (monoclonal mouse IgG1) | Thermo Fisher Scientific | Cat. #: 13-0289-82 RRID:AB_466415 | (1 µg/ml) |
| Antibody | Biotin rat IgG2a κ Isotype Control (Rat polyclonal) | eBioscience | Cat. #: 13-4321-82 RRID:AB_470084 | (1 µg/ml) |
| Antibody | Biotin IgG Isotype Control (Armenian hamster) | eBioscience | Cat. #: 14-4888-81 RRID:AB_470128 | (1 µg/ml) |
| Antibody | Brilliant Violet 711 anti-CD11b (monoclonal rat IgG2b) | BioLegend | Cat. #: 101241 RRID:AB_11218791 | (2 µg/ml) |
| Antibody | Alexa Fluor 488 anti-NK1.1 (monoclonal mouse IgG2a) | BioLegend | Cat. #: 108717 RRID:AB_493184 | (2 µg/ml) |
| Antibody | FITC anti-Ly-6C (monoclonal rat IgG2c) | BioLegend | Cat. #: 128005 RRID:AB_1186134 | (2 µg/ml) |
| Antibody | Brilliant Violet 510 anti-CD90.2 (monoclonal rat IgG2b) | BioLegend | Cat. #: 105335 RRID:AB_2566587 | (2 µg/ml) |
| Antibody | Brilliant Violet 421 anti-CD64 (monoclonal mouse IgG1) | BioLegend | Cat. #: 139309 RRID:AB_AB_2562694 | (2 µg/ml) |
| Antibody | APC/Fire 750 anti-CD45 (monoclonal rat IgG2b) | BioLegend | Cat. #: 103153 RRID:AB_2572115 | (2 µg/ml) |
| Antibody | APC/Fire 750 anti-CD45.1 (monoclonal mouse IgG2a) | BioLegend | Cat. #: 110751 RRID:AB_2629805 | (2 µg/ml) |
| Antibody | FITC anti-CD11c (monoclonal Armenian Hamster) | BioLegend | Cat. #: 117305 RRID:AB_313774 | (2 µg/ml) |
| Antibody | Pacific blue anti-MHC class II (monoclonal rat IgG2b) | BioLegend | Cat. #: 107619 RRID:AB_493528 | (2 µg/ml) |
| Antibody | APC anti-CD8a (monoclonal rat IgG2a) | BioLegend | Cat. #: 100711 RRID:AB_312750 | (2 µg/ml) |
| Antibody | Biotin anti-CCL3 (polyclonal Rabbit sera) | PeproTech | Cat. #: 500-P121BT | (1 µg/ml) |
| Peptide, recombinant protein | OVA$_{257-264}$ peptide, SIINFEKL | AusPep | Cat. #: 2711 | (1, 0.2, 0.1, 0.02, 0.001 µg/ml) |
| Peptide, recombinant protein | gB$_{498-505}$ peptide, SSIEFARL | Auspep | Cat. #: gb498-505 | (1 µg/ml) |
| Peptide, recombinant protein | Recombinant mouse IL-2 | BioLegend | Cat. #: 575408 | (100 ng/ml) |
| Peptide, recombinant protein | Recombinant human IL-7 | Miltenyi Biotech Australia | Cat. #: 130-095-362 | (10 ng/ml) |
| Peptide, recombinant protein | Recombinant human IL-15 | Miltenyi Biotech Australia | Cat. #: 130-095-764 | (5 ng/ml) |

*Continued*

| Reagent type (species) or resource | Designation | Source or reference | Identifiers | Additional information |
|---|---|---|---|---|
| Peptide, recombinant protein | Corning Collagen I Rat Tail Natural | In Vitro Technologies | Cat. #: FAL354236 | (~1.5 mg/ml) |
| Peptide, recombinant protein | Monobiotinylated H-2K$^b$/SIINFEKL | Tetramer Synthesis Service | John Curtin School of Medical Research | (1 µg/ml) |
| Peptide, recombinant protein | Recombinant human (cross reactive with murine) CCL1 | Peprotech | Cat. #: 300–37 | (1, 10 and 100 ng/ml) |
| Peptide, recombinant protein | Recombinant Mouse XCL1 | R & D systems | Cat. #: 486-LT-025 | (1, 10 and 100 ng/ml) |
| Peptide, recombinant protein | Recombinant Murine CCL3 | Peprotech | Cat. #: 250–09 | (1, 10 and 100 ng/ml) |
| Peptide, recombinant protein | Recombinant Murine CCL4 | Peprotech | Cat. #: 250–32 | (1, 10 and 100 ng/ml) |
| Peptide, recombinant protein | Recombinant Murine CCL5 | Peprotech | Cat. #: 250–07 | (1, 10 and 100 ng/ml) |
| Peptide, recombinant protein | Recombinant murine CCL9/10 | Peprotech | Cat. #: 250–12 | (1, 10 and 100 ng/ml) |
| Peptide, recombinant protein | Recombinant murine CXCL10 | Peprotech | Cat. #: 250–16 | (1, 10 and 100 ng/ml) |
| Peptide, recombinant protein | Recombinant Human CXCL12 | Peprotech | Cat. #: 300-28A | (1, 10 and 100 ng/ml) |
| Peptide, recombinant protein | Lysyl Endopeptidase, Mass Spectrometry Grade (Lys-C) | FUJIFILM Wako Pure Chemical Corp | Cat. #: 121–05063 | (1 µg) |
| Sequence-based reagent | TruSeq Stranded mRNA | Illumina | Cat. #: 20020594 | |
| Chemical compound, drug | CellTracker Orange 5-(and-6)-(((4-chloromethyl) benzoyl) amino) tetramethylrhodamine (CMTMR) | Invitrogen | Cat. #: C2927 | (5 µM) |
| Chemical compound, drug | CellTracker Deep Red | Invitrogen | Cat. #: C34565 | (1 µM) |
| Chemical compound, drug | CellTracker Green 5-chloromethylfluorescein diacetate (CMFDA) | Invitrogen | Cat. #: C7025 | (5 µM) |
| Chemical compound, drug | Ketamine | Provet Pty Ltd | Cat. #: KETAM I | (100 mg/kg) |
| Chemical compound, drug | Xylazine | Provet Pty Ltd | Cat. #: XYLA Z2 | (15 mg/kg) |
| Chemical compound, drug | Pertussis toxin from Bordetella pertussis (PTX) | Sigma Aldrich | Cat. #: P7208 | (100 ng/ml) |
| Chemical compound, drug | Inactive mutated version of Pertussis toxin from Bordetella pertussis (m-PTX) | Sigma Aldrich | Cat. #: PT-16.0003 | (100 ng/ml) |
| Chemical compound, drug | Maraviroc (CCR5 antagonist) | Sigma Aldrich | Cat. #: PZ0002-5mg | (10 µg/ml) |
| Chemical compound, drug | Cenicriviroc (CCR2 + CCR5 dual inhibitor) | AdooQ Bioscience | Cat. #: A13632 -10 (000–22038) | (1 µM) |
| Chemical compound, drug | CCR2 antagonist | Santa Cruz Biotechnology Inc | Cat. #: sc-202525 | (1 µM) |
| Chemical compound, drug | UCB35625 (CCR1 + CCR3 dual inhibitor) | R & D Systems | Cat. #: 2757/1 | (50 µM) |
| Chemical compound, drug | Iodoacetamide (IAM) | Sigma Aldrich | Cat. #: I6125-5G | (1 M) |
| Chemical compound, drug | Dithiothreitol (DTT) | Sigma Aldrich | Cat. #: 43816–10 ML | (50 mM) |

*Continued on next page*

*Continued*

| Reagent type (species) or resource | Designation | Source or reference | Identifiers | Additional information |
|---|---|---|---|---|
| Commercial assay or kit | EasySep Direct Human T Cell Isolation Kit | Stem Cell Technologies | Cat. #: 19661 | |
| Commercial assay or kit | T Cell TransAct, human | Miltenyi Biotech Australia | Cat. #: 130-111-160 | |
| Commercial assay or kit | EasySep Human CD19 Positive Selection Kit II | Stem Cell Technologies | Cat. #: 17854 | |
| Commercial assay or kit | RNeasy Mini Kit | Qiagen | Cat. #: 74104 | |
| Commercial assay or kit | High-Capacity cDNA Reverse Transcription Kit | ThermoFisher Scientific | Cat. #: 4368814 | |
| Commercial assay or kit | Sandwich ELISA kits (CCL1) | OriGene Technologies | Cat. #: EA100390 | |
| Commercial assay or kit | Sandwich ELISA kits (CCL9) | OriGene Technologies | Cat. #: EA100725 | |
| Commercial assay or kit | Cytometric Bead Array: Multi-Analyte Flow Assay Kit (CCL3, CCL4, CCL5, CXCL10, TNF-$\alpha$ and IFN-$\gamma$) | LEGENDplex | Custom mouse panel | |
| Commercial assay or kit | BD Cytometry Beads array Mouse MIP-1b Flex Set (Bead C9) | BD Biosciences | Cat. #: 558343 | |
| Commercial assay or kit | BD Cytometry Beads array Mouse MIP-1a Flex Set (Bead C7) | BD Biosciences | Cat. #: 558449 | |
| Commercial assay or kit | CellTrace CFSE Proliferation Kit | ThermoFisher Scientific | Cat. #: C34554 | |
| Commercial assay or kit | CellTrace Violet Proliferation Kit | ThermoFisher Scientific | Cat. #: C34571 | |
| Commercial assay or kit | Cytofix/Cytoperm Plus Kit (with BD GolgiStop) | BD Bioscience | Cat. #: 554715 | |
| Software, algorithm | FlowJo software version 10.2 | Tree Star Inc | | flowjo.com |
| Software, algorithm | Prism GraphPad Software version 8 | GraphPad | | Imaris.oxinst.com |
| Software, algorithm | Leica Application Suite (LAS) X software version 3.3.0.16794 | Leica Microsystems | | Leica-microsystems.com |
| Software, algorithm | FLUOstaromega software version 5.10 R2 | BMG Labtech GmbH | | Bmglabtech.com |
| Software, algorithm | CFX Manager Software | Bio-Rad | | Bio-rad.com |
| Software, algorithm | BioStation IM software | Nikon | | nikon.com |
| Software, algorithm | ImageJ/FIJI | US National Institutes of Health | | Imagej.net |
| Software, algorithm | Matlab | Mathworks | | Mathworks.com |
| Software, algorithm | R | R Core Team | | r-project.org |
| Software, algorithm | Plot2 for Mac 2.6.1 | Mike Wesemann | | apps.micw.org |
| Software, algorithm | motilisim | *Read et al., 2016* | | https://github.com/marknormanread/TcellSwarming |
| Software, algorithm | FastQC (version 0.11.5) | Babraham Institute | | bioinformatics.babraham.ac.uk |
| Software, algorithm | Trimmomatic (version 0.36) | Bolger, A. M., Lohse, M., and Usadel, B. | | usadellab.org |
| Software, algorithm | FeatureCounts (version 1.5.1) | Bioconductor | | bioconductor.org |
| Software, algorithm | STRING DB | ELIXIR | | string-db.org |
| Software, algorithm | MaxQuant (version 1.5.8.3) | Max Plank Insitute of Biochemistry | | maxquant.org |

*Continued on next page*

*Continued*

| Reagent type (species) or resource | Designation | Source or reference | Identifiers | Additional information |
|---|---|---|---|---|
| Software, algorithm | LFQ Analyst | Monash University | | bioinformatics.erc.monash.edu/apps/LFQ-Analyst |
| Software, algorithm | Limma | Bioconductor | | bioconductor.org |
| Software, algorithm | Motility_analysis package | Python Software Foundation | | https://github.com/marknormanread/TcellSwarming |
| Other | Streptavidin-coated polystyrene particles | Spherotech Inc | Cat. #: SVP-60–5 | 6.0–8.06 µm diameter |
| Other | SPHERO AccuCount blank particles | Spherotech Inc | Cat. #: QACBP-70–10 | |
| Other | Sera-Mag Speed beads | ThermoFisher Scientific | Cat. #: 65152105050250 | |
| Other | Sera-Mag Speed beads | ThermoFisher Scientific | Cat. #: 45152105050250 | |
| Other | Trypsin-gold | Promega | Cat. #: V5280 | |
| Other | Dimethyl sulfoxide (DMSO) | Sigma-Aldrich | Cat. #: D2650-100ml | |
| Other | Collagenase IV | Sigma-Aldrich | Cat. #: C5138-1G | |
| Other | 4′, 6-Diamidino-2-phenylindole (DAPI) | ThermoFisher Scientific | Cat. #: D3571 | (0.5 µg/ml) |
| Other | 10 kDA Dextran labelled with AlexaFluor647 (Dextran-AF647) | ThermoFisher Scientific | Cat. #: D22914 | (10 µg/ml) |
| Other | Fmoc-Asp-Wand resin | GL Biochem | Cat. #: 40501 | |
| Other | 1-hydroxybenzotriazole (HOBt) | GL Biochem | Cat. #: 2592-95-2 | |
| Other | 2-(1H-Benzotriazole-1-yl)—1,1,3,3-Tetramethyluronium hexafluorophosphate (HBTU) | GL Biochem | Cat. #: 94790-37-1 | |

## Mice, primary T cell isolation and cell culture

All mice were maintained at Australian BioResources (Moss Vale, NSW, Australia) and in the animal facility of the Lowy Cancer Research Centre, University of New South Wales. All animal breeding and experimentation were conducted in accordance with New South Wales state and Australian federal laws and animal ethics protocols overseen and approved by the University of New South Wales Animal Care and Ethics Committee (16/83B and 19/133B).

T cells were obtained from spleens of 7- to 12-week-old mice, either OT1 (specific for the $OVA_{257-264}$ peptide SIINFEKL in a H-2K$^b$ major histocompatibility complex class I context), hereafter referred to as OT1 CTLs, or OT1 ×Lifeact EGFP *Galeano Niño et al., 2020* referred to as OT1$^{GFP}$ CTLs. Mice were euthanised by cervical dislocation. Splenocytes were stimulated ex vivo with $OVA_{257-264}$ (SIINFEKL) peptide (AusPep, Melbourne, VIC, Australia) for 4 hr in T cell medium (TCM) consisting of Roswell Park Memorial Institute (RPMI) 1640, 10% foetal calf serum, 1 mM sodium pyruvate, 10 mM HEPES, 100 U/ml penicillin, 100 µg/ml streptomycin, 2 mM L-glutamine and 50 µM β2-mercaptoethanol (all from Gibco, ThermoFisher Scientific, Waltham, MA, USA). 100 ng/ml recombinant mouse IL-2 (R & D Systems, Minneapolis, MN, USA) were added on the following day and subsequently every 2 days. T cells were cryopreserved at day 3 after isolation according to established methods (*Galeano Niño et al., 2016*). Cryopreserved T cells were recovered by quick thawing in a 37°C water bath, the thawed cryopreservation solution was diluted in 10 ml TCM and the cells were then pelleted by centrifugation and resuspended in TCM supplemented with 100 ng/ml recombinant mouse IL-2, renewed every 2 days until use. All experiments were performed using T cells cultured to days 6 or 7 post-isolation.

gBT1 TCR transgenic mice (*Coles et al., 2003*) that had been crossed to UBI-GFP mice (expressing GFP under control of the human Ubiquitin C promoter) (gBT1-GFP) were a kind gift from W. R. Heath. The gBT1 TCR is specific for HSV glycoprotein B$_{498-505}$ peptide SSIEFARL in a H-2K$^b$ context.

SSIEFARL peptide was purchased from AusPep. gBT1 CTLs were generated following the same procedure as OT1 CTLs.

To generate polyclonal CD8$^+$ CD44$^+$ effector T cells (from wild-type (WT) or Ccr5$^{-/-}$ C57BL/6 mice, the latter kindly provided by G. Clarke), single splenocyte suspensions were stimulated ex vivo for 24 hr with 1 µg/ml anti-CD3 (clone 145–2 C11; BioLegend, San Diego, CA, USA), 1 µg/ml anti-CD28 (clone 37.51; BioLegend) and 100 ng/ml IL-2 in 10 ml of TCM. Cells were then washed and sorted on the FACS-Aria III flow sorter (BD Biosciences; Franklin Lakes, NJ, USA) based on CD8 and CD44 surface expression detected by antibody staining with 1 µg/ml anti-CD8a-Pacific Blue (clone 53–6.7; BD Biosciences) and anti-CD44-APC (clone IM7; BD Biosciences). After sorting, cells were expanded in TCM and IL-2 until day 6. In some experiments, cells were frozen on day 3 and thawed for expansion until day 6 or 7 before experimental use. Ccr5$^{-/-}$ C57BL/6 mice were crossed with OT1 transgenic mice to obtain Ccr5$^{-/-}$ OT1 CTLs.

The murine EL4 lymphoma cell line, originally obtained from the American Type Culture Collection (ATCC; TIB-39), was obtained at early passage (P3) from the Alexander lab. EL4 were cultured in TCM, with routine passaging three times per week, maintained at cell densities under $1 \times 10^6$/ml. For preparation of cognate cells for OT1 or gBT1 CTLs, EL4 cells were pulsed for 16 hr with 1 µg/ml SIINFEKL (SFKL) or SSIEFARL (SFARL) peptide, respectively (referred to as 'cognate tumour cells'). The cell line was further authenticated by engrafting into C57BL/6 immunocompetent mice, as well as verifying sensitivity to H-2K$^b$-specific cytotoxicity. Cells were free of mycoplasma contamination based on the MycoAlert Mycoplasma Detection Kit (Lonza, Basel, Switzerland) detected on a FLUOstar Omega microplate reader (BMG Labtech GmbH, Ortenberg, Germany).

The patient-derived glioblastoma cell line WK1 (*Day et al., 2013*) was cultured as previously described (*Stringer et al., 2019*). NIH/3T3 mouse fibroblast cells were cultured in Dulbecco's Modified Eagle's Medium (DMEM) supplemented with 10% feotal calf serum.

For in vivo experiments, we used either C57BL/6 mice, Rag-deficient B6.129S7-Rag < tm1mom > JAusb mice ('RAG1', Australian BioResources, NSW, Australia), hereafter referred to as Rag$^{-/-}$, or B6. SJL-PtprcaPepcb/BoyJAusb mice, referred to as PTPRCA mice.

## Isolation of primary human T cells

Human peripheral blood mononuclear cells (PBMCs) were obtained from healthy donors after informed consent and were used in experiments under a Human Research Ethics Committee (HREC) approved protocol (Sydney Children's Hospitals Network, LNR/13/SCHN/241). We isolated PBMCs by separation over Lymphoprep, and then enriched for T cell populations using a human T cell isolation kit according to the manufacturer's instructions (Stem Cell technologies, Tullamarine, Australia). T cells were activated with T Cell TransAct (Miltenyi Biotec Australia, Sydney, Australia) in CTS OpTmizer T cell expansion medium supplemented with CTS Immune cell SR (2.5%, ThermoFisher Scientific Australia) and recombinant human IL-7 and IL-15 (10 ng/mL and 5 ng/mL respectively, Miltenyi Biotech Australia). Human T cells were subsequently expanded in the above medium supplemented with IL-7 and IL-15 every 48 hr for ~12 days prior to use in experiments.

## Generating chimeric antigen receptor (CAR) T cells

Human T cells were isolated and activated as indicated above and transduced with a pRRL lentivirus (*Dull et al., 1998*) encoding an EphA2-specific CAR consisting of the EphA2-specific 4H5 scFv (*Chow et al., 2013*) and a 41BB.CD3ζ endodomain, linked by a 2A sequence with a truncated form of CD19 (*Yi et al., 2018*). The transduced T cells were assessed for CD19 expression at day 4 post-transduction and further expanded for another 6 days. CAR-expressing T cells were enriched using an EasySep release human CD19 positive selection kit (StemCell Technologies, Vancouver, Canada). Purity of the enriched CAR T cell population was validated by flow cytometry (see *Figure 6—figure supplement 1B*).

## 3D cell migration assays

CTLs at day 6 or 7 post-isolation were labelled with 5 µM CellTracker Orange 5-(and-6)-(((4-chloromethyl) benzoyl) amino) tetramethylrhodamine (CMTMR) (Invitrogen, California, USA) and EL4 tumour cells were labelled with CellTracker Deep Red dye (ThermoFisher Scientific) either pulsed with cognate peptide (cognate) or unpulsed (non-cognate). A total of 4–6 $\times 10^6$ cells (either tumour

cells alone or mixed 1:1 with labelled CTLs) suspended in phenol red-free TCM (20 µl) were added to ice cold liquid-phase rat-tail collagen I (25 µl at ~3 mg/ml; Corning, New York, NY, USA) containing 1 N NaOH (0.77 µl) and 10 × PBS (5 µl) for a total volume of 50 µl on ice. Where bead-containing side tumoroids were constructed, 20 µl of a collagen preparation containing 2–3 × $10^6$ each of cognate particles and CTLs were deposited on one side of a 14 mm microwell. A volume of 20 µl of the solution was rapidly transferred to one side of a 35 mm Petri dish containing a 14-mm microwell with a precision glass coverslip (MatTek, Ashland, MA, USA) and incubated at 37°C and 5% $CO_2$ for 10 min to allow the gel to polymerise into a 3D matrix with a high density of tumour cells (with or without co-embedded CTLs) that mimics a solid tumour mass. A second collagen gel was prepared containing 3 × $10^5$ CTLs that was deposited on the total surface of the microwell and allowed to polymerise at 37°C and 5% $CO_2$ for 10 min to create a 3D collagen matrix with dispersed cells (see *Figure 2A*). Finally, 1.5 ml cold phenol red-free TCM were gently added to the dish and incubated at 37°C and 5% $CO_2$ for 10 min, then four-dimensional confocal microscopy was performed as described below.

For experiments with a central tumouroid, 2 µl of a collagen preparation containing 3.5 × $10^5$ EL4 tumour cells (prepared as above) or 1.75 × $10^5$ cells each of CTLs and EL4 tumour cells were deposited in the centre of a well in a Greiner Sensoplate glass bottom 96-well optical plate (Sigma Aldrich). As above, a second collagen matrix (50 µl total volume) containing dispersed CTLs was polymerised on top of the first gel (see *Figure 1—figure supplement 1A*). The whole well was imaged in *x*, *y* over 14–16 hr by confocal microscopy as described below.

For experiments where WT OT1 and Ccr5$^{-/-}$ OT1 CTLs were co-embedded around a cognate tumouroid (containing WT OT1 and cognate tumour cells), the two CTL populations and the cognate tumour cells were labelled with CMTMR, CellTracker Green 5-chloromethylfluorescein diacetate (CMFDA) (Invitrogen, California, USA) or Deep Red dye (different combinations were used for different experiments).

For experiments with primary human T cells, cells that were embedded in the tumouroid were labelled with Deep Red dye and pre-stimulated with control or activating beads for 3 hr, whereas migrating cells were labelled with CMTMR. For experiments with CAR T cells, tumouroids were prepared with embedded CAR T cells and tumour cells (WK1 glioblastoma cells or NIH/3T3 control fibroblasts, the former EphA2-expressing, the latter EphA2-negative) labelled with Deep Red dye and migrating CAR T cells were labelled with CMFDA.

## Confocal fluorescence microscopy

Four-dimensional imaging data were collected through a 20 × water immersion objective, numerical aperture (NA) of 1.37 (Leica Microsystems, Wetzlar, Germany), on a Leica TCS SP5 confocal microscope equipped with a resonant scanner and an incubator that maintains 37°C and 5% $CO_2$ throughout imaging. LifeactGFP and CMFDA were excited at 488 nm, CMTMR at 561 nm and Deep Red at 633 nm wavelengths using a tunable white light laser (Leica Microsystems). Images were obtained from the *x*, *y* and *z* planes, with a total *z* thickness of 65 µm (with the lower 5 µm immediately above the glass rejected in order to avoid the inclusion of two-dimensional movements on the glass surface) and step size of 1.6 µm every 20 s for 1 hr, with three fields of view imaged sequentially at every timepoint. Imaging of whole wells with a central tumouroid was performed through a 10×/ 0.30 NA dry objective on the same instrument with an image taken every 5–6 min over 14–16 hr and the Leica LAS AF software was used to automate *x*, *y* tiling and facilitate image stitching (9 × 9 tiles).

## Long-term imaging of T cell migration on Nikon BioStation

For tumouroid samples that were imaged for 40 hr (*Video 1*), sample preparation was performed as described above, but in the centre of a 35 mm MatTek dish, using cognate tumour cells stained with CMTMR for the tumouroid and OT1GFP CTLs embedded in the surrounding collagen matrix. Time-lapse imaging data were collected through a 20×/0.8 NA air objective on a Nikon BioStation live-cell imaging platform (Nikon, Tokyo, Japan). Images were taken every 20 min and the BioStation IM software was used to automate *x*, *y* tiling and facilitate image stitching (9 × 11 tiles). Image processing for visualisation was performed in Fiji/ImageJ (US National Institutes of Health, Bethesda, MA, USA).

## Preparation of H-2K$^b$/SIINFEKL coated (cognate) beads or antibody-coated beads

Streptavidin-coated polystyrene particles (6.0–8.0 μm diameter; Spherotech Inc, Illinois, USA) were incubated with monobiotinylated H-2K$^b$/SIINFEKL (Tetramer Synthesis Service, John Curtin School of Medical Research, Australia National University, ACT, Australia) in BSA solution (phosphate-buffered saline with 20 mM HEPES, 150 mM NaCl and 2% bovine serum albumin) at 37°C for 45 min. The beads were then washed three times in BSA solution before use in subsequent experiments. For experiments with human T cells, beads were coated with 1 μg/ml IgG isotype control antibody or with 1 μg/ml anti-human CD3ε and anti-CD28 antibodies (ThermoFisher Scientific).

## 3D T cell migration assays using H-2K$^b$/SIINFEKL-coated (cognate) or non-cognate beads

For experiments where central tumoroids were constructed using Streptavidin-coated polystyrene particles, 2.5 μl of a collagen preparation containing $3.5 \times 10^5$ cognate (prepared as above) or non-cognate particles, or $1.75 \times 10^5$ each of cognate particles and CTLs were deposited in the centre of a well in a glass bottom 96-well optical plate. As above, a second collagen matrix (50 μl total volume) containing GFP expressing CTLs was polymerised on top of the first gel. The whole well was imaged in x, y over 16 hr by confocal microscopy as described above.

For experiments where CTLs were embedded around a cognate tumoroid containing H-2K$^b$/SIINFEKL coated polystyrene particles and Ccr5$^{-/-}$ OT1 CTLs, the embedded Ccr5$^{-/-}$ OT1 CTLs were labelled with Deep Red dye.

## Preparation of cognate and non-cognate supernatant

$2 \times 10^6$ cells each of OT1 CTLs and EL4 tumour cells (cognate or non-cognate) or polystyrene beads (coated with H-2K$^b$/SFKL or uncoated) were resuspended in 1.5 ml RPMI 1640 medium containing 0.5% bovine serum albumin (BSA) in 24-well plates. The plates were centrifuged for 5 min at $300 \times g$ followed by incubation at 37°C. After 3 hr, the samples were centrifuged and the supernatants collected and filtered through 0.22 μm pore polyethylsulfonate filters (Millipore, Burlington, MA, USA).

## Preparation of cognate supernatant for CBA time course following isolation of CTLs

EL4 cells were pulsed with 1 μg/ml SIINFEKL peptide overnight for 16 hr. $4 \times 10^6$ OT1 Ccr5$^{-/-}$ CTLs were labelled with CMFDA (Invitrogen, California, USA) and resuspended with $4 \times 10^6$ cognate EL4 cells in 5 ml TCM in a 6-well plate. The plate was centrifuged for 5 min at 300x g followed by incubation at 37°C. After 4 hr, the cells were pelleted, and the supernatant collected and filtered through a 0.22 μM pore polyethylsulfonate filter (Millipore, Burlington, MA, USA). The cells were collected, and $1.25 \times 10^6$ OT1 Ccr5$^{-/-}$ CTLs were sorted on the FACS-Melody flow sorter (BD Biosciences, Franklin Lakes, NJ, USA) based on CMFDA labelling. After sorting, the cells were split across seven wells in a 24-well plate, containing 620 μl of TCM each, and incubated at 37°C until supernatant collection. Supernatant from each well was collected at either 2, 4, 8, 16, 24, 48, or 72 hr post-sorting and filtered through a 0.22 μM pore polyethylsulfonate filter (Millipore, Burlington, MA, USA) for analysis by cytometric bead array (CBA).

## Transwell chemotaxis assay

Chemotaxis was assayed using a Transwell Chamber (Corning) with a 5 μm-pore size polycarbonate filter. Briefly, 600 μl of control medium (serum-free RPMI 1640 medium containing 0.5% BSA) or medium containing recombinant chemokines at 1, 10 or 100 ng/ml (Peprotech, Rocky Hill, NJ, USA) or filtered supernatant were placed in the lower chambers. $1 \times 10^6$ CTLs were placed in the upper chamber in 100 μl RPMI + 0.5% BSA and incubated at 37°C and 5% CO$_2$ for 3 hr (*Figure 2—figure supplement 1A*). For experiments with collagen gels, cells were embedded in liquid-phase collagen on ice (as per migration assay above). After 3 hr, cells were harvested from the lower compartment and analysed by flow cytometry using a BD LSR Fortessa X20 flow cytometer (BD Biosciences). Cell numbers were enumerated using SPHERO AccuCount blank particles (Spherotech Inc, Chicago, IL,

USA). Results were analysed using FlowJo software (FlowJo 10.2, Tree Star Inc, Ashland, OR, USA). The transmigration index is represented as bars ± SEM and was calculated as follows:

$$Transmigration\ index = \frac{number\ of\ cells\ transmigrated\ in\ sample}{number\ of\ cells\ transmigrated\ in\ control\ medium}$$

## Cytotoxicity assay

Cytotoxicity assays were performed using flow cytometry based on the ratio between live target and non-target cells. Effector CTLs (GFP+), cognate tumour cells and non-cognate tumour cells (either CMTMR+ or CMTMR- in different combinations) were mixed at a 1:1:1 ratio and the distribution of the three cell populations was measured by flow cytometry at 0 hr and 2 hr. Samples were run in duplicate in each experiment. The cytotoxicity index was calculated as:

$$Cytotoxicity\ index\,(\%) = \left[1 - \frac{\left(\frac{cognate\ cells\ (2h)}{non-cognate\ cells\ (2h)}\right)}{\left(\frac{cognate\ cells\ (0h)}{non-cognate\ cells\ (0h)}\right)}\right] \times 100$$

## Inhibitors and neutralising antibodies

In some experiments, the following inhibitors were used: 500 ng/ml Pertussis toxin from Bordetella pertussis (PTX) or inactive mutated version (m-PTX) (Sigma Aldrich, St Louis, MO, USA); 10 µg/ml Maraviroc (CCR5 antagonist) (Sigma Aldrich); 1 µM Cenicriviroc (CCR2 + CCR5 dual inhibitor) (AdooQ Bioscience, Irvine, CA, USA); 1 µM CCR2 antagonist (CAS 445479-97-0, Santa Cruz Biotechnology Inc, Dallas, TX, USA); 50 µM UCB35625 (CCR1 + CCR3 dual inhibitor) (R & D Systems); or DMSO (vehicle).

In some experiments, the following neutralising antibodies were used: anti-CCL1, anti-CCL3, anti-CCL4, anti-CCL5, anti-CCL9, and anti-XCL1 with corresponding isotype control antibodies (all from R & D Systems).

## Real-time quantitative PCR (qRT-PCR) of gene expression

To prepare cells for RNA extraction, $2 \times 10^6$ CTLs were incubated in 1.5 ml TCM in 24-well plates with equal numbers of either cognate or non-cognate EL4 tumour cells for 3 hr. Cells were then collected and centrifuged (300 × g, 5 min) and washed 2 × with PBS before being re-suspended in 2 ml cold TCM. A total of $1 \times 10^6$ GFP-expressing CTLs were sorted on the FACS Aria III flow sorter (BD Biosciences). Cells were then centrifuged (300 × g, 5 min) before RNA isolation using the RNeasy Mini Kit (Qiagen), according to manufacturer's instructions. 1.5 µg of RNA were reverse transcribed to cDNA using the High Capacity cDNA Reverse Transcription Kit (Applied Biosystems, ThermoFisher Scientific) according to manufacturer's instructions. Real-time quantitative PCR was carried out using chemokine and chemokine receptor primers predesigned and synthetised by Sigma-Aldrich (KiCqStart SYBR Green Primers). 20 ng cDNA were added to each well of a 96-well PCR plate with 1 µM forward and reverse primer for each gene. 40 cycles were performed, with denaturing temperature at 95°C for 15 s, annealing at 55°C for 30 s, and extension at 72°C for 30 s. The amount of amplicon was measured using SYBR Green and detected in a BIO-RAD CFX96 Real time system (Bio-Rad Laboratories, Hercules, CA, USA). The expression of each gene was normalised to the expression of the housekeeping genes β2-microglobulin (β2M) and ribosomal protein L13A (RPL13A).

The primer sequences used for qRT-PCR analysis of chemokine and chemokine receptor expression are found in the table below.

| Chemokine | Forward primer | Reverse primer |
|---|---|---|
| CCL1 | ATGCTTACGGTCTCCAATAG | TCTTCAGGTGATTTTGAACC |
| CCL3 | TTCTCTGTACCATGACACTC | CTCTTAGTCAGGAAAATGACAC |
| CCL4 | GGTATTCCTGACCAAAAGAG | TCCAAGTCACTCATGTACTC |
| CCL5 | AGGAGTATTTCTACACCAGC | CAGGGTCAGAATCAAGAAAC |
| CCL6 | CTTTCAAGACACTTCTTCAGAC | CTGCTGATAAAGATGATGCC |

*Continued on next page*

*Continued*

| Chemokine | Forward primer | Reverse primer |
|---|---|---|
| CCL7 | CTCTCTCACTCTCTTTCTCC | TCTGTAGCTCTTGAGATTCC |
| CCL8 | CTTCAACATGAAGATCTACGC | CTGGATATTGTTGATTCTCTCG |
| CCL9/10 | AATGTTTCACATGGGCTTTC | CAATGCATCTCTGAACTCTC |
| CCL12 | TGTGATCTTCAGGACCATAC | CATGAAGGTTCAAGGATGAAG |
| CCL17 | CATTCCTATCAGGAAGTTGG | CAGTCAGAAACACGATGG |
| CCL19 | TTCTTAATGAAGATGGCTGC | CTTTGTTCTTGGCAGAAGAC |
| CCL22 | CACATAACATCATGGCTACC | CAGAAGAACTCCTTCACTAAAC |
| CXCL4 | TAGCCACCCTGAAGAATG | GACATTTAGGCAGCTGATAC |
| CXCL9 | GAGGAACCCTAGTGATAAGG | GTTTGATCTCCGTTCTTCAG |
| CXCL10 | AAAAAGGTCTAAAAGGGCTC | AATTAGGACTAGCCATCCAC |
| CXCL11 | CGACAAAGTTGAAGTGATTG | GCACAGAGTTCTTATTGGAG |
| CXCL12 | GAAAGCTTTAAACAAGAGGC | GTGAAAGTACAGCAAAACTG |
| CXCL16 | CCATTCTTTATCAGGTTCCAG | CTTGAGGCAAATGTTTTTGG |
| **Chemokine receptor** | **Forward primer** | **Reverse primer** |
| CCR1 | ATACTCTGGAAACACAGACTC | GTCAAATTCTGTAGTTGTGGG |
| CCR2 | ACCACATGTGCTAAGAATTG | CTGGTTTTATGACAAGGCTC |
| CCR3 | TCACCAGAGACAAGTAGAATG | ACTCATATTCATAGGGTGTGG |
| CCR5 | AGACCTAAATCCTACCACAC | TGGCTTCAAACTATGGAAAC |
| **Housekeeping gene** | **Forward primer** | **Reverse primer** |
| β2-microglobulin | GTATGCTATCCAGAAAACCC | CTGAAGGACATATCTGACATC |
| RPL13A | CCTATGACAAGAAAAAGCGG | CAGGTAAGCAAACTTTCTGG |

## Mass spectrometry-based secretomics

T cell secretomes from interactions with cognate beads, cognate tumour cells, non-cognate-beads, non-cognate tumour cells, and secretomes from EL4 (tumour cells only) and OT1 (T cells only) (n = 3 per group) were prepared for mass spectrometry analysis as previously described (40, 41) with the following modifications. For all our experiments with magnetic beads, we used a 1:1 combination mix of the two types of commercially available carboxylate beads (Sera-Mag Speed beads, #65152105050250, #45152105050250, ThermoFisher Scientific). Beads were prepared freshly each time by rinsing with water three times prior to use and stored at 4°C at a stock concentration of 20 µg/µl. Samples were transferred to a 2 ml LoBind deep well plate (Eppendorf, Hamburg, Germany) and reduced with 2 M Dithiothreitol (DTT, 50 mM final conc.) for 1 hr at 37°C. Samples were then alkylated with 1M Iodoacetamide (IAM) (100 mM final conc.) for 30 min in the dark at room temperature (RT). Samples were quenched with 2M DTT (250 mM final conc.) and 5 µl of the concentrated bead stock carboxylate beads (20 µg/µl) were then added to each sample followed by the addition of acetonitrile (ACN) to a final concentration of 70% (v/v). Mixtures were left to incubate upright at RT for 20 min to allow proteins to precipitate onto the beads. The beads were placed on a magnetic rack and washed twice with 70% ethanol and once with ACN (500 µl washes). ACN was completely evaporated from the plate using a CentriVap (Labconco, Kansas City, MO, USA) prior to the addition of 40 µl digestion buffer (10% 2-2-2-Trifluorethanol (TFE)/100 mM NH4HCO3) containing 1 µg Trypsin-gold (Promega, V5280) and 1 µg Lys-C (Wako). The plate was briefly sonicated in a water bath to disperse the beads, and the plate transferred to a ThermoMixer instrument (Eppendorf) for enzymatic digestion at 37°C for 1 hr (1200 rpm). The supernatant comprising of peptides was then collected from the beads using a magnetic rack (Ambion, Thermo Fisher Scientific) and an additional elution (40 µl 2% Dimethyl sulfoxide, DMSO, Sigma) was performed on the beads. The eluates were pooled together and transferred to pre-equilibrated C18 stage tips for sample clean-up. Briefly, two plugs of C18 resin (3M Empore, 66883 U) were prepared in 200 µl unfiltered tips, pre-wetted with 100 µl methanol followed by sequential washes with 100 µl 80% ACN/5% formic acid (FA), 50%

ACN/5% FA and 5% FA. The pooled peptides were then added to the spin tip and the eluate collected into a fresh lo-bind Eppendorf tube. Bound peptides were washed twice with 5% FA. Elutions (50 µl) were performed sequentially with 50% ACN/5% FA followed by 80% ACN/5% FA and collected into fresh Eppendorf tubes. All spins were performed on a benchtop centrifuge at 500 × g (1000–2000 rpm) speeds. The eluates were lyophilized to dryness in MS vials using a CentriVap (Labconco) prior to reconstituting in 20 µl Buffer A (0.1% FA/2% ACN) ready for MS analysis.

## Detection of secreted chemokines in supernatant by ELISA and CBA

The absolute concentration of chemokines present in supernatants was assessed by ELISA for CCL1 and CCL9 (Sandwich ELISA kits, OriGene Technologies, Rockville, MD, USA) and Cytometric Bead Array (CBA) for CCL3, CCL4, CCL5, and CXCL10 (LEGENDplexTM Multi-Analyte Flow Assay Kit, BioLegend, San Diego, CA, USA) as per manufacturers' instructions. All samples were run in triplicate. Chemokine levels in the supernatants were interpolated from standard curves generated using recombinant proteins provided in the kits. IFN-γ and TNF-α were used as positive controls for CTL activation.

For ELISA, 100 µl of diluted supernatant and 100 µl of assay diluent were incubated in a 96-well antibody-coated plate for 2 hr. After 3–5 washes, the samples were incubated with 100 µl of detection antibody for 1 hr followed by the addition of 100 µl of streptavidin-HRP secondary antibody for 30 min. Finally, the HRP substrate tetramethyl benzidine was added and plates were analysed on a FLUOstar Omega microplate reader (BMG Labtech GmbH, Ortenberg, Germany) by reading the absorbance at 450 nm.

For CBA, 25 µl of each supernatant were mixed with 25 µl of captured beads against the desired chemokine for 2 hr in a V-bottom plate. After two washes, the samples were incubated with 25 µl of detection antibody for 1 hr followed by the addition of 25 µl of PE-conjugated secondary antibody. Data were acquired on a Fortessa X20 flow cytometer and analysed by FlowJo software.

## Preparation and staining of 3D CCL3 gradient

As described previously, a high density tumouroid of 1:1 cognate EL4s and OT1 CTLs in a 3D collagen matrix was prepared on one side of a 35-mm Petri dish containing a 14 mm microwell with a precision glass coverslip (MatTek, Ashland, MA, USA). 1 ml of CBA beads from the LEGENDplex Multi-Analyte Flow Kit (BioLegend, San Diego, CA, USA) were pelleted at 250 x g for five mins and supernatant was removed. The beads were resuspended in a second collagen gel mixture that was deposited adjacent to the tumouroid, and allowed to polymerise at 37°C and 5% CO₂ for 10 min to create a 3D collagen matrix with dispersed CBA beads. 2 ml of warm TCM was added to the dish, which was then returned to 37°C for either 2 or 3 hr to allow formation of a CCL3 gradient from the tumouroid. The TCM was then removed and 1 ml of 10% Neutral Buffered Formalin (NBF, Sigma Aldrich) was added to the dish to fix the gel, which was placed on a plate rocker to incubate for 1 hr at room temperature. The 10% NBF was then removed, and the dish washed 3x with 2 ml FACS wash for 30 min per wash, on the plate rocker at room temperature. The FACS wash was then removed, and the gel resuspended in 2 ml of 1 µg/ml anti-murine CCL3 biotinylated rabbit antibody in FACS wash (PeproTech, Rocky Hill, NJ, USA) and incubated for 3 hr at room temperature, rocking. The gel was then washed twice with FACS wash as described previously, and then a third time with an overnight incubation at room temperature. The wash buffer was then removed and replaced with Streptavidin-PE (also from the LEGENDplexTM Multi-Analyte Flow Kit) and incubated for 45 min at room temperature, rocking. The gel was then washed twice with FACS wash for 30 min per wash, rocking and then a third time overnight before imaging.

## Generation of mCCL3-mScarletI, mCCL4-miRFP670, SSIEFARL(HSV)-mTagBFP2, and SIINFEKL(OVA)-mTagBFP2 fusion proteins

The coding sequences for all fusion proteins were synthesized as gBlocks by Integrated DNA Technologies, Inc (Skokie, IL, USA) and cloned into the MSCV-based retroviral expression vector 'LENC' (kind gift from Johannes Zuber) that we modified to contain multiple cloning sites. The design of these sequences was based on published sequences of murine CCL3 (GenBank accession # NM_011337.2), murine CCL4 (#NM_013652.2), mScarletI (# KY021424.1) and miRFP670 (KX421097.1). Sequences encoding the mCCL3-mScarletI and mCCL4-miRFP670 fusion proteins

were separated by the 16-residue SGGGGSGGGGSGGGGS linker and cloned into the AgeI/HpaI sites. Sequence encoding bicistronic expression of mTagBFP2 fluorescent protein (*Subach et al., 2011*) with the SSIEFARL or SIINFEKL epitopes were linked by the viral F2A sequence VKQTLNFD LLKLAGDVESNPGP and cloned into the BglII/HpaI sites. The following coding sequences were synthesized as oligos for integration into retroviral expression vector 'LENC'.

## mCCL3-mScarletI

ctaccctcgtaaaggatccttcgaagatctacgtatgcatacgcgtataccggtgccaccatgaaggtctccaccactgcccttgctgttcttc
tctgtaccatgacactctgcaaccaagtcttctcagcgccatatggagctgacaccccgactgcctgctgcttctcctacagccggaagattc
cacgccaattcatcgttgactattttgaaaccagcagcctttgctcccagccaggtgtcattttcctgactaagagaaaccggcagatctgcg
ctgactccaaagagacctgggtccaagaatacatcactgacctggaactgaatgcctccggaggaggaggatccggaggaggaggat
ccggaggaggaggatccgtgagcaagggcgaggcagtgatcaaggagttcatgcggttcaaggtgcacatggagggctccatgaac
ggccacgagttcgagatcgagggcgagggcgagggccgcccctacgagggcacccagaccgccaagctgaaggtgaccaagggtg
gccccctgcccttctcctgggacatcctgtcccctcagttcatgtacggctccagggccttcatcaagcaccccgccgacatccccgactact
ataagcagtccttccccgagggcttcaagtgggagcgcgtgatgaacttcgaggacggcggcgccgtgaccgtgacccaggacacctc
cctggaggacggcaccctgatctacaaggtgaagctccgcggcaccaacttccctcctgacggcccccgtaatgcagaagaagacaatgg
gctgggaagcgtccaccgagcggttgtaccccgaggacggcgtgctgaagggcgacattaagatggccctgcgcctgaaggacggc
ggccgctacctggcggacttcaagaccacctacaaggccaagaagcccgtgcagatgcccggcgcctacaacgtcgaccgcaagttgg
acatcacctcccacaacgaggactacaccgtggtggaacagtacgaacgctccgagggccgccactccaccggcggcatggacgagct
gtacaagtgaccgcggttaactgcagcgctagcatatgtcgacagtttgtttg.

## mCCL4-miRFP670

ctaccctcgtaaaggatccttcgaagatctacgtatgcatacgcgtataccggtgccaccatgaagctctgcgtgtctgccctctctctcctct
tgctcgtggctgccttctgtgctccagggttctcagcaccaatgggctctgaccctcccacttcctgctgtttctcttacacctcccggcagctt
cacagaagctttgtgatgatgattctatgagaccagcagtctttgctccaagccagctgtggtattcctgaccaaaagaggcagacagatctg
tgctaaccccagtgagccctgggtcactgagtacatgagtgacttggagttgaactccggaggaggaggatccggaggaggaggatc
cggaggaggaggatccgtagcaggtcatgcctctggcagccccgcattcgggaccgcctctcattcgaattgcgaacatgaagagatc
caacctcgccggctcgatccagccgcatggcgcgcttctggtcgtcagcgaacatgatcatcgcgtcatccaggccagcgccaacgccg
cggaatttctgaatctcggaagcgtactcggcgttccgctcgccgagatcgacggcgatctgttgatcaagatcctgccgcatctcgatc
ccaccgccgaaggcatgccggtcgcggtgcgctgccggatcggcaatccctacggagtactgcggtctgatgcatcggcctccgga
aggcgggctgatcatcgaactcgaacgtgccggcccgtcgatcgatctgtcaggcacgctggcgccggcgctggagcggatccgcac
ggcgggttcactgcgcgcgctgtgcgatgacaccgtgctgctgtttcagcagtgcaccggctacgaccgggtgatggtgtatcgtttcg
atgagcaaggccacggcctggtattctccgagtgccatgtgcctgggctcgaatcctatttcggcaaccgctatccgtcgtcgactgtcc
cgcagatggcgcggcagctgtacgtgcggcagcgcgtccgcgtgctggtcgacgtcacctatcagccggtgccgctggagccgcg
gctgtcgccgctgaccgggcgcgatctcgacatgtcgggctgcttcctgcgctcgatgtcgccgtgccatctgcagttcctgaaggaca
tgggcgtgcgcgccaccctggcggtgtcgctggtggtcggcggcaagctgtggggcctggttgtctgtcaccattatctgccgcgctt
catccgtttcgagctgcgggcgatctgcaaacggctcgccgaaaggatcgcgacgcggatcaccgcgcttgagagctgaccgcgg
ttaactgcagcgctagcatatgtcgacagtttgtttg.

## mTagBFP2_F2A_SIINFEKL(OVA)

ctaccctcgtaaaggatccttcgaagatctacgtatgcatacgcgtataccggtgccaccatgagcgaactgatcaaagagaacatgcaca
tgaagctgtacatggaaggcaccgttgacaaccaccacctttaagtgcacgtctgagggtgagggtaagccgtacgaaggcacccaaacc
atgcgtatcaaagttgtggagggcggtccactgccgttcgcttttgacattctggcgaccagcttcctgtacggttccaaaacgttcattaac
catactcagggcattccggatttcttcaaacagagctttccggaaggtttcacctgggagcgtgtcaccacgtatgaagatggtggtgtgtt
gaccgccacccaagatacctccctgcaagatggctgtctgatctataacgtgaaaattcgtggcgtcaactttacgagcaatggtccggtg
atgcagaagaaaccctgggttgggaggcgtttacggaaaccctgtatccggccgatggtggcctggagggccgtaacgacatggca
ctgaagctggttggtggcagccatttgatcgcaaatgccaagacgacgtaccgcagcaagaaaccggcgaaaaatctgaagatgccgg
gtgtttactatgtcgactaccgtctggaacgcattaaagaagcgaataatgagacttacgtggagcagcacgaggttcagtcgcgcgct
attgcgacttgcctagcaagctgggtcataaactgaatctcgaggtaggctccggagtgaaacagactttgaattttgaccttctcaagttg
gcgggagacgtggagtccaacccagggcccaccggtatgggttgctgcttctccaagaccggctccggatatccatatgatgtgccgga
ttatgctagtgggagtgggacaatgagcatgttggtgctgttgcctgatgaagtctcaggccttgagcagcttgagagtataatcaactt
gaaaaactgactgaatggaccagttctaatgttatggaagagaggaagatcaaagtgtacttacctcgcatgaagatggaggaaaaata
caacctcacatcgtcttaatggctatgggcattactgacgtgtttagtcttcagccaatctgtctggcatctcctcagcagagagcctgaag
atatctcaagctgtccatgcagcacatgcagaaatcaatgaagcaggcagagaggtggtagggtcatcacaattggacccggcatggga
gcgcaacgaccctacgcagcagatccccaagctggtcgcaaacaacacccggctatgggtttattgcgggaacggcaccccgaacgag

ttgggcggtgccaacatacccgccgagttcttggagaacttcgttcgtagcagcaacctgaagttccaggatgcgtacaacgccgcggg
cgggcacaacgccgtgttcaacttcccgcccaacggcacgcacagctgggagtactggggcgctcagctcaacgccatgaagggtgac
ctgcagagttcgttaggcgccggccatatgcaattgacgcgttaaagcttatcgatccgcggttaactgcagcgctagcatatgtcgacag
tttgtttg.

### mTagBFP2-F2A-SSIEFARL(HSV)

ctaccctcgtaaaggatccttcgaagatctacgtatgcatacgcgtataccggtgccaccatgagcgaactgatcaaagagaacatgcaca
tgaagctgtacatggaaggcaccgttgacaaccaccactttaagtgcacgtctgagggtgagggtaagccgtacgaaggcacccaaacc
atgcgtatcaaagttgtggagggcggtccactgccgttcgcttttgacattctggcgaccagcttcctgtacgggttccaaaacgttcattaac
catactcagggcattccggatttcttcaaacagagctttccggaaggtttcacctgggagcgtgtcaccacgtatgaagatggtggtgtgtt
gaccgccacccaagatacctccctgcaagatggctgtctgatctataacgtgaaaattcgtggcgtcaactttacgagcaatggtccggtg
atgcagaagaaaaccctgggttgggaggcgtttacggaaacccctgtatccggccgatggtggcctggagggccgtaacgacatggca
ctgaagctggttggtggcagccatttgatcgcaaatgccaagacgacgtaccgcagcaagaaaccggcgaaaaatctgaagatgccgg
gtgtttactatgtcgactaccgtctggaacgcattaaagaagcgaataatgagacttacgtggagcagcacgaggttgcagtcgcgcgct
attgcgacttgcctagcaagctgggtcataaactgaatctcgagggaggctccggagtgaaacagactttgaattttgaccttctcaagttg
gcgggagacgtggagtccaacccagggcccaccggtatgggttgctgcttctccaagaccggctccggatatccatatgatgtgccgga
ttatgctagtgggagtgggacaatgagcatgttggtgctgttgcctgatgaagtctcaggccttgagcagcttgagagtagtatagagttt
gccaggctgactgaatggaccagttctaatgttatggaagagaggaagatcaaagtgtacttacctcgcatgaagatggaggaaaaata
caacctcacatctgtcttaatggctatgggcattactgacgtgtttagctcttcagccaatctgtctggcatctcctcagcagagagcctgaag
atatctcaagctgtccatgcagcacatgcagaaatcaatgaagcaggcagagaggtggtagggtcatcacaattggacccggcatggga
gcgcaacgaccctacgcagcagatccccaagctggtcgcaaacaacacccggctatgggtttattgcgggaacggcaccccgaacgag
ttgggcggtgccaacatacccgccgagttcttggagaacttcgttcgtagcagcaacctgaagttccaggatgcgtacaacgccgcggg
cgggcacaacgccgtgttcaacttcccgcccaacggcacgcacagctgggagtactggggcgctcagctcaacgccatgaagggtgac
ctgcagagttcgttaggcgccggccatatgcaattgacgcgttaaagcttatcgatccgcggttaactgcagcgctagcatatgtcgacag
tttgtttg.

### Retroviral transduction of EL4 cells for generation of CCL3/CCL4-secreting tumour cells

To transduce EL4 cells, retrovirus pseudotyped with the vesicular stomatitis virus (VSV-G) envelope was produced by polyethylenimine (PEI, molecular weight 4000, PolySciences Catalogue No 24885–2, Warrington, PA, USA) transfection of GP2-293 cells (Clontech, Palo Alto, CA, USA). NaOH-neutralised PEI (1 mg/ml) was complexed with 6.8 µg of rMSCV-mCCL3-mScarletI and rMSCDV-mCCL4-miRFP670 and 3.2 µg of pMD2.G plasmid (VSVG coding sequence expressed from the CMV promoter, kind gift of Didier Trono) for 30 min at room temperature before addition to $7 \times 10^6$ GP2-293 cells. At 72 hr after transfection, viral supernatant was used to transduce EL4 cells and fluorescent EL4 cells were sorted (BD FACS Aria III) 72 hr after transduction. GP2-293 cells were maintained in DMEM (Gibco) containing 4.5 g/L glucose, 4 mM L-glutamine and 1 mM sodium pyruvate supplemented with 10% heat-inactivated FBS. Sorted CCL3/CCL4-secreting or mTagBFP2-expressing EL4 tumour cells were cultured in TCM with routine passaging three times per week, maintained at cell densities under $1 \times 10^6$/ml.

### Measurement of in vivo leukocyte infiltration into tumours by flow cytometry

A total of $1 \times 10^6$ EL4 cells transduced to express CCL3-mScarletI and CCL4-miRFP670 or WT EL4 cells were injected subcutaneously into contralateral flanks of 8-week-old Rag$^{-/-}$ or PTPRCA mice. The tumours were allowed to grow for 7–10 days before day 6 or 7 effector CTLs were adoptively transferred via tail vein injection in 200 µl PBS. 24 to 72 hr post T cell transfer, mice were euthanised by $CO_2$ asphyxiation. The spleens and both tumours were collected and dissociated with 1 mg/ml collagenase IV (Sigma-Aldrich) for 30 min at 37°C (shaking at 800 rpm). The samples were filtered through 70 µm cell strainers to obtain single-cell suspensions. Tumours or spleens were resuspended in final volumes of 2 ml or 5 ml FACS wash buffer (2% HI-FCS, 2 mM EDTA and 0.02% sodium azide in $1 \times$ PBS), respectively. 100 µl of cells from these suspensions were mixed with 100 µl of buffer containing $2 \times 10^4$ AccuCount Blank Particles to determine the absolute number of infiltrating cells. Remaining cells were stained with fluorescent conjugated antibodies against CD11b (Brilliant Violet 711; clone M1/70), NK1.1 (Alexa Fluor 488; PK136), Ly-6C Antibody (FITC; HK1.4), CD90.2 (Brilliant

Violet 510; 30-H12), CD64 (Brilliant Violet 421; X54-5/7.1) and CD45 (APC/Fire 750; 30-F11), CD11c (FITC; N418), I-A/I-E or MHC II (Pacific Blue, M5/114.15.2), or CD45.1 (APC/Fire 750; A20) for 30 min on ice in FACS wash buffer containing 10% normal mouse serum. Final cell suspensions were prepared in 200 µl cold FACS wash buffer containing 0.5 µg/ml 4′, 6-diamidino-2-phenylindole (DAPI) and acquired on the BD Fortessa X20 flow cytometer. Flow cytometry data were analysed with FlowJo software.

To quantify recruitment of WT and Ccr5$^{-/-}$ CTLs to non-cognate or CCL3/CCL4-secreting tumours, Rag$^{-/-}$ mice were inoculated subcutaneously with $1 \times 10^6$ WT EL4 or CCL3/CCL4-secreting EL4 tumour cells on contralateral flanks. On day 10, equal numbers of WT and Ccr5$^{-/-}$ effector OT1 CTLs ($12.5 \times 10^6$ each), distinguished by CellTracker Deep Red, CellTracker Green, or CellTracker Orange CMTMR labelling, were co-injected intravenously. OT1 CTLs in single-cell suspensions prepared from both tumours and the spleen were enumerated by flow cytometry 22 hr later. Ratios of Ccr5$^{-/-}$: WT CTLs in tumours were normalised to the ratio of Ccr5$^{-/-}$:WT in spleens for each mouse. Data in *Figure 5F–H* are from four independent experiments.

## Measurement of tumour clearance in vivo

68-week-old Rag$^{-/-}$ mice were engrafted by subcutaneous injection of $1 \times 10^6$ EL4.OVA tumour cells on both flanks. Tumour growth was monitored by daily caliper measurements in two orthogonal dimensions, where tumour volume in mm$^3$ is calculated as $= 0.5 \times$ length (mm) x width (mm)$^2$. On day 9 post-engraftment, $5 \times 10^6$ OT1 or $5 \times 10^6$ Ccr5$^{-/-}$ OT1 CTLs were adoptively transferred into the mice by tail-vein injection. Each tumour volume was normalised to volumes on the day of T cell transfer.

## Preparation of shear-thinning hydrogel

A trifluoroacetic acid (TFA)-free, self-assembling peptide derivative, Fmoc-DDIKVAV, was synthesized using standard Fmoc chemistry procedure on a 1.5 mmol scale. Fmoc-Asp-Wang resin, Fmoc protected amino acids, 1-hydroxybenzotriazole (HOBt), N,N-diisopropylethylamine (DIPEA), and 2-(1H-Benzotriazole-1-yl)−1,1,3,3-Tetramethyluronium hexafluorophosphate (HBTU) were purchased from GL Biochem (Shanghai, China), with other reagents sourced from Sigma-Aldrich (Australia). Solution-phase anion exchange with excess aqueous hydrochloric acid (HCl) was used to remove the TFA counterion, followed by lyophilization. Reverse phase high performance liquid chromatography (RP-HPLC) confirmed 95% purity. Gelation was initiated using a well-established pH switch (*Rodriguez et al., 2013*). Briefly, one wt% hydrogels were prepared from amorphous Fmoc-DDIK-VAV powder. This was suspended in deionized water, before the pH was subsequently raised with a minimal amount of 0.5 M NaOH to ensure solubilization. Gelation occurred spontaneously when the pH was lowered to 7.4 using dropwise 1 M HCl, and water used to ensure a concentration of 20 mg/ml. Fourier transform infrared (FTIR), circular dichroism (CD) spectroscopy, and transmission electron microscopy were performed to verify synthesis and structure of the desired nanofibrillar structure (data not shown). CCL3 was loaded within the molecular hydrogel via our recent shear-containment methodology (Nisbet D. R.; *Nisbet et al., 2018*). Briefly, the peptide hydrogels were disrupted via the application of shear force until a gel-solution transition was observed and then 10 µg/ml of recombinant CCL3 and 20 µg/ml of 10 kDa Dextran labelled with AlexaFluor647 (Dextran-AF647) were added prior to syringe administration and spontaneous re-assembly.

## Intravital microscopy

-week-old Rag$^{-/-}$ mice were engrafted by subcutaneous injection of $1 \times 10^6$ EL4 tumour cells into contralateral flanks. On day five post-engraftment (48 hr before imaging), a total of $40 \times 10^6$ OT1 and Ccr5$^{-/-}$ OT1 CTLs (prepared 1:1) were adoptively transferred into the mice by tail-vein injection. The two CTL populations were labelled with CFSE and Cell Trace Violet (Invitrogen) at 100 mM final concentration (dye selections were inverted for different experiments).

Mice (17–18 g) were anaesthetized by intraperitoneal injection of 100 mg/kg body weight keta-mine and 15 mg/kg body weight xylazine. Tumours were surgically exposed and prepared for intravital microscopy by skin flap surgery. The tumour was stabilized on a coverslip on a microscope stage with intact vasculature and innervation. Intravital imaging was performed on a Nikon A1R inverted laser scanning confocal microscope fitted with a CFI APO LWD Lambda series 20×/0.95 NA water

immersion objective, an Okolab humidified temperature-controlled microscope enclosure, objective heater and a custom-made stage insert. Heating was adjusted to maintain the temperature at 34°C within the chamber. Cell Trace Violet was excited with a 405 nm laser, CFSE with a 488 nm laser and Dextran-AF647 with a 640 nm laser. Images were taken using NIS Elements software (Nikon).

During imaging, tumours were injected with sub-µl volumes of hydrogel containing recombinant CCL3 chemokine and Dextran-AF647 by using a customized stereotaxic injection unit (Kopf, Model 5000 microinjection unit, Tujunga, CA, USA) equipped with a syringe and 29-gauge needle with 45° bevel angle (Hamilton Company Inc, Reno, NV, USA).

Intravital imaging data were analysed with Imaris 9.2.1 software (Bitplane AG, Zurich, Switzerland) by use of the Spots function to determine the number of cells in the Ccr5$^{-/-}$ and WT channels at each time point. Data were then graphed in Plot2 for Mac 2.6.1 (apps.micw.org). In *Figure 5—figure supplement 3B*, densities were linearly interpolated at regular 10 min intervals in Matlab (MathWorks) to obtain the mean curve (due to differences in sampling frequencies between the independent experiments).

## T cell migration imaging analysis

Image analysis for the tracking of CTL movement in 3D was performed with Imaris 9.2.1 software (Bitplane AG, Zurich, Switzerland). Cells were segmented by creating surfaces with a filter below 100 µm$^3$ to discard cell debris. Cells were tracked using autoregressive motion, applying a threshold of 10 min to filter out tracks of insufficient duration. Intensity, morphological and tracking data were then exported, yielding multiple motility parameters used to quantify population-wide migration behaviours.

Track displacement is the net distance between first and last position:

$$D = \sqrt{D_x(t_L, t_F)^2 + D_y(t_L, t_F)^2 + D_z(t_L, t_F)^2}$$

where D is net displacement, $D_i$ is the net displacement in the *i*-axis between $t_L$, the position of the cell at the last timeframe of the track, and $t_F$, the position of a cell at the first timeframe of the track.

The mean speed was calculated by dividing the total length a cell travels by the duration of the track.

The Forward Migration Index (FMI) is a measure of the directionality of the cell trajectory along the *x* axis (i.e. towards the tumour). The FMI is defined as the displacement in *x* ($D_x$) divided by the net displacement (D), therefore equivalent to the cosine of the angle between the displacement vector and the vector pointing towards the tumouroid.

$$FMI = \frac{D_x}{D}$$

## T cell migration analysis in experiments with central tumouroids

Image analysis for the population-wide behaviour of CTLs in whole wells was performed with Imaris 9.2.1 software. Tumouroids and CTLs were segmented using the Imaris 'surface' and 'spots' functions, respectively. The shortest distance between each CTL and the tumouroid surface was then calculated for each timepoint using the 'Distance transform' ImarisXT module employing MATLAB (MathWorks Inc, Natick, MA, USA), for CTLs located either outside or inside the tumouroid surface. MATLAB was then used to calculate the fraction of total CTLs infiltrated into the tumouroid as well as the mean infiltration depth into the tumouroid (measured as the shortest distance from the tumouroid edge for infiltrated CTLs). For visualisation and preparation of movies and figures, Fiji/ImageJ was employed in conjunction with Imaris. For the visualisation of experiments with human T cells, Fiji/ImageJ bleach correction (histogram matching) was applied to eliminate fluctuations in laser power that occurred over the 14 hr of imaging.

Kymographs were generated to quantify the density of cells with respect to the tumouroid interface over time. Briefly, the cumulative sum of cells residing within radial distance *r* of the tumouroid interface was calculated for all *r*, for each imaging frame. These data were then normalised for cell counts over area of *r*, and smoothed over *r* using the 'fda' (version 2.4.8) functional regression package in R. The smoothed functions are differentiated with respect to *r* to yield the density of cells with respect to distance from tumouroid.

## Quantification of T cell swarming with 'M' metric

We quantify the spatial organisation of cells with respect to the tumouroid through a swarming index (the 'M' metric) (code available). Briefly, this index ranges between −1 and 1, with −1 indicating an even distribution of all cells along the well perimeter, and one indicating all cells as residing within the tumouroid. A value of 0 represents a uniform distribution of cells within the well, but outside of the tumouroid (*Figure 1C*). The swarming index is independently quantified for each imaging frame, and the subsequent timeseries denotes the spatial evolution of cells with respect to the tumouroid over time. Custom code and notes for generating 'M' are available at https://github.com/marknormanread/TcellSwarming (*Niño et al., 2020*; copy archived at swh:1:rev:74c6678c55317a0aac98a70939e0c92fb29e58ad).

## Mathematical modelling of T cell attraction

Simulations of T cell motility and attraction were conducted through an adaptation of the agent-based simulation, 'motilisim' (*Read et al., 2016*). Code is available at https://github.com/marknormanread/TcellSwarming. T cells are modelled as non-overlapping spheres of 12 μm diameter in a spatially explicit 3D environment. We modelled a whole well environment of 6.8 mm diameter containing a concentric tumouroid environment of 2.4 mm diameter, both with height 60 μm; these boundaries are impermeable. In all cases, a total of 35,000 cells are simulated within this environment.

T cell motility patterns are sampled from the tracks of OT1 CTLs observed through 3D ex vivo imaging of space immediately adjacent to a tumouroid (*Figure 2A,B*). The sampling methodology is performed via 'bootstrapping': each modelled T cell samples (with replacement) and then re-enacts 10 min blocks of observed ex vivo CTL motility (*Figure 1—figure supplement 1B, C*). The pool of blocks to be sampled comprises all unique consecutive 10 min durations of track data, across all imaged CTLs in the source data. Each block describes a given CTL's sequence of displacements, and their relative orientations, within the 10 min window.

Modelled agents conducting 'undirected' motion sample their motilities from experiments involving random migration of CTLs in the absence of a tumouroid. Conversely, 'chemotactic' agents sample the motilities of CTLs imaged in the presence of a tumouroid containing cognate tumour cells and embedded CTLs (*Figure 1—figure supplement 1A*). In this case, each block captures CTL motility with respect to the tumouroid, and the modelled agent reinterprets this with respect to the chemokine gradient (*Figure 1—figure supplement 1B*); 'up' ($0x$, $0y$, $1z$) is maintained during these rotational translations. Agents entering the tumouroid environment cease bootstrapped motility reconstruction and instead move towards the tumouroid centre through the $xy$ plane, holding a stable $z$-location, at 0.15 μm/min. Upon entering the tumouroid, agents in the 'positive attraction' simulation scenario (*Figure 1F*) secrete a soluble chemotactic factor. No such secretion takes place in the simulated 'no attraction' scenario.

Chemokine-secreting agents do so at 1000 molecules/min. The chemokines have a diffusion coefficient of 250 μm$^2$/s ($2.5 \times 10^{-6}$ cm$^2$/sec), given CCL3 and CCL4 molecular weights of 10 kDa approximately. The chemokine concentration at a given point in space and time is resolved through applying the heat kernel to all prior secretion events, having recorded their location, time and quantity. The heat kernel is a numerical solution for the modelling of diffusion. Agents determine the chemokine concentration at six points around their spherical extremity (where the sphere intersects the $\pm x, \pm y$ and $\pm z$ axes, relative to the agent), and from this determine the chemokine gradient direction. Agents are 'chemotactic' only whilst the maximum perceived chemokine concentration lies above a 'chemotaxis actuation' threshold, and below another 'desensitisation' threshold. Both thresholds are agent-specific, with each agent sampling a threshold from a log-normal distribution; agents hence differ in their sensitivity to chemokines.

Whilst agents are permitted to move freely within this three-dimensional space during simulation, only the locations of agents intersecting an $xy$ plane at depth 30 μm are recorded for downstream analysis; this is reflective of the restricted $z$-depth of the whole well imaging experiments, facilitating comparison of results. Agent location changes are updated in 20 s increments of simulated time, but recorded at 5 min intervals, in line with experimental whole well imaging. Agent diameters are enlarged to 48 μm to facilitate visualisation in simulations.

## RNA sequencing (RNAseq) and bioinformatics analysis

Non-cognate or cognate EL4 tumour cells were labelled with 10 μM CMTMR for 20 min in serum-free RPMI 1640 medium and washed twice. Labelled cells were returned to TCM to recover for 2 hr at 37°C and 5% $CO_2$ before use. Effector OT1 CTLs were stimulated with non-cognate or cognate tumour cells at a 1:1 ratio for a final cell concentration of $6 \times 10^6$/ml in 250 μl of TCM per well in 96-well U-bottomed plates. The plates were centrifuged at $300 \times g$ for 3 min before incubation for 2.5 hr at 37°C and 5% $CO_2$. OT1 and CMTMR+ tumour cells were then sorted (two-way sort) on the FACS-Aria III flow sorter (BD Biosciences). To prepare OT1 stimulated by cognate beads, OT1 CTLs were incubated with beads (prepared as described above) for 3 hr. Non-cognate (OT1 CTLs conjugated with EL-4) or cognate (OT1 CTLs conjugated with EL-4 pulsed with SIINFEKL as described above) supernatants were prepared and filtered as described above and were then used to incubate $3 \times 10^6$/ml in 250 μl OT1 CTLs for 3 hr prior to RNA extraction. Total RNA was prepared from up to $10^6$ cells using the RNeasy Mini RNA Isolation Kit (Qiagen, Hilden, Germany) as per manufacturer's instructions. The library was prepared using the TruSeq Stranded mRNA and sequenced on the Illumina NextSeq500 (Illumina, San Diego, CA, USA) generating paired-end 75 bp read lengths (Ramaciotti Centre for Genomics, University of New South Wales, Sydney, NSW, Australia).

For data analysis of RNAseq data, quality check of the raw reads was performed with FastQC (version 0.11.5). Trimmomatic (version 0.36) was used to trim low quality reads with low-quality scores using the following parameters: leading = 3; trailing = 3; window_len = 4; window_qual = 15; minlen = 50; avgqual = 20. Low-quality reads that did not pass this step were removed from downstream analysis. The reads were aligned against the GRCm38 reference mouse genome with Tophat2 using default parameters. Read counts were quantified by FeatureCounts (version 1.5.1) (results provided in *Supplementary file 1*). The distribution of read counts revealed a uniform distribution across samples; therefore, no further normalisation was necessary.

Differential expression analysis was performed from the total read counts by first calculating the fold change between pairs of samples. For the pairwise differential expression analyses, only genes with at least 100 reads in at least one of the two analysed samples were considered. The detection of differentially expressed genes was based on the fold-change between two conditions.

The heatmap in *Figure 3B* was obtained by filtering for genes that encode secreted proteins (based on UniProtKB database; available at www.uniprot.org) from the total RNAseq data (*Supplementary file 1*). Fold changes were used to identify genes differentially expressed between pairs of samples. Read counts corresponding to these genes were then utilised to cluster the samples using the package pheatmap in R with the option of 'scale = row'. Statistical tests and heatmaps were performed with the pheatmap R package.

The heatmap shows the differential expression of genes via z-scores (also called standard scores), obtained by pairwise comparisons between all conditions for each gene. For each gene, the z-scores are calculated by subtracting the mean of the pairwise comparisons from the read count of each condition and then dividing the difference by the standard deviation for the gene.

$$Z_{i,j} \frac{read\ count(gene\ i,\ condition\ j) - mean(gene\ i)}{standard\ deviation(gene\ i)}$$

where $z_{i,j}$ is the z-score for gene $i$ in condition $j$.

To identify genes that interact with GPCRs, an up-to-date list of GPCRs was obtained from UniProtKB (all entries indicated as olfactory, odorant, taste or vomeronasal receptors were removed) and fed into STRING DB (https://string-db.org) as multiple protein entries under *Mus musculus* along with gene entries from the heatmap in *Figure 4B*. Genes of secreted proteins with >0.9 combined STRING score for interactions with GPCRs are highlighted in red in *Figure 4B*.

## Mass spectrometry analysis

Peptides (5 μl) were separated by reverse-phase chromatography on a C18 fused silica column (I.D. 75 μm, O.D. 360 μm × 25 cm length) packed into an emitter tip (IonOpticks, Melbourne, VIC, Australia), using a nano-flow HPLC (M-class, Waters, UK). The HPLC was coupled to an Impact II UHR-QqTOF mass spectrometer (Bruker, Bremen, Germany) using a CaptiveSpray source and nanoBooster at 0.20 Bar using acetonitrile. Peptides were loaded directly onto the column at a constant flow rate of 400 nl/min with buffer A (99.9% Milli-Q water, 0.1% FA) and eluted with a 90 min linear

gradient from 2% to 34% buffer B (99.9% acetonitrile, 0.1% FA). MS spectra were acquired in a data-dependent manner including an automatic switch between MS and MS/MS scans using a 1.5 s duty cycle and 4 Hz MS1 spectra rate followed by MS/MS scans at 8–20 Hz dependent on precursor intensity for the remainder of the cycle. MS spectra were acquired between a mass range of 200–2000 m/z. Peptide fragmentation was performed using collision-induced dissociation (CID).

For data analysis, raw files consisting of high-resolution MS/MS spectra were processed with Max-Quant (version 1.5.8.3) for feature detection and protein identification using the Andromeda search engine. Extracted peak lists were searched against the *Mus musculus* database (UniProt, October 2016), as well as a separate reverse decoy database to empirically assess the false discovery rate (FDR) using strict trypsin specificity allowing up to two missed cleavages. The minimum required peptide length was set to seven amino acids. In the main search, precursor mass tolerance was 0.006 Da and fragment mass tolerance was 40 ppm. The search included variable modifications of oxidation (methionine), amino-terminal acetylation, the addition of pyroglutamate (at N-termini of glutamate and glutamine) and a fixed modification of carbamidomethyl (cysteine). The 'match between runs' option in MaxQuant was used to transfer identifications made between runs on the basis of matching precursors with high mass accuracy. Peptide-spectrum match (PSM) scores and protein identifications were filtered using a target-decoy approach at a false discovery rate (FDR) of 1%.

Only unique and razor peptides were considered for quantification with intensity values present in at least two out of three replicates per group. Statistical analyses were performed using LFQAnalyst (https://bioinformatics.erc.monash.edu/apps/LFQ-Analyst/), whereby the LFQ intensity values were used for protein quantification. Missing values were replaced by values drawn from a normal distribution of 1.8 standard deviations and a width of 0.3 for each sample (Perseus-type). Protein-wise linear models combined with empirical Bayes statistics were used for differential expression analysis using Bioconductor package Limma whereby the adjusted p-value cutoff was set at 0.05 and $\log_2$ fold change cutoff set at 1. The Benjamini-Hochberg method of FDR correction was used.

## Statistical analysis

Statistical analyses were performed using Prism 8.0 (GraphPad Software, La Jolla, CA, USA) or R (The R Project for Statistical Computing). D'Agostino and Pearson normality tests or Kolmogorov-Smirnov test were used to determine whether or not the data follow a Gaussian distribution. For data with a non-Gaussian distribution, Mann-Whitney U tests were used to compare medians between two groups and Kruskal-Wallis tests to compare medians between more than two groups followed by Dunn's multiple comparison tests. For transwell experiments, statistical significance between means of two groups was determined by performing two-tailed, unpaired Student's $t$ tests, and multiple means were compared with one-way ANOVA and Tukey's multiple comparisons test.

For the FMI, we used a two-tailed Wilcoxon signed rank test to compare medians with a hypothetical median value of 0. n = 3 independent experiments for each condition, 3 fields of view per condition per experiment. Box-plots (as depicted in *Figure 2B*): the box represents the 25th to 75th percentiles of the data points. The interquartile range (IQR) is the difference between the 25th and 75th percentiles. The upper whisker indicates the 75th percentile plus 1.5 times the IQR and the lower whisker indicates the 25th percentile minus 1.5 times the IQR. The data points above or below the whiskers are outliers. Mean is indicated by thick red line; median by thin black line. Coloured circles represent the means of individual experiments. For each condition, cell numbers are indicated underneath the plot. Coloured 'violin' plots for infiltration depth into tumouroid depict the distribution probability density; black bar represents the median, grey dots are data points. Mann Whitney tests were performed to compared medians of violin plots. In statistical analysis, p>0.05 is indicated as not significant (ns), whereas statistically significant values are reported in the figures.

Instantaneous FMI are calculated from cellular FMI values at each timeframe. Surfaces fitted and plotted through generalised additive models (such as in *Figure 3E*) were generated through custom code. Instantaneous FMI values were extracted from spatiotemporal positional data exported from Imaris using the 'motility_analysis' python package, available at https://github.com/marknorman-read/TcellSwarming. Generalised additive models were then fitted using the 'mgcv' package; code also available at the aforementioned repository.

For statistical and biological robustness, each experiment was performed at least three times with cells from different mice, except when stated otherwise.

## Acknowledgements

We thank PL Newman, A Masedunskas, J Chou, Y Wang, the BioMedical Imaging Facility and Biological Resources Imaging Laboratory of UNSW for technical assistance, and R Germain, N Plachta, M Coelho, J Rossy, A Yap and J Howard for helpful discussions. MB acknowledges Bitplane AG for an Imaris Developer license. This work was supported by NSERC grants RGPIN 50503–10477 and 50503–10476 to GR, NHMRC Fellowship GNT1135687 to DRN, the University of Sydney Centre for Advanced Food Enginomics to MNR and EMBL Australia to MB.

## Additional information

### Funding

| Funder | Grant reference number | Author |
| --- | --- | --- |
| NSERC | RGPIN 50503-10477 | Gregory Rice |
| National Health and Medical Research Council | GNT1135687 | David R Nisbet |
| University of Sydney | Centre for Advanced Food Enginomics | Mark N Read |
| EMBL Australia | | Maté Biro |
| NSERC | 50503-10476 | Gregory Rice |

The funders had no role in study design, data collection and interpretation, or the decision to submit the work for publication.

### Author contributions

Jorge Luis Galeano Niño, Conceptualization, Formal analysis, Investigation, Methodology; Sophie V Pageon, Formal analysis, Investigation, Writing - original draft; Szun S Tay, Formal analysis, Investigation, Methodology, Writing - review and editing; Feyza Colakoglu, Validation, Investigation; Daryan Kempe, Formal analysis, Validation, Investigation, Methodology; Jack Hywood, Software, Formal analysis; Jessica K Mazalo, James Cremasco, Matt A Govendir, Formal analysis, Investigation; Laura F Dagley, Data curation, Formal analysis; Kenneth Hsu, Resources, Methodology; Simone Rizzetto, Jerzy Zieba, Formal analysis; Gregory Rice, Formal analysis, Methodology; Victoria Prior, Geraldine M O'Neill, Richard J Williams, Resources; David R Nisbet, Belinda Kramer, Resources, Supervision; Andrew I Webb, Fabio Luciani, Formal analysis, Supervision; Mark N Read, Software, Formal analysis, Supervision, Investigation, Visualization; Maté Biro, Conceptualization, Formal analysis, Supervision, Funding acquisition, Investigation, Methodology, Writing - original draft, Project administration, Writing - review and editing

### Author ORCIDs

Sophie V Pageon http://orcid.org/0000-0003-1701-5551
Szun S Tay http://orcid.org/0000-0003-0186-8154
Laura F Dagley http://orcid.org/0000-0003-4171-3712
Simone Rizzetto http://orcid.org/0000-0003-3881-8759
Victoria Prior http://orcid.org/0000-0002-2285-5398
Maté Biro https://orcid.org/0000-0001-5852-3726

### Ethics

Human subjects: Human peripheral blood mononuclear cells (PBMCs) were obtained from healthy donors after informed consent and were used in experiments under a Human Research Ethics Committee (HREC) approved protocol (Sydney Children's Hospitals Network, LNR/13/SCHN/241).

Animal experimentation: All animal breeding and experimentation were conducted in accordance with New South Wales state and Australian federal laws and animal ethics protocols overseen and approved by the University of New South Wales Animal Care and Ethics Committee (ACEC) under protocols 16/83B and 19/133B.

Decision letter and Author response
Decision letter https://doi.org/10.7554/eLife.56554.sa1
Author response https://doi.org/10.7554/eLife.56554.sa2

## Additional files

### Supplementary files

• Supplementary file 1. Transcriptomics. Transcriptomics data of CTLs alone, in indicated conjugations or exposed to indicated supernatants from conjugations with tumour cells. Values are read counts obtained from featureCounts after alignment with TopHat2 to the GRCm38 reference genome.

• Supplementary file 2. Secretomics. Tables detailing secreted proteins identified by quantitative mass spectrometry analysis, including the proteins exhibiting significant differences between the cognate versus non-cognate cells and beads.

• Transparent reporting form

### Data availability

All data generated or analysed during this study are included in the manuscript and supporting files. Source data files with extensive statistical information have been provided for all figures containing bar, box or violin plots. Complete transcriptomics and secretomics data are available in Supplementary Files 1 and 2 respectively. Custom code and notes are available at https://github.com/marknormanread/TcellSwarming copy archived at https://archive.softwareheritage.org/swh:1:rev:74c6678c55317a0aac98a70939e0c92fb29e58ad/.

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
