## [Decision Letter]

Thank you for submitting your article "Cytotoxic T cells swarm by homotypic chemokine signalling" for consideration by *eLife*. Your article has been reviewed by Satyajit Rath as the Senior Editor, a Reviewing Editor, and two reviewers. The reviewers have opted to remain anonymous.

The reviewers have discussed the reviews with one another and the Reviewing Editor has drafted this decision to help you prepare a revised submission.

Summary:

In this manuscript, the authors employ an elegant mouse tumor organoid model to study the migration dynamics of cytotoxic T lymphocytes (CTLs) in controllable 3D environments. The authors provide strong data that upon engaging cognate antigen on tumour cells or artificial APCs, CTLs produce CCL3/4, which promote directed recruitment of distant CTLs via CCR5 in a positive feedback loop. These core findings are reproduced using human T cells and engineered tumour cells and also with a CAR-T cell model. The in vivo data show observations consistent with these interpretations. Based on these findings, the authors suggest that homotypic, long-range signalling could lead to a CTL swarming response and enhance tumour killing in vivo.

While the manuscript is of great interest, there are concerns that require to be addressed as detailed below.

Essential revisions:

1) Issues related to the in vitro work:

1A) A major argument of this manuscript is that early "searcher" CTLs that arrive at the tumour sites are responsible for the sustained, feed-forward amplification of further recruitment of the later "responder" CTLs in a multi-step process (Figure 7). This hypothesis is not adequately supported by the tumour organoid experiment in which the authors have pre-embedded CTLs with the tumour cells in a 1:1 ratio and showed that CTLs were positively attracted to the tumour site when pre-embedded antigen-specific CTLs are present (Figure 2). At 2x10^6 CTLs and tumour cells each, the starting experiment condition implies that a fairly large number of CTLs have already been recruited into the tumour site. This is ~7-fold more CTLs than the 'dispersed' CTLs (3x10^5) that were seeded outside of the tumour site for their migration to be tracked. To demonstrate that early arriving CTLs can induce and drive a cascade of CTL self-recruitment, the authors should titrate the pre-embedded CTLs to much lower numbers and examine if biased directionality in adjacent CTLs can still be observed in this case.

1B) The authors observe enhanced infiltration of CTLs when a separate CTL population is pre-embedded in the tumour organoid carrying cognate antigen. They use this observation as evidence of enhanced directed recruitment of distant CTLs. However, pre-embedding CTLs should also enhance killing of the tumour cells and alleviate physical space constraints within the tumouroid itself (based on the increased number of "black" holes in the tumoroid observed in Video 2). How do the authors distinguish between increased CTL infiltration due to enhanced directed recruitment versus more physical space available in the tumouroid due to killing? This "physical constraint" issue could be addressed by comparing infiltration of a non-embedded tumouroid vs. CTL-embedded cognate tumouroid by CCR5- effector T cells of a different specificity. This would be a simple extension of the study shown in Figure 2 using two different specificities of CD8^+^ T cells, but with infiltration in a central tumouroid now examined rather than the FMI alone in an asymmetric model.

1C) The data suggest that CCL3 and/or CCL4 gradients form; do the authors have experimental evidence for this?

1D) 6- Are both CCL3 and CCL4 needed? Or are individual chemokines enough?

1E) In Figure 2B, cell speed also increases when tumours are preloaded with CTLs in the presence of non-cognate antigens. The likely explanations need to be explained.

1F) Why are the transcriptomic profiles of CTLs embedded with antigen-loaded tumor cells or antigen-loaded beads different (Figure 4B)?

1G) 7- If CTLs and neutrophils use different mechanisms to be recruited to swarm, why are their recruitments mutually exclusive (Figure 4 and Figure 4—figure supplement 2)?

1H) Methodological clarifications: the authors should clearly describe (a) what 'cognate' and 'non-cognate' antigens are (in addition to Materials and methods), and (b) how beads are co-embedded with CTLs (Figure 2C legend).

1I) The results need to be discussed in light of Hugues et al., (2007), where the authors have shown that in lymph nodes, the antigen-specific CD8 T cells that interact with DCs attract CD8 T cells of other antigen specificities in a CCR5-dependent manner. This important paper is not cited or discussed and clearly provides the initial report of a CD8 T cell capacity to induce recruitment of other CD8 T cells. The present manuscript expands on this core observation and goes further to provide data on the capacity of peripheral effector T cells to drive such a circuit.

2) Issues related to the in vivo work:

2A) Unfortunately, the authors do not provide direct and specific data on the role of this mechanism in tumour rejection in vivo and they understate the existing evidence that homotypic signalling occurs among CD8^+^ T cells. There is prior evidence, from dynamic imaging work, of T cell production of chemokines that leads to attraction of other T cells to facilitate immunity. The new aspect proposed here is the operation of such circuits in the peripheral tumour microenvironment rather than in secondary lymphoid organs. The organoid data are somewhat limited with respect to tumour relevance, as made clear by the ability of the authors to get the same in vitro data using antigen-bearing beads. The in vivo experiments do not completely address the point of the manuscript with respect to the plausible role of a homotypic CCL3/4-CCR5 signalling pathway in CD8^+^ T cell mediated tumor infiltration and killing. This is because the experiments in vivo are all with tumor cells engineered to express CCL3/4, not with non-engineered tumor cells and T cells with an inability to make the relevant chemokines, to express the relevant receptor, or where these interactions are inhibited by some of the drugs used in vitro in the paper. Transfer studies done in mice lacking their own cognate CD8^+^ T cells response to a tumor expressing the antigen relevant to the transferred CD8^+^ T cells (e.g., in RAG KO) would be the easiest approach to addressing this question, as the transferred cells can be made to lack CCR5 or CCL3/4 expression or the animals treated with inhibitors of this pathway as done in vitro. Without such experiments, the authors provide no functional data to demonstrate that CTL engagement of a tumour cell expressing cognate antigen drives recruitment of distant CTLs in vivo that in turn contributes to tumor growth inhibition or destruction.

Similarly, the authors have engrafted EL4 or CCL3/4-secreting EL4 on Rag KO mice to show that CTL recruitment was enhanced in CCL3/4-secreting tumours and have argued that these "sustained production of CCL3 and CCL4 by activated CTLs is sufficient to induce swarming and tumor infiltration." However, the experimental result here does not support the claim that CTL tumor infiltration is mediated by direct, long-range CCL3/4 chemotactic sensing by CD8 T cells nor that this increased recruitment is caused by CTL-secreted CCL3/4, as other cell types, including myeloid cells (such as pDCs, NK cells, some of which the authors also show in Figure 4—figure supplement 2J) can also respond to CCL3/4 signals and drive further inflammation and chemotactic factors that amplify CTL recruitment indirectly. The critical control of CCR5- null CD8bT cells is not included in this set of experiments, only in subsequent studies.

Further, in Figure 5E-F, the authors performed a dual flank experiment where CCL3/4-secreting and non-secreting EL4 tumor cells were engrafted on contralateral flanks and transferred WT and CCR5 KO CTLs and argue that CCR5 is required for CTL recruitment. However, it is possible to interpret these data, of an almost equal ratio of WT:CCR5 KO CTLs in the non-secreting tumor, to suggest that CCR5-mediated recruitment is not the primary mechanism that results in CTL tumor infiltration.

A major claim of the paper, and a new advance if it were to be properly demonstrated, is that this circuit plays a relevant and important role in tumors AFTER the action of such circuits in secondary lymphoid organs. However, additional data, as pointed out above, are needed to make this point. This is especially important as several recent papers have shown a key role for the other chemokine-chemokine receptors circuits such as CXCL9/10 – CXCR3 in mediating effective CD8 T cells immunity in tumors (for example, Chow et al., 2019 andCD8^+^ Chheda et al., 2016), so whether different sets of CK and CR contribute in non-reductant ways to anti-tumor immunity is unknown. Given that these other data cited involve in vivo tumor experiments, one needs comparable in vivo studies to place the present findings with CCL3/4 in context.

2B) Given the observation of peripheral rimming of T cells in the in vitro studies in the absence of CCL3/4 and invasion with these mediators operative, transcriptomic datasets from human samples that are classified as cold (excluded) and hot (infiltrated) would be useful to probe to see if there is a difference in CCL3/4 mRNA expression in the two cases, and would be even better if this involved single cell work that ascribed production to the CD8 T cells themselves. This would be in silico work to see if the right data sets exist and if mined, show the expected outcome, although it is possible, of course, that no such differences may be identifiable in CCL3/4 expression between hot and cold tumors, which would neither invalidate nor support a potential role played by these chemokines in CD8 T cell swarming in tumours.

2C) In the absence of data from additional work as in 2A above, and/or if efforts in 2B above do not show any informative correlation, it is recommended that the paper be re-written to focus on effector CD8 T cells driving concerted recruitment to sites of antigen recognition, with the underlying mechanism carefully documented, and with only the discussion suggesting that this might be a mechanism active in the in vivo tumor setting, making clear that future experiments will show whether this mechanism is indeed prominent in primary tumors in vivo.

[Editors' note: further revisions were suggested prior to acceptance, as described below.]

Thank you for submitting your article "Cytotoxic T Cells Swarm by Homotypic Chemokine Signalling" for consideration by *eLife*. Your article has been reviewed by Satyajit Rath as the Senior Editor, a Reviewing Editor, and two reviewers. The following individuals involved in review of your submission have agreed to reveal their identity: Ana-Maria Lennon-Duménil (Reviewer #2).

The reviewers have discussed the reviews with one another and the Reviewing Editor has drafted this decision to help you prepare a revised submission.

We would like to draw your attention to changes in our revision policy that we have made in response to COVID-19 (https://elifesciences.org/articles/57162). Specifically, when editors judge that a submitted work as a whole belongs in *eLife* but that some conclusions require a modest amount of additional new data, as they likely do with your paper, we are asking that the manuscript be revised to either limit claims to those supported by data in hand, or to explicitly state that the relevant conclusions require additional supporting data.

Our expectation is that the authors will eventually carry out the additional experiments advisable and report on how they affect the relevant conclusions either in a preprint on bioRxiv or medRxiv, or if appropriate, as a Research Advance in *eLife*, either of which would be linked to the original paper.

Summary:

The revised manuscript addresses most concerns of the reviewers and is thus of great interest. There remain some concerns to be addressed as detailed below.

Essential revisions:

1) The authors need to clarify the results in Figure 2B (right panel) and Figure 2F. In Figure 2B, the exogenously added CTLs exhibit swarming (increased FMI) towards the cognate tumour even in the absence of pre-embedded antigen-specific CTLs. However, in Figure 2F, exogenously added CTLs do not exhibit swarming unless there are pre-embedded antigen-specific CTLs. What is the cause of this discrepancy?

2) Subsection “CTL recruitment is driven by secretion of CCL3 and CCL4” – Clarification is required. The authors state "only CCL3 and CCL4 inhibition disrupted CTL transmigration towards cognate supernatants." However, in Figure 4—figure supplement 1F, CCL4 blocking Abs alone have no effect on transmigration, despite that fact that both CCL3 and CCL4 blocking antibodies are required to block directional migration in Figure 4G. Can the authors provide an explanation for this discrepancy in the text?

3) Figure 5F – OT1 WT and OT1 CCR5-/- both exhibit a large increase in recruitment to the tumour when gBT1 cells are pre-transferred. This observation suggests that CCL3 and CCL4 actually play a very minor role in T cell "swarming" in this in vivo system. The authors should alter the wording of the abstract since it currently states that "CTLs engaging cognate targets accelerate the recruitment of distant T cells through long-range homotypic signalling via the diffusion of secreted chemokines CCL3 and CCL4". This statement suggests that CCL3/CCL4 play a major role in T cell swarming, but this claim isn't fully supported by the in vivo data.

4) Figure 5 and accompanying supplements – The small differences observed between WT and CCR5-/- CTLs in vivo, both in terms of recruitment and tumour killing, could be due to the fact that the authors co-transferred 5x106 of each CTL into recipient mice. This high transfer number may diminish the role of CCR5/CCL3/CCL4 mediated recruitment of CTLs into the tumour in vivo. At the very least, the authors should comment on this issue in the text.

5) Figure 5—figure supplement 2 – This figure would be even more convincing if the authors showed the distribution of the WT and CCR5-/- CTLs in a cognate tumour without an exogenous source of CCL3.

6) Subsection “CTL recruitment is driven by secretion of CCL3 and CCL4” – Figure 4—figure supplement 4A-C. Overexpression of CCL3/4 in the tumors seems to have no effect (at least statistically) on the recruitment of OT1 CTLs or endogenous CTLS into the tumor in vivo. These observations seem to contradict the main point of the paper. The authors should provide a convincing explanation.

Reviewer #2:

The authors have properly addressed all my concerns and provide an important number of new experiments. To my opinion, this manuscript is now acceptable for publication in *eLife*.

Reviewer #4:

The authors have addressed most of the in vitro concerns (although there a few points that still require clarification) and performed a series of new in vivo experiments. However, the results from these new in vivo experiments are difficult to interpret and, in some instances, seem to diminish the in vitro conclusions. Further clarification is required – this may be possible with text changes, but it does appear that there are issues with the in vivo experiments that could require more bench work.

- The authors need to clarify the results in Figure 2B (right panel) and Figure 2F. In Figure 2B, the exogenously added CTLs exhibit swarming (increased FMI) towards the cognate tumour even in the absence of pre-embedded antigen-specific CTLs. However, in Figure 2F, exogenously added CTLs do not exhibit swarming unless there are pre-embedded antigen-specific CTLs. What is the cause of this discrepancy?

- Subsection “CTL recruitment is driven by secretion of CCL3 and CCL4” – Clarification is required. The authors state "only CCL3 and CCL4 inhibition disrupted CTL transmigration towards cognate supernatants." However, in Figure 4—figure supplement 1F, CCL4 blocking Abs alone have no effect on transmigration, despite that fact that both CCL3 and CCL4 blocking antibodies are required to block directional migration in Figure 4G. Can the authors provide an explanation for this discrepancy in the text?

- Figure 5F – OT1 WT and OT1 CCR5-/- both exhibit a large increase in recruitment to the tumour when gBT1 cells are pre-transferred. This observation suggests that CCL3 and CCL4 actually play a very minor role in T cell "swarming" in this in vivo system. The authors should alter the wording of the abstract since it currently states that "CTLs engaging cognate targets accelerate the recruitment of distant T cells through long-range homotypic signalling via the diffusion of secreted chemokines CCL3 and CCL4". This statement suggests that CCL3/CCL4 play a major role in T cell swarming, but this claim isn't fully supported by the in vivo data.

- Figure 5 and accompanying supplements – The small differences observed between WT and CCR5-/- CTLs in vivo, both in terms of recruitment and tumour killing, could be due to the fact that the authors co-transferred 5x106 of each CTL into recipient mice. This high transfer number may diminish the role of CCR5/CCL3/CCL4 mediated recruitment of CTLs into the tumour in vivo. At the very least, the authors should comment on this issue in the text.

- Figure 5—figure supplement 2 – This figure would be even more convincing if the authors showed the distribution of the WT and CCR5-/- CTLs in a cognate tumour without an exogenous source of CCL3

- Subsection “CTL recruitment is driven by secretion of CCL3 and CCL4” – Figure 4—figure supplement 4A-C. Overexpression of CCL3/4 in the tumors seems to have no effect (at least statistically) on the recruitment of OT1 CTLs or endogenous CTLS into the tumor in vivo. These observations seem to contradict the main point of the paper. The authors should provide an explanation in the text.

---

## [Author Response]

Essential revisions:1) Issues related to the in vitro work:1A) A major argument of this manuscript is that early "searcher" CTLs that arrive at the tumour sites are responsible for the sustained, feed-forward amplification of further recruitment of the later "responder" CTLs in a multi-step process (Figure 7). This hypothesis is not adequately supported by the tumour organoid experiment in which the authors have pre-embedded CTLs with the tumour cells in a 1:1 ratio and showed that CTLs were positively attracted to the tumour site when pre-embedded antigen-specific CTLs are present (Figure 2). At 2x10^6 CTLs and tumour cells each, the starting experiment condition implies that a fairly large number of CTLs have already been recruited into the tumour site. This is ~7-fold more CTLs than the 'dispersed' CTLs (3x10^5) that were seeded outside of the tumour site for their migration to be tracked. To demonstrate that early arriving CTLs can induce and drive a cascade of CTL self-recruitment, the authors should titrate the pre-embedded CTLs to much lower numbers and examine if biased directionality in adjacent CTLs can still be observed in this case.

We thank the reviewers for raising this critical point, which indeed is central to our model. First, a point of clarification: in the experiments of Figure 2, the CTLs embedded within the tumouroid and those that are dispersed adjacent to the tumouroid are differentially labelled, and thus only the latter are tracked in terms of speed and directionality towards the tumouroid. Any CTLs pre-embedded in the tumouroids therefore do not contribute to the CTL migration data shown in the figure.

We have followed the reviewers’ suggestion and now include a titration of the number of CTLs embedded in the cognate tumouroids, whilst keeping the tracked CTL population adjacent to the tumouroid constant. New Figure 2B shows motility data for cases where CTLs are pre-embedded in the cognate tumouroid at a 1:1 ratio with tumour cells, at a 1:5 ratio and in the complete absence of pre-embedded CTLs.

In all 3 cases, even in the absence of CTLs pre-embedded in the cognate tumouroid, the dispersed CTLs display enhanced directionality towards the tumouroid (FMI > 0). In the context of non-cognate tumouroids, whether CTLs were pre-embedded or not, the surrounding CTLs do not display directional motion. These results demonstrate that CTLs randomly arriving at a cognate target in the absence of a chemotactic signal (‘searcher’ CTLs), engage the target in an antigen-specific manner (turning into ‘engager/recruiters’) and induce directional motility in distant CTL populations (which become ‘responders’) as illustrated in Figure 7.

1B) The authors observe enhanced infiltration of CTLs when a separate CTL population is pre-embedded in the tumour organoid carrying cognate antigen. They use this observation as evidence of enhanced directed recruitment of distant CTLs. However, pre-embedding CTLs should also enhance killing of the tumour cells and alleviate physical space constraints within the tumouroid itself (based on the increased number of "black" holes in the tumoroid observed in Video 2). How do the authors distinguish between increased CTL infiltration due to enhanced directed recruitment versus more physical space available in the tumouroid due to killing? This "physical constraint" issue could be addressed by comparing infiltration of a non-embedded tumouroid vs. CTL-embedded cognate tumouroid by CCR5- effector T cells of a different specificity. This would be a simple extension of the study shown in Figure 2 using two different specificities of CD8^+^ T cells, but with infiltration in a central tumouroid now examined rather than the FMI alone in an asymmetric model.

We thank the reviewers for the excellent suggestion to extend our central tumouroid data to control for infiltration not merely arising due to additional space created by the elimination of tumour cells by pre-embedded CTLs. We have implemented the suggested experiments by replacing tumour cells with antigen-coated beads that cannot be lysed (as previously done in the asymmetric model depicted in Figure 2C) and therefore cannot progressively give way to more physical space for infiltrating CTLs. As suggested, we tested the following conditions:

- Cognate beads: a central bead mass coated with cognate antigen (SIINFEKL), without pre-embedded CTLs and surrounded by OT1 CTLs.

- Cognate beads + CTLs: a central bead mass coated with cognate antigen (SIINFEKL), pre-embedded with CCR5^-/-^ OT1 CTLs and surrounded by gBT1 CTLs (non-cognate for the SIINFEKL antigen).

and the following control:

- Non-cognate beads: a central mass with uncoated beads, surrounded by OT1 CTLs.

The cognate bead scenarios are illustrated in Figure 1—figure supplement 1A.

Similarly to tumouroids composed of real tumour cells and pre-embedded tumour-reactive CTLs, in the condition where the bead mass contains pre-embedded cognate CTLs, the surrounding CTLs infiltrate the mass more than when the cognate bead mass does not contain pre-embedded CTLs (Figure 1—figure supplement 1B-E). Thus, this increased infiltration occurs independently of any alleviation of space constraints. Interestingly, in cognate contexts the infiltration depth is greater in the case of the pre-embedded bead-mass than real cell tumouroid (compare to Figure 1D); this is due to the lack of antigen specificity of the surrounding gBT1 CTLs, which therefore do not arrest to engage the beads coated with antigen specific for the pre-embedded OT1 CTLs. The absence of this antigen-specific arrest is most evident in the lack of accumulation at the edge of the pre-embedded bead mass (see confocal images and density kymographs of Figure 1—figure supplement 1D and E). Together, these results demonstrate that antigen-specific activation of CTLs deep within a solid mass induces enhanced directed recruitment of distal CTLs (that do not need to be cognate for the target), which then also infiltrate the target mass to a greater extent.

1C) The data suggest that CCL3 and/or CCL4 gradients form; do the authors have experimental evidence for this?

This helpful question prompted us to devise an experiment in which the chemokine gradient formed by CTLs engaging cognate targets can be visualised and quantified. CCL3 and CCL4 chemokines are small molecules (~10kDa) that rapidly diffuse in aqueous mediums and cannot be readily visualised in solution. We therefore employed cytometric bead array (CBA) particles (beads), which are normally used in solution and subjected to flow cytometry to measure soluble chemokine and cytokine concentrations in supernatants (see for example Figure 4—figure supplement 1C). Here, we embedded CBA beads in a 3D collagen matrix surrounding a cognate tumouroid to immobilise CCL3 diffusing from the tumouroid. Following fixation, the addition of an anti-CCL3 antibody allowed for the imaging of the extent of diffusion of CCL3 away from the tumouroid, as shown in Figure 4—figure supplement 2D and 2E. We obtained such samples at 2 and 3 hours following embedding of the tumour-reactive CTLs in the tumouroid. Simple image analysis, consisting of a maximum intensity z-projection followed by quantification of mean intensities along the x-axis (which defines the distance from the tumouroid), revealed that the CBA beads were uniformly distributed in the matrix surrounding the tumouroid. By contrast, the anti-CCL3 signals displayed a clear gradient with higher concentrations near the tumouroid than at distant positions, and a steeper gradient and increased overall signal at 3hours compared to 2hours, consistent with the continual secretion and diffusion of CCL3 from CTLs in the tumouroid (Figure 4—figure supplement 2F). These results provide direct evidence for the establishment of chemokine gradients from the site of antigen-specific target engagement by CTLs, which in turn leads to the progressive recruitment of distant CTLs.

1D) 6- Are both CCL3 and CCL4 needed? Or are individual chemokines enough?

This question was partly addressed in the original submission but prompted us to expand our data. In Figure 4—figure supplement 1G we had used neutralising antibodies to selectively block CCL3 and CCL4 signalling, and in combination, in 3D collagen matrices containing a side (asymmetric) tumouroid. Those data showed that CCL3 and CCL4 act redundantly. Inhibition of CCL3 or CCL4 alone was insufficient to abolish the directional recruitment of distant CTLs, whereas concurrent inhibition of both CCL3 and CCL4 fully abolished the directional recruitment. Thus, both CCL3 and CCL4 are not needed simultaneously to underpin homotypic recruitment of CTLs. In order to conclusively address the second part of the question as to whether the diffusion of individual CCL3 or CCL4 chemokines are sufficient to generate swarming, we expanded our data to include tumouroids that exclusively secrete CCL4 (new Figure 4—figure supplement 2I-K), in addition to the double CCL3/4-secreting and CCL3-secreting contexts shown previously in Figure 4—figure supplement 2C-H. Untransduced EL4 tumour cells do not secrete any factors that induce directional recruitment of CTLs (see Figure 2B). EL4 tumouroids engineered to exclusively secrete CCL3 or CCL4 both give rise to CTL swarming. These experiments show that the diffusion of either CCL3 or CCL4 alone is sufficient to generate CTL swarming around a source. Importantly, swarming is observed in the case where the tumour cells are non-cognate, and thus in these experiments arriving CTLs cannot contribute to a chemotactic signal, which is exclusively composed of tumour-secreted CCL3 or CCL4.

1E) In Figure 2B, cell speed also increases when tumours are preloaded with CTLs in the presence of non-cognate antigens. The likely explanations need to be explained.

This is an interesting question that centres on the concept of chemokinesis, increased motility without a directional component, as opposed to chemotaxis, which involves directional motion towards a source of a gradient (Weninger, Biro and Jain, 2014); the former as well as the latter are often observed in response to cytokine or chemokine signalling (Worbs et al., 2007). It is highly interesting that CTLs do recognise non-cognate tumour cells and in response upregulate various genes, including some that are exclusively upregulated in the case of non-cognate contacts and not during cognate contacts (for full transcriptomic data see Supplementary file 1).

Amongst genes that are exclusively up in non-cognate contacts are netrin-1, semaphorin 5A, and autotaxin. Netrins are well-characterised proteins that provide axonal guidance cues in neurons, and promote general cell migration. Netrin-1 has indeed been shown to induce chemokinesis in CD4^+^ T cells in vitro and enhance in vivo inflammation (Boneschansker et al., 2016). The netrin-1 receptors neogenin, uncoordinated-5 (UNC5)A, and UNC5B that are upregulated by CD4^+^ T cells upon activation (Boneschansker, 2015) are also expressed by OT1 CTLs.

Similar to netrins, the action of semaphorins in axon growth and guidance extend beyond the brain to regulate migration in diverse cell types, including leukocytes (Takegahara, Kumanogoh and Kikutani, 2005). Sema5a drives angiogenesis, and is a potent activator of T cells and NK cells (Gras et al., 2014). Its receptor plexin3b is expressed by OT1 CTLs but it is not known whether Sema5A triggers chemokinesis in CTLs.

A third candidate that could drive chemokinesis is the ectoenzyme, ectonucleotide pyrophosphatase-phosphodiesterase 2 (ENPP2). Widely known as “autotaxin” (Ninou, Magkrioti and Aidinis, 2018), ENPP2 is a secreted lysophospholipase D largely responsible for extracellular lysophosphatidic acid (LPA) production. LPA is a bioactive phospholipid that has been shown to induce LFA-1– independent chemokinesis in T cells in vitro (Katakai et al., 2014). LPA signals through at least six type I rhodopsin-like receptors (LPARs) (Yung, Stoddard and Chun, 2014), three of which (LPAR 2,5 and 6) are expressed by OT1 CTLs. It will be interesting in future studies to identify the specific triggers of non-cognate contact-induced chemokinesis. Here, we simply discuss the interesting observation in the Discussion section.

1F) Why are the transcriptomic profiles of CTLs embedded with antigen-loaded tumor cells or antigen-loaded beads different (Figure 4B)?

This is an interesting question that principally addresses the molecular mechanisms, and structure and function of antigen-specific activation via T cell receptors, the dependence on co-receptors and the sufficiency of peptide-MHC in the absence of other surface signals to invoke complete T cell activation; as such it is beyond the scope of the present study.

It should however be noted that the beads are only coated with peptide-MHC and completely lack integrin ligands such as ICAM-1. Expression of the integrin LFA-1 by CTLs and its binding to ICAM1 on targets plays an important role in the arrest of CTLs upon target recognition, synapse formation and ultimately in target killing (Evans et al., 2011; Hogg, Patzak and Willenbrock, 2011). Furthermore, mounting evidence suggests that the mechanical landscape of target cells plays an important role in the degree of activation and efficiency of effector function delivery of T cells (Basu et al., 2016). Beads are evidently considerably stiffer than our target tumour cells.

Whether the transcriptomic differences we observe are due to co-receptor engagement, paracrine signalling through factors secreted by real tumour cells, by integrin or other surface receptor activations, or biophysical effects such adhesion strengths or mechanical properties of the targets, remains to be explored in future studies.

Whilst the question as to why there are differences in gene expression between bead- and tumour cell-conjugated CTLs is interesting per se, it should be noted that the transcriptomic profiles of these two samples are far more similar to one another than to resting CTLs or those engaging non-cognate tumour cells (as shown by the hierarchical clustering tree at the top of Figure 4B). Observed differences are mostly in the extent of expression, as most genes, with the exception of only a few, are either similarly up- or down-regulated in both the cognate bead and cognate tumour cell samples. Indeed, genes that code for proteins that are important in this study are well correlated in expression between the cognate bead and cognate tumour cell samples. We used beads here in conjunction with tumour cells precisely to identify effects that arise due to antigen-mediated activation of the CTLs and are therefore consistent between the two samples, and not to investigate the underlying molecular and structural mechanisms that govern TCR-activation.

1G) 7- If CTLs and neutrophils use different mechanisms to be recruited to swarm, why are their recruitments mutually exclusive (Figure 4 and Figure 4—figure supplement 2)?

We thank the reviewers for this question, which prompted us to further evaluate the concept of mutual exclusion between neutrophils and CTLs. It should be noted that we did not claim to have demonstrated that such a mutually exclusive recruitment mechanism is in force. We had found that upon increased CTL infiltration into tumours there was an associated decrease in infiltrating neutrophils as a percentage of CD45+ leukocytes, and simply reflected that this is “consistent with a recently suggested mutually exclusive recruitment mechanism between CTLs and neutrophils to the TME” et al.(Stoll et al., 2018). A more recent report also indicates an inverse correlation between CD4^+^ or CD8^+^ T cell tumour infiltration on the one hand, and neutrophils on the other (Jackstadt et al., 2019). However, in response to minor point 1 below where it was requested that we plot absolute counts of tumour-infiltrating cells rather than as percentages, we found there to no longer be a significant difference in neutrophil infiltration.

In any event, the reviewers are correct in that neutrophils and T cells use dissimilar mechanisms to achieve similar swarming phenomena. Swarming by immune cells was first described in neutrophils (Chtanova et al., 2008; Lammermann et al., 2013; Ng et al., 2011), which instead of chemokine signalling utilise lipid signalling via Leukotriene B4 (LTB4) and its corresponding receptor LTB4R1 to achieve homotypic signalling and swarming around targets. CCR5 is only minimally expressed by circulating neutrophils (Rudd et al., 2019) and it is therefore highly unlikely that T cell-secreted CCL3 and CCL4 can constitute a direct cross-signal that would influence neutrophils. Interestingly however, CTLs do express LTBR1 (transcriptomics data available in Supplementary file 1), which raises the possibility that neutrophils may attract T cells via LTB4. Such an asymmetric attraction would on its own invalidate the concept of mutual exclusion. We cannot exclude that activated tissue-infiltrating CTLs may still generate a signal that somehow directly influence neutrophils via factors other than CCL3 and CCL4. However, the reviewers’ question led us to realise that for the avoidance of confusion for the readers, it would be prudent not to suggest that such a mechanism may be at play here. We have therefore removed the comment and associated reference from the main text.

1H) Methodological clarifications: the authors should clearly describe (a) what 'cognate' and 'non-cognate' antigens are (in addition to Materials and methods), and (b) how beads are co-embedded with CTLs (Figure 2C legend).

a) Murine CTLs were generated from the OT1 and gBT1 T cell receptor transgenic mice that recognise ovalbumin (SIINFEKL) or herpes simplex virus glycoprotein B (SSIEFARL) residues respectively, both in the context of the H-2K^b^ class I major histocompatibility complex. For preparation of target cells to present cognate antigens for OT1 or gBT1 CTLs, EL4 tumour cells were pulsed for 16 hours with 1 µg/ml SIINFEKL or SSIEFARL peptide, respectively. Non-cognate controls were un-pulsed EL4, or EL4 cells pulsed with the irrelevant peptide. In the in vivo tumour homing and rejection experiments performed during revision, cognate antigen was presented by EL4 cells transduced to express SIINFEKL- or SSIEFARL-expressing proteins fused to mTagBFP2. Non-cognate controls included untransduced EL4, EL4 expressing mCherry, or EL4 expressing the irrelevant peptide epitope.

b) For stimulating OT1 CTLs with “cognate beads”, streptavidin-coated polystyrene particles (6-8 µm diameter) were bound to monobiotinylated H-2K^b^/SIINFEKL for 45min and washed before mixing with OT1 CTLs in liquid phase collagen at 4ºC, before the matrix was allowed to polymerise for at least 10 min at 37ºC. Non-cognate beads were uncoated. Where side (asymmetric) tumouroids were constructed, 20 µl of the cold collagen preparation containing 3 x 10^6^ each of cognate beads and CTLs were deposited on one side of a 14-mm microwell within a 35-mm glass-bottom dish. Where central tumoroids were constructed, 2.5 µl of the cold collagen preparation containing 3.5 x 10^5^ cognate or non-cognate particles, or 1.75 x 10^5^ each of cognate particles and CTLs were deposited in the centre of a well in a glass-bottom 96-well optical plate.

We have now provided these details in the main text and figure legends, as well as in Materials and methods section.

1I) The results need to be discussed in light of Hugues et al., (2007), where the authors have shown that in lymph nodes, the antigen-specific CD8 T cells that interact with DCs attract CD8 T cells of other antigen specificities in a CCR5-dependent manner. This important paper is not cited or discussed and clearly provides the initial report of a CD8 T cell capacity to induce recruitment of other CD8 T cells. The present manuscript expands on this core observation and goes further to provide data on the capacity of peripheral effector T cells to drive such a circuit.

We thank the reviewer for highlighting the relevant publication by Hugues et al., 2007et al. that shows that naive CD8^+^ T cells engaging cognate antigen presented by dendritic cells within lymph nodes provide “help” to recruit polyclonal CD8^+^ T cells of other antigen specificities. We had only cited the publication by Castellino et al., 2006 that shows CD4^+^ T cells guide CD8^+^ T cells to sites of CD4^+^ T cell-dendritic cell interactions. In the Discussion section, we have now referenced the Hugues et al., study, together with additional studies that implicate CCR5 and its ligands in the directional migration of T cells within tumour-draining lymph nodes (Gonzalez-Martin et al., 2011) and infected lymph nodes (Hickman et al., 2011). In these studies, naïve CD8^+^ T cell recruitment within lymph nodes to sites of antigen presentation was CCR5-dependent, but such an effect had not been shown in peripheral effector CTLs. In addition, the sources of the CCR5 ligands were not unambiguously ascribed to the T cells, but merely to interactions between DCs and T cells.

2) Issues related to the in vivo work:2A) Unfortunately, the authors do not provide direct and specific data on the role of this mechanism in tumour rejection in vivo and they understate the existing evidence that homotypic signalling occurs among CD8^+^ T cells. […] Given that these other data cited involve in vivo tumor experiments, one needs comparable in vivo studies to place the present findings with CCL3/4 in context.

We agree with the reviewers that there have been prior reports of homotypic signalling leading to attraction of distant naïve CD8^+^ T cells in lymph nodes, to sites of interactions between dendritic cells and CD4^+^ T cells (Castellino et al., 2006) or naïve CD8^+^ T cells (Hugues et al., 2007), and have discussed these references in our manuscript as per point 1I above. We also agree that investigating CTL homing towards non-secreting tumours containing activated CTLs would more directly address if homotypic signalling drives recruitment of distant CTLs in vivo. As suggested by the reviewers, we performed adoptive transfer experiments in Rag1-deficient (*Rag^-/-^RAG^-/-^*) mice (that lack endogenous T- and B cells) bearing non-secreting EL4 tumours.

*Rag^-/-^RAG^-/-^* mice engrafted with SSIEFARL-expressing tumours received a primary transfer of tumour-reactive gBT1 CTLs (cognate). 48 h later a secondary cohort of WT and CCR5^-/-^ OT1 CTLs (both non-cognate) were co-transferred (Figure 5E). The presence of tumour-reactive CTLs (Figure 5—figure supplement 3 E,F) markedly increased the overall recruitment of subsequent non-cognate CTLs into the tumours (Figure 5F). Furthermore, subsequent to a primary transfer of cognate CTLs, significantly more non-cognate WT CTLs infiltrated the tumours than the co-transferred CCR5^-/-^ CTLs (p=0.015). In the absence of a primary transfer of cognate CTLs, there was no significant difference in the ability of WT and CCR5^-/-^ CTLs to infiltrate the tumours (p=0.394), in line with what we had previously found in the control tumours engrafted on the contralateral flanks of CCL3/4-secreting tumours (previously Figure 5E,F; now Figure 5—figure supplement 3A,B).

It should be noted that CCR5-/- CTLs also home in greater numbers to tumours containing previously transferred tumour-reactive CTLs. Therefore, circuits other than the homotypic CCL3/4-CCR5 signalling axis we identify here must also be at play. Whether these are indirect circuits arising in inflamed tumours, mediated for instance via NK and dendritic cells, or more direct circuits driven by alternate chemokine receptor-ligand interactions or pro-inflammatory cytokines remains to be established (see also below for response regarding the CXCR3 circuit). For instance, CCL3 or CCL4-producing tumours recruit NK cells, where they directly contribute to IFNγ, CCL5 and XCL1 production (Allen et al., 2018; Bottcher and Reis e Sousa, 2018), and recruit DC intermediaries that amplify CXCL9, CXCL10 and CCL5 production (Allen et al., 2018; Bottcher et al., 2018; Spranger, Bao and Gajewski, 2015; Wong et al., 2013), in turn facilitating the accumulation of effector T cells. Nevertheless, our results show that CTLs engaging a cognate tumour can drive the recruitment of distant CTLs into the tumour, and that this recruitment is partly dependent on CCR5.

In light of the reviewers’ comments on the relevance of the in vivo experiments with CCL3/4-secreting tumours, we have moved the corresponding data previously shown in Figure 5E-H to Figure 5—figure supplement 3A-D, and instead include the above new findings in Figure 5E, F.

As suggested by the reviewers, we also investigated tumour clearance in *Rag^-/-^RAG^-/-^* mice. We found that SIINFEKL-expressing tumour clearance by adoptively transferred cognate CCR5^-/-^ OT1 CTLs was not impaired compared to WT OT1 CTLs (in separate hosts) (Figure 5—figure supplement 3G). Therefore, whilst CTL engagement of tumours expressing cognate antigen may increase recruitment of distant CTLs, CCR5 expression on transferred CTLs was not critical for tumour clearance. This finding may not be completely unexpected given that (i) CCR5-deficient CTLs also home relatively effectively into tumours with active antigen-specific engagement (as per above, Figure 5E,F), (ii) our results and reports by others implicating circuits other than the CCR5-dependent one in antitumour immunity (see further response below), and (iii) CCR5^-/-^ CTLs are fully effective at eliminating cognate antigen-presenting target cells (Figure 5—figure supplement 1H).

We are confident that the CCR5-dependent homotypic signalling circuit is operational and influences CTL recruitment into tumours in vivo, but it is only one such circuit among a complex array of chemotactic, pro-inflammatory and retention signals, and abrogation of this circuit alone does therefore not result in the impairment of tumour rejection over many days by adoptively transferred CTLs. The long time-scales and concerted immunological interactions involved in tumour rejections may over time mask the role of the homotypic CTL signalling circuit. Acute, local chemotactic effects occurring over a shorter time period may in fact be operational but are not reflected in macroscopic tumour volume data. In order to test whether CTLs can indeed effectively swarm towards a CCL3 source within the dense and complex landscape of real tumour tissue, we used intravital microscopy to interrogate the positioning of WT and CCR5^-/-^ OT1 CTLs up to 6h around an acute release of CCL3, locally delivered by stereotactic injection of a shear-reversible hydrogel containing the chemokine (Figure 5—figure supplement 2). Indeed, we found a CCR5-dependent accumulation of CTLs in the tumour tissue around the delivered CCL3.

We also agree with the reviewers that without performing experiments with CCL3/CCL4-deficient CTLs, which we do not have timely access to, we cannot rule out that increased recruitment into tumours containing cognate CTLs is also caused by other CCR5 ligand-secreting cell types. Therefore, we have followed the reviewer’s advice to amend the text and remove the unsupported claim that “sustained production of CCL3 and CCL4 by activated CTLs is sufficient to induce swarming and tumour infiltration”, also in line with point 2C below. We now state that “sustained release of CCL3 and CCL4 is sufficient to promote CCR5-dependent homing into tumours in vivo, and that the presence of tumour-reactive CTLs in a tumour promotes the recruitment of distant CCR5^+^ CTLs. Homotypic CTL recruitment via CCR5 is however likely not the only or primary signalling circuit that results in CTL tumour infiltration or clearance.”

In a first experiment in *Rag^-/-^RAG^-/-^*mice, before we had conclusively identified CCR5 as the receptor mediating the homotypic signal, we had interrogated the ability of OT1 CTLs to home to CCL3/4-secreting tumours compared to non-secreting tumours engrafted on the contralateral flanks (previously Figure 4—figure supplement 2I, now Figure 4—figure supplement 4A). In parallel, we also performed an experiment to evaluate endogenous leukocyte recruitment to the CCL3/4-secreting tumours in PTPRCA host mice via CD45.1 staining (previously Figure 4—figure supplement 2J, now Figure 4—figure supplement 4B), in the absence of CTL transfer. The difference between these experiments was indicated in the legends but was not clear in the main text, for which we apologise. We have now amended the text to better distinguish the different in vivo experiments. We have also followed the reviewers’ suggestion and reported absolute cell counts per tumour, and included a new experiment evaluating endogenous dendritic cell recruitment into CCL3/4-secreting tumours. In Figure 4—figure supplement 4, this new analysis indicates a significant (p=0.035) increase in NK cells within CCL3/4-secreting tumours, but not other cell types, including dendritic cells.

Subsequently, to investigate the role of CCR5 in tumour recruitment, we co-transferred WT OT1 CTLs and CCR5^-/-^ OT1 CTLs (the “CCR5- null CD8b T cells” requested by the reviewers)(previously Figure 5F, now Figure 5—figure supplement 3B) in mice bearing CCL3/4-secreting tumours on one flank, and non-secreting tumours on the contralateral flank. These experiments showed impaired recruitment of CCR5^-/-^ CTLs into CCL3/4-secreting tumours. With these experiments, we aimed to study how T cells and leukocytes in general respond to the CCR5-dependent homotypic signalling axis, and not the CCR5-dependence to elicit such a signal.

We acknowledge the prior studies that reported potentially homotypic chemokine signalling in secondary lymphoid organs. Here, we demonstrated the action of the CCR5-CCL3/4 signalling circuit in the recruitment of CTLs to tumours, in particular with the 2-step adoptive transfer experiment where a secondary population of non-cognate CTLs homes in a partially CCR5-dependent manner towards tumours containing previously transferred cognate CTLs (new Figure 5E,F). We thank the reviewers for pointing out the papers that highlight the role of the alternate chemokine signalling circuit centred on CXCR3 in T cell antitumour immunity. In the mentioned studies, chemokine receptor-mediated tumour trafficking and infiltration were shown to be dependent on CXCR3 and BLT1/Ltbr4 using a variety of experiments involving adoptive transfer of receptor-deficient CTLs into immune competent or Rag^-/-^RAG^-/-^ mice, tumour engraftment of receptor- or chemokine-deficient hosts, and chemokine blockades (Chheda et al., 2016; Chow et al., 2019). In fact, two other publications report similar findings using adoptive transfer models most similar to our experimental design (Mikucki et al., 2015; Sharma et al., 2013). Sharma et al., 2013 and Mikucki et al., 2015 showed impaired homing of CXCR3- or BLT1-deficient tumour-experienced CD8^+^ T cells that were adoptively transferred into tumour-bearing Rag^-/-^RAG^-/-^ hosts compared to WT CD8^+^ T cells, resulting in impaired tumour clearance. All of these studies used B16 melanoma tumour models. We had carefully selected the EL4 tumour model as these cells do not themselves secrete chemokines that elicit a directional response in CTLs (see Figure 2, Figure 3C-E, Figure 3—figure supplement 1 B,C, and Figure 4—figure supplement 1 C). In fact, we had considered using the B16 model but when we tested chemokine expression via qPCR, we found that our B16 cancer cells secrete significant levels of chemokines CCL2, CCL3, CCL4, CCL7, CXCL9 and CXCL10, and therefore judged that tumour model to be unsuitable to study homotypic recruitment between T cells.

**Author response image 1. sa2fig1:** mRNA expression levels of various chemokines in B16 tumour cells relative to EL4 tumour cells as measured by qPCR.

The redundant role of CCR5 in tumour clearance in our model may be due to a switch to alternative receptor-ligand pairs. OT1 CTLs also express CCR1, CCR2, CXCR3, XCR1 and BLT1 (Figure 5—figure supplement 1D and Supplementary file 1), and their ligands may be generated by various stromal or tumour infiltrating cells. Tumour cell lysis may also contribute to the attraction of neutrophils that self-amplify production of LTB4, the BLT1 ligand (Lammermann et al., 2013).

Furthermore, IFNγ, abundant in inflamed tumours, can induce CXCL9 and CXCL10 secretion in various intermediary cells, including tumour-infiltrating myeloid cells (Dobrzanski, Reome and Dutton, 2001; Gordon-Alonso et al., 2017; Hickman et al., 2015), which may enhance the role of CXCR3 in homing. These points are briefly discussed in the discussion section.

2B) Given the observation of peripheral rimming of T cells in the in vitro studies in the absence of CCL3/4 and invasion with these mediators operative, transcriptomic datasets from human samples that are classified as cold (excluded) and hot (infiltrated) would be useful to probe to see if there is a difference in CCL3/4 mRNA expression in the two cases, and would be even better if this involved single cell work that ascribed production to the CD8 T cells themselves. This would be in silico work to see if the right data sets exist and if mined, show the expected outcome, although it is possible, of course, that no such differences may be identifiable in CCL3/4 expression between hot and cold tumors, which would neither invalidate nor support a potential role played by these chemokines in CD8 T cell swarming in tumours.

We thank the reviewers for suggesting this idea, which led us to a number of studies where transcriptomic data from ‘hot’ tumours show significant CCR5 ligand expression, including CCL3 and CCL4, by tumour-infiltrating T cells.

In a 2015 study, metastatic melanomas from 197 cancer patients were grouped according to low (cold) vs high (hot) expression of T cell signature genes. Gene expression profiling indicated that activation of the β-catenin signalling pathway in cold tumours was associated with CCL3 and CCL4 down-regulation and reduced T cell infiltration (Spranger, Bao and Gajewski, 2015).

More recently, Jerby-Arnon and colleagues studied 33 human melanoma tumours and obtained 7,186 single cell transcriptomic (scRNA-seq) profiles of both malignant and non-malignant cells within the tumours (Jerby-Arnon et al., 2018).

The main finding of the study is that malignant cells of ‘cold’ tumours express a resistance program that excludes T cells from the tumours. In ‘hot’ tumours, infiltrated by T cells, CCL3 and CCL4 expression was strongly associated with CD8^+^ T cells (Author response image 2). At the single cell level, CCL3 and CCL4 were found to be significantly expressed by tumour-infiltrating CD8^+^ T cells, as well as by NK cells and macrophages (Jerby-Arnon et al., 2018).

In an even more recent study of patients with B cell non-Hodgkin lymphomas, in malignant lymph nodes, intratumoural CTLs were specifically found to express CCL4 and CCL5 (both CCR5 ligands) by scRNA-seq (Roider et al., 2020).

Finally, a report on head and neck squamous cell carcinoma patients found that hot tumours display significantly higher CTL activity and increased CTL:Treg ratios, as well as upregulation of several chemokines and chemokine receptors, including CCL5 and CCR5 (Chakravarthy et al., 2018).

Taken together, these reports support a role in CD8^+^ T cell tumour infiltration for CCR5 ligands, produced by both malignant cells and tumour-infiltrating T cells, and which in addition to CCL3 and CCL4 may also include CCL5 in humans. We now cite and discuss the above-mentioned studies in the Discussion section.

2C) In the absence of data from additional work as in 2A above, and/or if efforts in 2B above do not show any informative correlation, it is recommended that the paper be re-written to focus on effector CD8 T cells driving concerted recruitment to sites of antigen recognition, with the underlying mechanism carefully documented, and with only the discussion suggesting that this might be a mechanism active in the in vivo tumor setting, making clear that future experiments will show whether this mechanism is indeed prominent in primary tumors in vivo.

We thank the reviewers for prompting us to revise the text regarding the role of homotypic CTL recruitment towards sites of antigen recognition versus tumour infiltration. In light of the reviewers’ suggestion, and our new data on tumour infiltration and rejection from above, we have elected to err on the side of caution and carefully restrict our language on the homotypic recruitment mechanism to act towards antigen specific targets in general, rather than playing a major role in tumour infiltration in vivo. We specify explicitly what role CCR5-mediated recruitment has in our intravital imaging and in vivo tumour homing experiments, without ascribing the effects to homotypic signalling directly, instead discussing the potential for a contribution of such a direct signalling circuit amongst others. We have already indicated some of these changes above in the context of responses to specific comments on the in vivo work. We have for instance amended the language as follows:

- In the Abstract, we changed “Thus, CTLs recognising a tumour target can induce a localised mass response by amplifying the direct recruitment of additional T cells independently of other leukocytes” to “Thus, CTLs recognising a cognate target can induce a localised mass response by amplifying the direct recruitment of additional T cells independently of other leukocytes.”

- In the Introduction, we changed “Furthermore, we show that sustained secretion of CCL3 and CCL4 mediates the swarming behaviour and can promote deep tumour infiltration of primary mouse CTLs” to “Furthermore, we show that local chemokine delivery triggers directed CTL movement through dense tumour tissue in vivo, and sustained secretion of CCL3 and CCL4 from tumours promotes CTL recruitment”.

- In our multistep recruitment model (Figure 7), we now depict and refer more generally to “cognate targets” or simply “targets” rather than “cognate tumour cells”.

- In the Discussion section, we state that “it is also evident that CCR5-mediated homotypic recruitment is not the exclusive signalling circuit underpinning tumour infiltration in vivo, and in fact CCR5^-/-^ CTLs were not impaired in their ability to clear tumours in mice”.

[Editors' note: further revisions were suggested prior to acceptance, as described below.]

Essential revisions:1) The authors need to clarify the results in Figure 2B (right panel) and Figure 2F. In Figure 2B, the exogenously added CTLs exhibit swarming (increased FMI) towards the cognate tumour even in the absence of pre-embedded antigen-specific CTLs. However, in Figure 2F, exogenously added CTLs do not exhibit swarming unless there are pre-embedded antigen-specific CTLs. What is the cause of this discrepancy?

We thank the reviewer for their perceptiveness and for raising the lack of clarity in the difference between the two above-mentioned experiments. In order to keep the volume and density of tumouroids constant, they are always constituted by similar total number of cells, whether they contain tumour cells alone, or a combination of CTLs and tumour cells. In Figure 2F, all tumouroids contain pre-embedded CTLs at a 1:1 ratio with the tumour cells. Hence, these tumouroids contain only half the number of tumour cells compared to the samples of Figure 2B where the tumouroids contain no pre-embedded CTLs. In the short timespans of these experiments (1hour), the probability of surrounding CTLs arriving at the tumouroid via random search, engaging cognate tumour cells and inducing long-range attraction in distant CTLs is evidently diminished when the tumour cells are interspersed with non-tumour reactive CTLs (Figure 2F), compared to scenarios where the tumouroid is entirely made up of cognate tumour cells (fourth from left, Figure 2B). We have now clarified the discrepancy in the figure legend. It should be noted that the experiments of Figure 2F where designed to evaluate whether a pre-embedded CTL population engaging a cognate target can induce the directional recruitment of distant CTLs of different specificities, and not to evaluate whether first arriving randomly searching CTLs can induce directional movement in other CTLs of the same population, an effect appreciable in Figure 2B but best conveyed in the longer time-course experiments with central tumouroids in Figure 1 and Figure 1—figure supplement 1.

2) Subsection “CTL recruitment is driven by secretion of CCL3 and CCL4” – Clarification is required. The authors state "only CCL3 and CCL4 inhibition disrupted CTL transmigration towards cognate supernatants." However, in Figure 4—figure supplement 1F, CCL4 blocking Abs alone have no effect on transmigration, despite that fact that both CCL3 and CCL4 blocking antibodies are required to block directional migration in Figure 4G. Can the authors provide an explanation for this discrepancy in the text?

We thank the reviewer for highlighting this lack of clarity in the text. Migration across transwells (Boyden chambers) towards cognate supernatant was abrogated by CCL3 inhibition but not by CCL4 inhibition (now clarified in the text), whereas migration within 3D collagen gels towards cognate tumoroids was inhibited only when both CCL3 and CCL4 were blocked. The redundant activities of CCL3 and CCL4 is apparent only in the 3D collagen system, which is a more physiological assay than migration across transwells, mainly employed as a screening tool to identify chemokines of interest. CTLs produce about ~2.5 fold more CCL3 than CCL4 during cognate interactions (Figure 4—figure supplement 1C). In addition, at similar concentrations, CCL3 appears to be more potent than CCL4 in inducing CTL transmigration in transwells (Figure 4—figure supplement 1D), which may further account for observed differences between the two systems.

3) Figure 5F – OT1 WT and OT1 CCR5-/- both exhibit a large increase in recruitment to the tumour when gBT1 cells are pre-transferred. This observation suggests that CCL3 and CCL4 actually play a very minor role in T cell "swarming" in this in vivo system. The authors should alter the wording of the abstract since it currently states that "CTLs engaging cognate targets accelerate the recruitment of distant T cells through long-range homotypic signalling via the diffusion of secreted chemokines CCL3 and CCL4". This statement suggests that CCL3/CCL4 play a major role in T cell swarming, but this claim isn't fully supported by the in vivo data.

We have now amended the abstract to state that recruitment through homotypic signalling is only partly mediated by CCL3 and CCL4, to more accurately represent both the in vitro and in vivo data. The sentence now reads: “CTLs engaging cognate targets accelerate the recruitment of distant T cells through long-range homotypic signalling, in part mediated via the diffusion of chemokines CCL3 and CCL4”. In light of our in vivo findings, we had already stated in the Results section that “Homotypic CTL recruitment via CCR5 is however likely not the only or primary signalling circuit that results in CTL tumour infiltration or clearance”, and in the Discussion section that “the presence of antigen-specific CTLs within a tumour promotes the recruitment of distant CTLs into the tumour in a manner partially dependent on CCR5”.

4) Figure 5 and accompanying supplements – The small differences observed between WT and CCR5-/- CTLs in vivo, both in terms of recruitment and tumour killing, could be due to the fact that the authors co-transferred 5x106 of each CTL into recipient mice. This high transfer number may diminish the role of CCR5/CCL3/CCL4 mediated recruitment of CTLs into the tumour in vivo. At the very least, the authors should comment on this issue in the text.

We thank the reviewer for highlighting this point, which we now comment on in the discussion with relevant references. It is indeed likely that different parameters across tumour models (e.g. numbers of T cells transferred, when they are transferred with respect to tumour engraftment, their state of activation, tumour growth, antigenicity and tumour composition) influence the kinetics of T cell infiltration and tumour clearance, as well as chemokine receptor dependence. We now state in the Results section that “It remains to be explored whether adoptive transfers using lower CTL numbers or earlier following tumour engraftment (Sharma et al., 2013, Chheda et al., 2016) would result in a more dominant role for CCL3 and CCL4-mediated homotypic signalling in recruitment into solid tumours.”

5) Figure 5—figure supplement 2 – This figure would be even more convincing if the authors showed the distribution of the WT and CCR5-/- CTLs in a cognate tumour without an exogenous source of CCL3.

The aim of this experiment was to use intravital real-time microscopy to investigate whether long-range, CCR5-dependent CTL recruitment can occur within the dense and complex tissue of real tumours. The acute injection of exogenous CCL3 into the non-cognate tumours allowed us to exert temporal control over the initiation of such a recruitment, and to subsequently compare the dynamics of WT and CCR5^-/-^ CTLs. This would not have been feasible in cognate tumour tissue within the time constraints imposed by imaging live animals, because CTLs arrest upon engagement of cognate target cells and demonstrate severely reduced motility, especially during the earlier phases of tumour rejection that we studied (Boissonas et al., 2007).

6) Subsection “CTL recruitment is driven by secretion of CCL3 and CCL4” – Figure 4—figure supplement 4A-C. Overexpression of CCL3/4 in the tumors seems to have no effect (at least statistically) on the recruitment of OT1 CTLs or endogenous CTLS into the tumor in vivo. These observations seem to contradict the main point of the paper. The authors should provide a convincing explanation.

We thank the reviewer for highlighting this point, which we in fact tried to address since the last submission. In Figure 4—figure supplement 4A a couple of the adoptive transfers had in essence failed, with only very scarce cells found in both contralateral tumours, but we nevertheless included those results for the sake of transparency and completeness. We had since repeated the experiment with 4 additional mice, and have now added these additional data to the figure and performed associated statistical tests, which show that CCL3/4-secreting tumours indeed do recruit more OT1 CTLs than contralateral non-secreting tumours (p=0.0408 for all data, n=10 mice). As to Figure 4—figure supplement 4C, that experiment was performed with only 3 mice in response to a reviewer request to analyse dendritic cell infiltration of the tumours, and given additional capacity in our flow antibody panel, we had also counted endogenous CD8^+^ T cells. As is evident from the figure, in 2 out of the 3 mice, the CCL3/4-secreting tumours recruited more endogenous CD8^+^ T cells. We have not performed additional repeats of this particular experiment as we believe that the figure is clear even in the absence of the statistics that would accompany a larger cohort study, and combined with the above-mentioned new figure, the data conclusively show that more T cells infiltrate the CCL3/4-secreting tumours.